# BHLHE40/41 regulate microglia and peripheral macrophage responses associated with Alzheimer's disease and other disorders of lipid-rich tissues

Anna Podleśny-Drabiniok [1,9], Gloriia Novikova[1,8,9], Yiyuan Liu[1], Josefine Dunst [2,3], Rose Temizer [1], Chiara Giannarelli [4,5], Samuele Marro[6,7], Taras Kreslavsky [2,3], Edoardo Marcora [1,6,10] ✉ & Alison Mary Goate [1,6,7,10] ✉

Genetic and experimental evidence suggests that Alzheimer's disease (AD) risk alleles and genes may influence disease susceptibility by altering the transcriptional and cellular responses of macrophages, including microglia, to damage of lipid-rich tissues like the brain. Recently, sc/nRNA sequencing studies identified similar transcriptional activation states in subpopulations of macrophages in aging and degenerating brains and in other diseased lipid-rich tissues. We collectively refer to these subpopulations of microglia and peripheral macrophages as DLAMs. Using macrophage sc/nRNA-seq data from healthy and diseased human and mouse lipid-rich tissues, we reconstructed gene regulatory networks and identified 11 strong candidate transcriptional regulators of the DLAM response across species. Loss or reduction of two of these transcription factors, BHLHE40/41, in iPSC-derived microglia and human THP-1 macrophages as well as loss of Bhlhe40/41 in mouse microglia, resulted in increased expression of DLAM genes involved in cholesterol clearance and lysosomal processing, increased cholesterol efflux and storage, and increased lysosomal mass and degradative capacity. These findings provide targets for therapeutic modulation of macrophage/microglial function in AD and other disorders affecting lipid-rich tissues.

Tissue-resident and monocyte-derived macrophages are myeloid cells that specialize in the phagocytic clearance of host tissue-derived cellular debris (efferocytosis). This core function of macrophages is essential for the maintenance of tissue homeostasis and immune tolerance, the resolution of inflammation, and the repair of damaged tissue. Macrophage dysfunction has been implicated in aging and the pathogenesis of numerous diseases, including Alzheimer's disease (AD), demyelinating disorders, schizophrenia, obesity, steatosis, atherosclerosis, and several other autoimmune and chronic inflammatory diseases[1-6].

In AD, genetic and experimental evidence strongly implicates myeloid cells (including microglia, the brain-resident macrophages) in disease pathogenesis and suggests that AD-associated alleles and genes may modulate disease risk by altering the transcriptional and cellular responses of macrophages (like microglia) to damaged lipid-rich tissues (such as the aging or degenerating brain, the most lipid-rich organ of the human body[7]). Specifically, recent single-cell/nucleus RNA sequencing (sc/nRNA-seq) studies identified similar transcriptionally activation states in subpopulations of macrophages in aging and degenerating brains (often referred to as disease-associated

microglia or DAM) and in other diseased lipid-rich tissues (e.g., TREM2[hi] macrophages in atherosclerotic plaques, and lipid-associated macrophages/LAMs in fatty liver and obese adipose tissue)[6,8–10]. For brevity in this manuscript, we collectively refer to these subpopulations of microglia and peripheral macrophages as DLAMs.

Importantly, DLAMs are characterized by increased expression of genes involved in the phagocytic clearance of lipid-rich cellular debris (efferocytosis), including several AD risk genes[11,12]. Some of these AD risk genes (*TREM2*, *PLCG2*, and *APOE*) are critical for the development and function of DLAMs in response to lipid overload due to host tissue damage in mouse models of AD, demyelination, and other disorders of lipid-rich tissues[8,13–15]. Therefore, a deeper understanding of the transcriptional regulation of the DLAM response may offer valuable biological insights and aid the prioritization of therapeutic targets for AD and other disorders of lipid-rich tissues.

The DLAM response, like other transcriptional and cellular responses, is likely regulated by transcription factors (TFs) that orchestrate the coordinated expression of DLAM response effector genes through direct binding to their promoters and other *cis*-regulatory elements. Reconstruction of TF-centric gene regulatory networks (GRNs) can lead to the identification of TF target genes and co-regulated gene modules (regulons)[16]. In turn, enrichment of DLAM genes in these regulons can point to candidate TFs that are likely to direct the DLAM response[16]. Integration of multiple microglial bulk microarray datasets, including transcriptome profiles of microglial cells acutely isolated from AD mouse brains led to the identification of potential regulators of several microglial transcriptional states and identified *Cebpα*, *Irf1*, and *Lxrα/β* as positive transcriptional regulators of DLAM markers such as *Apoe*, *Cxcr4*, and *Trem2*[17,18]. Another group characterized microglia co-expression modules and found that *Bhlhe40*, *Rxry*, *Hif1α*, and *Mitf* were coexpressed in modules of neurodegeneration- and demyelination-related microglia that resemble the DAM transcriptional state[19].

Here, we used sc/nRNA-seq data of human and mouse macrophages from healthy and diseased brains and other lipid-rich tissues to reconstruct GRNs and identify TFs whose regulons are enriched for DLAM genes (DLAM TFs) across species, diseases and tissues. We nominate 11 strong candidate DLAM TFs shared across human and mouse networks (*BHLHE41*, *HIF1A*, *ID2*, *JUNB*, *MAF*, *MAFB*, *MEF2A*, *MEF2C*, *NACA*, *POU2F2*, and *SPI1*). We also demonstrate a strong enrichment of AD risk alleles in the cistrome of *BHLHE41* (and its close homolog *BHLHE40*), thus implicating its regulon in the modulation of disease susceptibility. Loss or reduction of *BHLHE40/41* expression in human THP-1 macrophages and iPSC-derived microglia as well as loss of *Bhlhe40/41* in mouse microglia led to increased expression of a subset of DLAM response genes, specifically those involved in cholesterol clearance and lysosomal processing, with a concomitant increase in cholesterol efflux and storage, as well as lysosomal mass and degradative capacity. Taken together, this study nominates transcriptional regulators of the DLAM response, experimentally validates BHLHE40/41 in human and mouse microglia and THP-1 macrophages, and provides targets for therapeutic modulation of macrophage/microglial function in AD and other disorders of lipid-rich tissues.

## Results

### Reconstruction of gene regulatory networks in human and mouse macrophages/microglia using single-cell/nucleus RNA sequencing data

To reconstruct GRNs and nominate candidate transcriptional regulators of the DLAM response (henceforth referred to as DLAM TFs), we used eight human sc/nRNA-seq and ten mouse scRNA-seq macrophage/microglia datasets from healthy and diseased brains and other lipid-rich tissues (Methods, Supplementary Dataset 1). Human datasets included microglia from healthy and AD brains, macrophages from atherosclerotic plaques, macrophages from adipose

tissue of healthy and obese individuals, and Kupffer cells from healthy and cirrhotic livers[6,10,20–25]. Mouse datasets included microglia from control mice and mouse models of amyloid deposition (i.e AppNL-G-F, APP/PS1, and 5XFAD), microglia from aging and demyelinating brains (i.e., upon cuprizone treatment), macrophages from atherosclerotic plaques, macrophages from adipose tissue of healthy and obese mice, and Kupffer cells from healthy, non-alcoholic steatohepatitis (NASH), and cirrhotic livers[6,9,14,20,21,26–29]. GRNs were reconstructed by computing mutual information between annotated transcriptional regulators (642 human and 641 mouse) and all other expressed genes in each dataset using ARACNe[30] (Methods). To filter out likely indirect TF-target interactions, we used data processing inequality (DPI) in ARACNe, thus enriching the regulon for potential direct targets[30] (Methods).

### Geneset enrichment analysis of TF regulons nominates candidate DLAM TFs in human and mouse macrophages/microglia

To nominate candidate DLAM TFs, we tested the regulons of all TFs in human and mouse GRNs for over-representation of DLAM genesets from three different studies: 1) DAM microglia from brains of 5XFAD mice (Table S2 from ref. [8]); 2) TREM2[hi] macrophages from atherosclerotic plaques of *Ldlr*[−/−] mice on high-fat diet (Online Table I, cluster 11 from ref. [9]) and 3) lipid-associated macrophages (LAM) from human and mouse obese adipose tissue (Dataset S6 and Dataset S5 from ref. [6]) (Methods and Supplementary Dataset 1). This approach allowed us to identify TFs whose regulons displayed a statistically significant enrichment for these DLAM genesets, suggesting they may at least partially regulate these transcriptional responses. Since DLAM response genes are partially conserved across species[25], we focused on seventy-four DLAM TFs that were nominated in at least two mouse and two human GRNs (Fig. 1A, B).

To generate a shortlist of DLAM TFs, we required each transcription factor to be nominated in at least half of human and mouse GRNs and to be expressed in human microglia[31]. Using this more stringent set of criteria, we nominated 11 genes (*BHLHE41*, *HIF1A*, *ID2*, *JUNB*, *MAF*, *MAFB*, *MEF2A*, *MEF2C*, *NACA*, *POU2F2,* and *SPI1*) as strong candidate DLAM TFs in both human and mouse for further analyses. We were unable to identify a candidate transcriptional regulator for a small number of DLAM genes, while some DLAM genes were predicted to be regulated by only a handful of TFs in both human and mouse networks (Fig. 1A, B). A potential explanation for this observation could be that the low expression of these genes in macrophages/microglia prevented us from identifying robust associations. Indeed, some of these genes, such as *CD5*, *CXCL14*, and *KIF1A*, have very low expression in microglia[31] despite being significantly upregulated in DAM from 5XFAD mice[8,31].

Although we focused on TFs nominated in both human and mouse GRNs for our downstream analyses, we also identified transcription factors specifically nominated as DLAM TFs in one species but not the other (i.e., nominated in half of the mouse GRNs, but not in human GRNs and vice versa). For example, *Bhlhe40*, *Atf3*, and *Fli1* are nominated in mouse GRNs, while *CEBPD*, *CEBPB*, and *EGR1* are nominated in human GRNs (Supplementary Dataset 1). This is in agreement with a recently published study that nominated *CEBPB*, *CEBPD*, and *EGR1* as candidate drivers of a DAM-like transcriptional state observed in cultured human iPSC-derived microglia (iMGLs) exposed to cholesterol/lipid-rich cellular debris in vitro[32]. These differences between species could be driven by true differences in DLAM gene expression profiles, differences in the expression levels of these TFs, as well as technical differences between human and mouse datasets such as single-nucleus vs single-cell RNA-seq transcriptome profiling, microglia isolated from different human and mouse brain regions, and the varying quality of microglia isolated from post-mortem human brains vs freshly dissected mouse brains. Nevertheless, taken together, these analyses identified conserved and

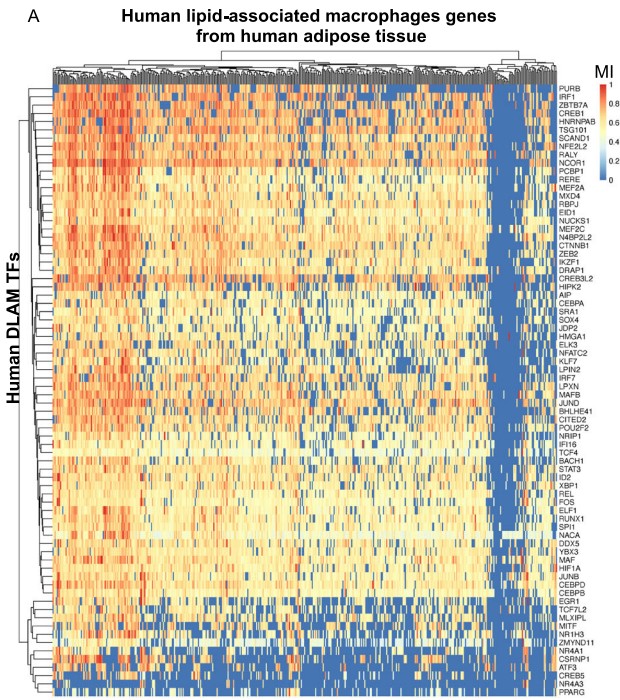

**Fig. 1 | Gene regulatory network analysis of sc/snRNA-Seq datasets from human and mouse macrophages/microglia nominates 74 transcriptional regulators of DLAM genes (DLAM TFs). A** Heatmap showing normalized mutual interaction (MI) values for each DLAM gene - DLAM transcription factor (DLAM TF) pair in a meta-analyzed human network (8 human networks were meta-analyzed). Human LAM genes are from adipose macrophages from ref. 6. (Dataset S6, FDR Adj.*P* value <

0.05). **B** Heatmap showing normalized MI values for each DLAM gene - DLAM TF pair in a meta-analyzed mouse network (10 mouse networks were meta-analyzed). Mouse DAM genes are from DAM microglia reported by ref. 8. (Table S2). The 74 DLAM TFs were nominated in at least two human and mouse networks, were conserved between species, and their regulons were significantly enriched for all three DLAM genesets (DAM, LAM and TREM2[hi], Supplementary Dataset 1).

species-specific putative transcriptional regulators of the DLAM response.

## Human genetics implicates candidate DLAM TFs BHLHE40/41 and their cistromes in the etiology of AD

Several AD risk genes (*APOE*, *PLCG2*, and *TREM2*) have been shown to be critical for the development and function of DLAMs in response to lipid overload due to host tissue damage in mouse models of AD and demyelination, and other diseases of lipid-rich tissues, highlighting a potential causal link between regulation of the DLAM response and modulation of AD risk[8,13–15]. Therefore, we investigated whether any of the candidate DLAM TFs nominated in this study could also be implicated in the modulation of AD risk. We previously showed that AD risk alleles are significantly enriched in the cistrome of PU.1 (encoded by *SPI1*, a candidate AD risk gene[33] as well as one of the candidate DLAM TFs nominated in this study), suggesting a role for SPI1/PU.1 regulated genes in the etiology of AD[33]. Since then, this finding has been independently replicated multiple times[34,35]. To investigate whether any of the other candidate DLAM TFs exhibit a similar enrichment, we used stratified LD score regression to perform partitioned AD heritability analysis using open chromatin regions (ATAC-seq peaks) that contain binding motifs for the TF of interest (henceforth referred to as TF proxy-binding sites) in microglia and macrophages, as previously described[4]. We could not perform these analyses for *NACA* and *ID2*, since these transcriptional regulators do not have a DNA binding motif in the HOMER database or lack a DNA binding domain[36]. We replicated the significant enrichment of AD risk alleles in SPI1/PU.1 proxy-binding sites in human microglia (*P* value = 6.9e-03) and monocyte-derived macrophages (*P* value = 1.9e-03) (Fig. 2A, B). Our approach using TF proxy-binding sites for partitioned heritability analysis has been previously validated by estimating a similar enrichment of AD risk alleles in SPI1/PU.1 binding sites obtained from SPI1/PU.1 ChIP-seq (as opposed to ATAC-seq) peaks in human monocytes and macrophages[33].

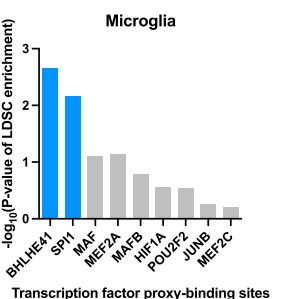
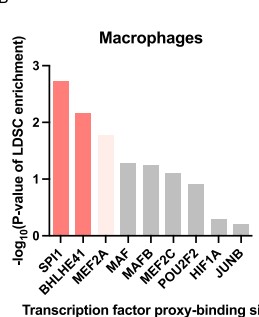

**Fig. 2 | Alzheimer's disease (AD) risk alleles are enriched in the BHLHE41 and SPI1/PU.1 cistromes. A** −log10 of the nominal enrichment *P* values obtained from the output of the stratified LD Score Regression (LDSC) analysis of AD GWAS SNP heritability partitioned by transcription factor (TF) proxy-binding sites, which were obtained by stratifying ATAC-Seq peaks in human microglia by the presence of binding motifs for each candidate LAM TF listed on the x-axis. **B** −log10 of enrichment P values obtained from stratified LDSC analysis of AD GWAS SNP heritability partitioned by TF proxy-binding sites, which were obtained by stratifying ATAC-Seq peaks in human monocyte-derived macrophages by the presence of binding motifs for each candidate DLAM TF listed on the x-axis. Bars in dark blue or dark red indicate significant enrichments (FDR Adj.*P* value < 0.05), bars in light red indicate nominally significant enrichments (*P* value < 0.05), while gray bars indicate non-significant enrichments.

In addition to SPI1/PU.1, we uncovered a significant enrichment of AD risk alleles in the proxy-cistrome of BHLHE41 in human microglia and monocyte-derived macrophages (*P* value = 2.2e-03 and 6.8e-03, respectively), implicating its target gene network in AD risk modification (Fig. 2A, B). In monocyte-derived macrophages, AD risk alleles were also nominally enriched in the proxy-cistrome of MEF2A (P value = 0.02) (Fig. 2B).

Interestingly, a recent AD GWAS in African-Americans uncovered a genome-wide significant locus near *BHLHE40*, a close homolog of *BHLHE41*[37]. Due to the high DNA binding motif similarity between BHLHE40 and BHLHE41 *(*similarity score of 0.99 in HOMER motif database*)*, BHLHE40 proxy-binding sites were also enriched in AD heritability in both microglia (*P* value = 3.7e-03) and monocyte-derived macrophages (P value = 5.8e-03). BHLHE40 and BHLHE41 (also known as DEC1/SHARP2/BHLHB2/STRA13 and DEC2/SHARP1/BHLHB3, respectively) belong to a family of basic helix-loop-helix transcriptional regulators that counteract transcriptional responses induced by other transcription factors like LXR:RXR nuclear receptors and members of the MiT/TFE family, which includes transcription factor EB (TFEB) and microphthalmia-associated transcription factor (MITF)[38–40]. In addition, BHLHE40 and BHLHE41 often exhibit functional redundancy[41–43]. Taken together, these results suggest that BHLHE40/41 and their target gene networks may modulate AD risk through their ability to regulate the DLAM response in microglia and other macrophages.

## BHLHE41 and SPI1/PU.1 likely regulate DLAM genes through direct binding to their promoters

Regulation of gene expression usually involves an interplay between multiple proximal and distal cis-regulatory elements and the gene promoter, mediated by specific three-dimensional chromatin arrangements[44]. These gene regulatory elements contain short and specific DNA sequences (motifs) to which TFs can bind to promote or repress transcription[44]. To further refine our list of candidate DLAM TFs, we investigated which of them likely regulate their predicted target genes directly by binding to their promoters. To address this question we used open chromatin (ATAC-seq) profiles of human and mouse microglia (Methods). First, for each DLAM TF, we quantified the number of instances of the corresponding binding motifs in open chromatin regions (ATAC-seq peaks) within the promoters of genes belonging to the DLAM TF regulon. We observed prominent peaks in *BHLHE41* and *SPI1* motif instances, suggesting our GRN reconstruction approach was indeed able to identify regulons that are enriched for direct targets of these DLAM TFs (Supplementary Fig. 1A and B).

To further explore the question of DLAM gene expression regulation, we stratified ATAC-seq peaks in human and mouse microglia by the presence of DLAM TF binding motifs and quantified the proportion of DLAM genes (ref. 6, Dataset S6, FDR Adj.*P* value < 0.05 for human; ref. 8, Table S2 for mouse) that contain DLAM TF proxy-binding sites in their promoters. BHLHE41 and SPI1/PU.1 proxy-binding sites were observed in the highest proportion of DLAM gene promoters in both human and mouse microglia and monocyte-derived macrophages (Fig. 3A–D, Supplementary Fig. 1A, B). Example epigenomic tracks highlighting open-chromatin regions that contain a BHLHE41 motif in DLAM gene promoters are shown in *CD63* and *CTSB* loci in mouse and human macrophages and microglia (Fig. 3E–F). This suggests that BHLHE41 and SPI1/PU.1 may be direct regulators of the DLAM transcriptional response, while other TFs identified by our analyses with a smaller proportion of proxy-binding sites in DLAM gene promoters might regulate fewer DLAM response genes and/or regulate them indirectly or by binding to other gene regulatory elements (e.g., enhancers).

## BHLHE40 and BHLHE41 are predicted to regulate DLAM genes involved in several pathways including cholesterol clearance and lysosomal processing

Since we and others have already been investigating the role of SPI1/PU.1 in microglia[33,34,45–47], we decided to focus our functional validation

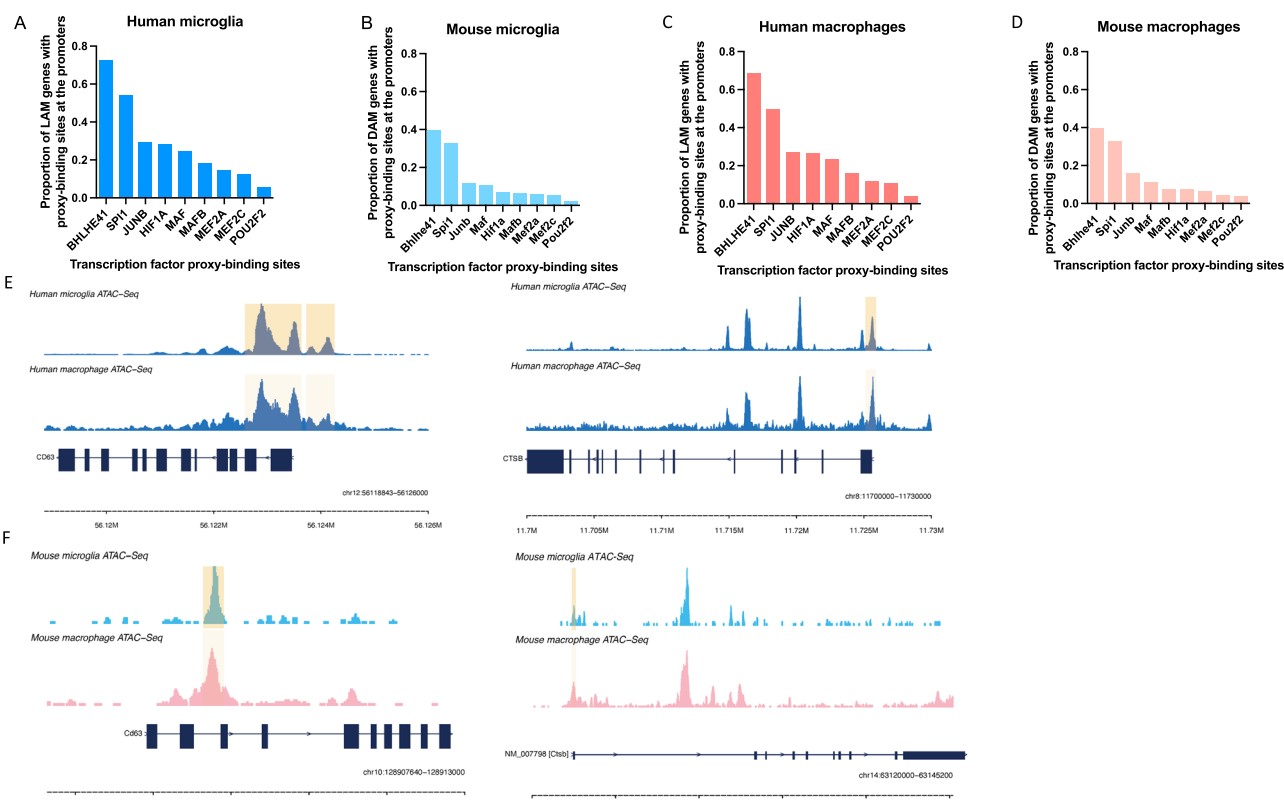

**Fig. 3 | BHLHE41 and SPI1/PU.1 likely regulate DLAM genes through binding to their promoters.** Proportions of DLAM genes that contain TF proxy-binding sites in their promoter in (**A**) human microglia, (**B**) mouse microglia, (**C**) human macrophages, (**D**) mouse macrophages, for each candidate DLAM TF listed on the x-axis. In A and C LAM genes are selected from ref. 6, Dataset S6, FDR < 0.05, in B and D DAM genes are selected from ref. 8, Table S2. **E** Human microglia and monocyte-derived macrophage open chromatin profiles in the *CTSB* and *CD63* loci with the regions containing a *BHLHE41* motif highlighted. **F** Mouse microglia and bone marrow-derived macrophage open chromatin profiles in the *Ctsb* and *Cd63* loci with the regions containing a *BHLHE41* motif highlighted.

efforts for this study on the other strong candidate DLAM TF BHLHE41 and its close homolog BHLHE40, which we (this study) and others[13,19,48] have nominated as a candidate transcriptional regulator of DAM and similar neurodegeneration- and demyelination-related microglial transcriptional responses in mice, and which resides in the vicinity of a locus recently associated with AD risk in a GWAS of African-American individuals[37].

To investigate which of the DLAM response-associated pathways are regulated by BHLHE40/41, we performed pathway analysis of 1) LAM genes ([6] Dataset S6, FDR < 0.05) 2) LAM genes with promoters containing BHLHE40 and/or BHLHE41 proxy-binding sites and 3) LAM genes belonging to the BHLHE40 and/or BHLHE41 regulons. We observed that BHLHE40 proxy-binding sites are a subset of BHLHE41 proxy-binding sites, thus we combined LAM genes in (2) and refer to them as "LAM genes with promoters proxy-bound by BHLHE40/41" (Supplementary Dataset 2). Additionally, we observed a significant overlap between LAM genes in the BHLHE40 and BHLHE41 regulons (P value = 1.59e-17). Hence, we combined LAM response genes in (3) and refer to them as "LAM genes in BHLHE40/41 regulons" (Supplementary Dataset 2).

To conduct pathway analysis we used Ingenuity Pathway Analysis (IPA, QIAGEN)[49] and identified pathways shared between human "LAM genes" (ref. [6], Dataset S6, FDR < 0.05), "LAM response genes with promoters proxy-bound by BHLHE40/41" and "LAM response genes in BHLHE40/41 regulons". They include "Phagosome maturation", "Oxidative phosphorylation", "Antigen presentation", "Glycolysis", "Atherosclerotic signaling", "LXR/RXR activation", and "mTOR signaling" (Supplementary Fig. 2A). Similarly, we have run pathway analysis to determine the overlap between mouse DAM (ref. [8], Table S2), "DAM genes in Bhlhe40/41 regulons", and "DAM genes with promoters proxy-bound by Bhlhe40/41". We have found similar sets of shared pathway which includes "EIF2 signaling", "mTOR signaling", "Oxidative Phosphorylation", "CLEAR signaling pathway", "Glycolysis", "LXR/RXR Activation", "Phagosome Maturation" (Supplementary Fig. 2B). Interestingly, these pathways are also regulated by other transcription factors such as LXR:RXR nuclear receptors and members of the MiT/TFE family TFEB and MITF, master regulators of cholesterol clearance and lysosomal processing, respectively[50]. These TFs are also known to directly target and stimulate the expression of BHLHE40/41 which in turn have been shown to act as repressors of transcriptional responses induced by LXR:RXR and MiT/TFE family TFs to form a negative feedback loop[38,39,51]. This suggests that regulation of the aforementioned DLAM response-associated pathways is shared between known regulators of these pathways and BHLHE40/41. In addition, DLAM responses seem to be regulated similarly by BHLHE40/41 in human and mouse

### Knockout of BHLHE40/41 partially recapitulates DLAM transcriptional responses in human iPSC-derived microglia (iMGLs)

To investigate the role of BHLHE40/41 as potential DLAM TFs we generated an isogenic set of CRISPR-edited *BHLHE40* and *BHLHE41* single and double knockout (40KO, 41KO, and DKO) iPSC lines (Methods and Supplementary Fig. 3A). Following microglial differentiation[52], we confirmed loss of BHLHE40 protein expression in 40KO and DKO iMGLs as well as loss of BHLHE41 protein expression in 41KO and DKO iMGLs by western blot (Supplementary Fig. 3B).

Next, we performed RNA-seq analysis to determine the genes and pathways that are altered in 40KO, 41KO, and DKO iMGLs compared to iMGLs derived from the parental (WT) iPSC line. We sequenced isogenic iMGLs of all four genotypes (WT, 40KO, 41KO, and DKO) across five independent microglial differentiation for each genotype. Results of differential gene expression and gene set enrichment analyses are included in Supplementary Dataset 3.

Since each candidate DLAM TF is predicted by our GRN analysis to strongly influence the expression of only a subset of DLAM genes, we did not expect that transcriptional changes associated with loss of BHLHE40 and/or BHLHE41 in iMGLs in vitro could largely recapitulate DLAM transcriptional responses observed in microglia and other macrophages in vivo. Indeed, transcriptional changes in all KO iMGLs are significantly and positively but only modestly correlated with the transcriptional signature of LAM macrophages from human obese adipose tissue published by ref. [6] (Dataset S6, FDR Adj.P value < 0.05; Spearman's correlation of ranks ($\rho$); $\rho_{40KO}$ = 0.13 [0.04, 0.22], $P$ value$_{40KO}$ = 0.003; $\rho_{41KO}$ = 0.19 [0.1, 0.27], $P$ value$_{41KO}$ < 0.0001; $\rho_{DKO}$ = 0.30 [0.21, 0.37], $P$ value$_{DKO}$ < 0.0001). Therefore, we used the rank-rank hypergeometric overlap (RRHO) analytical approach[53,54] (Methods) to more precisely identify statistically significant overlaps between transcriptional changes associated with loss of BHLHE40 and/or BHLHE41 (KO vs WT) in iMGLs and the aforementioned human LAM signature. We used an improved stratified RRHO method[54] because this method can identify and visualize statistically significant overlaps between two transcriptional signatures regardless of whether they change in the same (concordant) or opposite (discordant) direction. This analytical approach revealed statistically significant and concordant overlaps in the bottom-left quadrant (i.e., between genes upregulated in each pair of transcriptional signatures; hypergeometric overlap $P$ value: $P$ value$_{40KO}$ = 1.90e-3, $P$ value$_{41KO}$ = 2.28e-5, $P$ value$_{DKO}$ = 2.74e-9) and in the top-right quadrant (i.e., between genes down-regulated in each pair of transcriptional signatures; hypergeometric overlap $P$ value: $P$ value$_{40KO}$ = 5.30e-3, $P$ value$_{41KO}$ = 5.38e-9, $P$ value$_{DKO}$ = 1.40e-16) (Fig. [4]A, B). Interestingly, the most statistically significant and concordant overlaps were observed between the transcriptional signature of LAM macrophages from human obese adipose tissue and transcriptional changes that occur in DKO iMGLs (Fig. [4]A, B) suggesting that BHLHE40 and BHLHE41 may act together to regulate the expression of a subset of DLAM genes in human microglia. In addition, we have also shown that transcriptomic signatures of cluster 2 and 8 of iMGLs treated with lipid-rich brain phagocytic substrates in vitro[32] are highly concordant with the transcriptomic signature of DKO iMGLs (Supplementary Fig. 9A) suggesting that exposure to lipid-rich brain tissue debris or reduction of BHLHE40/41 are both able to induce a DLAM-like transcriptional state in vitro.

In addition to RRHO analysis, we performed gene set enrichment analysis (GSEA) analysis to investigate whether transcriptional changes that occur in DKO iMGLs are positively or negatively enriched for genes up- and down-regulated in human DLAMs from several studies: foamy macrophages from human atherosclerotic plaques by ref. [10], LAM macrophages from human obese adipose tissue by ref. [6], human iPSC-derived microglia treated with lipid-rich brain phagocytic substrates in vitro (cluster 2 and 8) by ref. [32], amyloid β-associated microglia from the brain of AD patients (cluster 10) by ref. [55], human iPSC-derived microglia xenotransplanted in the brain of 5XFAD mice (DAM cluster) by ref. [56]. We found that transcriptional changes that occur in DKO iMGLs are positively and negatively enriched for, respectively, genes up- and down-regulated in cluster 2 and 8 of human iPSC-derived microglia treated with lipid-rich brain phagocytic substrates in vitro[32] (Fig. [4]D). Similarly, we found a positive enrichment for genes up-regulated in 1) cluster 10 corresponding to amyloid β-associated microglia from the brain of AD patients (cluster 10)[55], 2) foamy macrophages from human atherosclerotic plaques[10], and 3) LAM macrophages from human obese adipose tissue[6]. Surprisingly, we found a negative and positive enrichment for, respectively, genes up- and down-regulated in the DAM cluster of human iPSC-derived microglia xenotransplanted in the brain of 5XFAD mice[56]. This may be due to the fact that human DAM microglia derived from iMGLs xenotransplanted in the brain of immunodeficient and humanized

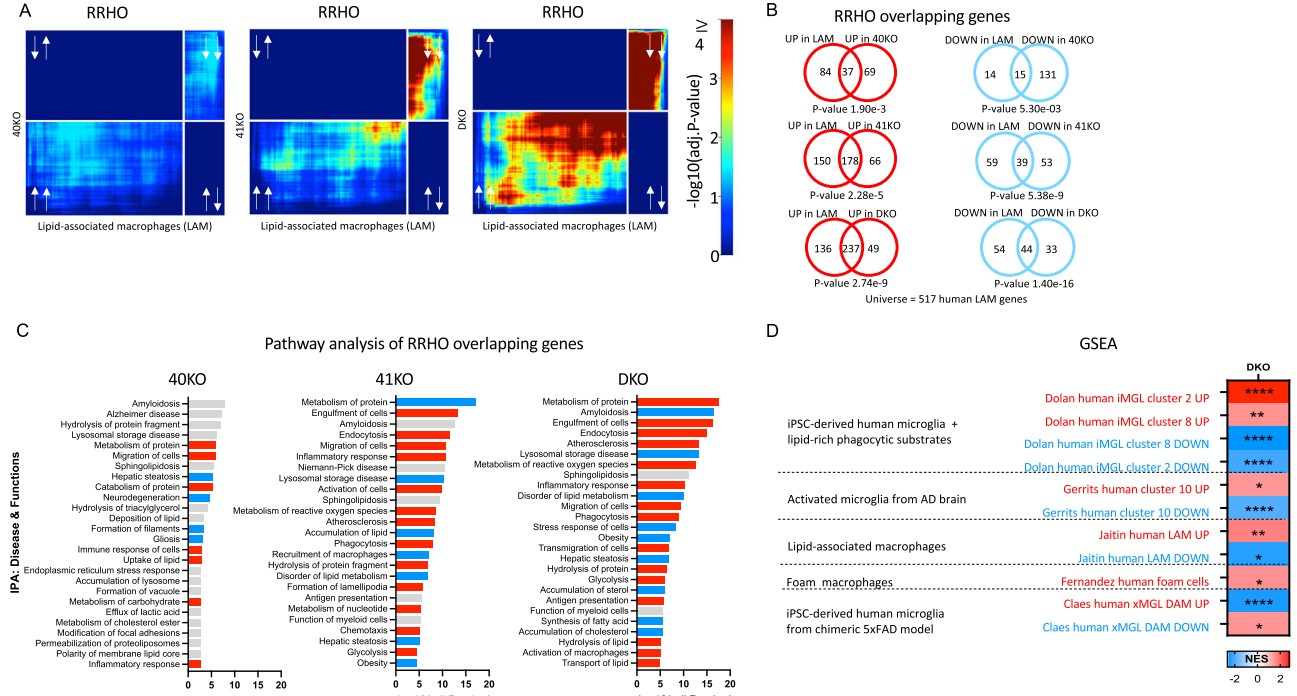

**Fig. 4 | Knockout of *BHLHE40/41* partially recapitulates DLAM transcriptional responses in human iPSC-derived microglia (iMGLs). A** Rank-rank hypergeometric overlap (RRHO) heatmaps visualizing significant overlaps in gene expression changes between each pair of BHLHE40/41 knockout (KO) iMGL and human LAM (Jaitin et al.[6], Dataset S6, FDR Adj.*P* value < 0.05) transcriptional signatures. Adj.*P* value in color temperature scale represents Benjamini-Hochberg corrected P value of hypergeometric overlap test. White arrows indicate whether genes are up- or downregulated. **B** Venn diagrams of most significant overlaps between genes upregulated in both *BHLHE40/41* KO iMGL and human LAM transcriptional signatures, corresponding to the warmest pixel in the bottom-left quadrant of the respective heatmap (left) and Venn diagrams of most significant overlaps between genes down-regulated in both BHLHE40/41 KO iMGL and LAM transcriptional signatures, corresponding to the warmest pixel in the upper-right quadrant of the respective heatmap (right). *P* values are calculated using the hypergeometric overlap test restricted to the universe of 517 human LAM genes (ref. 6, Dataset S6, FDR Adj.*P* value < 0.05). **C** Pathways ("diseases and biological functions" category) found by IPA to be significantly enriched for RRHO overlapping genes. Red bars represent pathways predicted by IPA to be more active (i.e., with positive Z-scores), blue bars represent pathways predicted by IPA to be less active (i.e., with negative Z-scores), gray bars represent pathways with non-attributed Z-scores. Adj.*P* value on the x-axis represents the Benjamini-Hochberg multiple hypothesis correction of *P* value calculated using right-tailed Fisher's Exact Test. 40KO = BHLHE40 KO iMGLs, 41KO = BHLHE41 KO iMGLs, DKO = BHLHE40 and BHLHE41 double KO iMGLs, all compared to iMGLs derived from the parental iPSC line. **D** Gene sets enrichment analysis (GSEA) of DLAM genesets in DKO transcriptome. DLAM genesets are listed in Supplementary Dataset 1 NES - normalized enrichment score, * FDR *q* value < 0.05, ** FDR *q* value < 0.01, **** FDR *q* value < 0.0001.

(hCSF) 5XFAD mice is distinct from mouse DAM microglia found in the brain of standard 5XFAD mice (discussed in ref. 57). Next, We used IPA to perform pathway enrichment and activity analysis of RRHO overlapping up- and down-regulated genes from Fig. 4B. Significantly enriched biological functions and diseases predicted to be more active (i.e., have positive Z-scores) in KO compared to WT iMGLs include "Endocytosis", "Engulfment of cells", "Transport of lipids", "Migration of cells", and "Antigen presentation". Significantly enriched biological functions and diseases predicted to be less active (i.e., have negative Z-scores) in KO compared to WT iMGLs include "Accumulation of cholesterol", "Obesity", and "Lysosomal storage disease" (Fig. 4C). Interestingly, these pathways point to increased activity of LXR:RXR and MiT/TFE family TFs which, as noted in the previous section, are master regulators of cholesterol clearance and lysosomal processing, respectively.

To further support these findings, we performed targeted differential gene expression analyses using RT-qPCR. We found that the expression of genes involved in cholesterol efflux/lipoprotein metabolism and lysosomal function was elevated in all KO iMGLs compared to WT iMGLs but reached statistical significance only in a few comparisons, likely due to low statistical power (Fig. 5A, Supplementary Fig. 4). Taken together these data suggest that loss of BHLHE40 and/or BHLHE41 induces the expression of DLAM genes involved in cholesterol clearance and lysosomal processing.

## Knockout of BHLHE40/41 increases cholesterol efflux and lipid droplet accumulation in iMGLs

Given that 1) loss of *BHLHE40/41* led to increased expression of LXR:RXR target genes (Fig. 5A, Supplementary Fig. 4), 2) pathway analysis indicated "LXR/RXR activation" is shared between BHLHE40/41 direct targets and DLAM genes (Supplementary Fig. 2), and 3) BHLHE40/41 have been shown to repress LXR:RXR target genes (including cholesterol efflux genes *APOE, LPL, ABCA1, ABCG1*)[38], we tested our KO and DKO iMGLs for LXR:RXR activation by measuring expression of LXR:RXR target genes *APOE* and *ABCA1*, as well as cholesterol efflux, in comparison to WT iMGLs. We first investigated whether the increased *APOE* and *ABCA1* mRNA levels that we observed using RT-qPCR (Fig. 5A, Supplementary Fig. 4) also resulted in increased protein levels. To this end, we performed western blot analysis and found significantly higher levels of APOE protein in 40KO iMGLs (effect size (ES) [95% confidence interval]; $ES_{40KO\text{-}WT} = 1.04$ [0.43, 1.68], *P* $value_{40KO\text{-}WT} = 0.0050$, $N = 3$/group), as well as elevated levels of ABCA1 in all KO iMGLs but reaching statistical significance only in 41KO iMGLs ($ES_{41KO\text{-}WT} = 0.42$ [0.21, 0.63], *P* $value_{41KO\text{-}WT} = 0.0021$, $N = 3$/group) (Fig. 5B, C). We also observed significantly increased levels of secreted APOE in conditioned media from 40KO iMGLs cultures, consistent with the significantly higher levels of *APOE* mRNA and protein observed in these cells (Fig. 5D) ($ES_{40KO\text{-}WT} = 5.76$ [2.03, 9.49], *P* $value_{40KO\text{-}WT} = 0.0048$, $N = 5$/group).

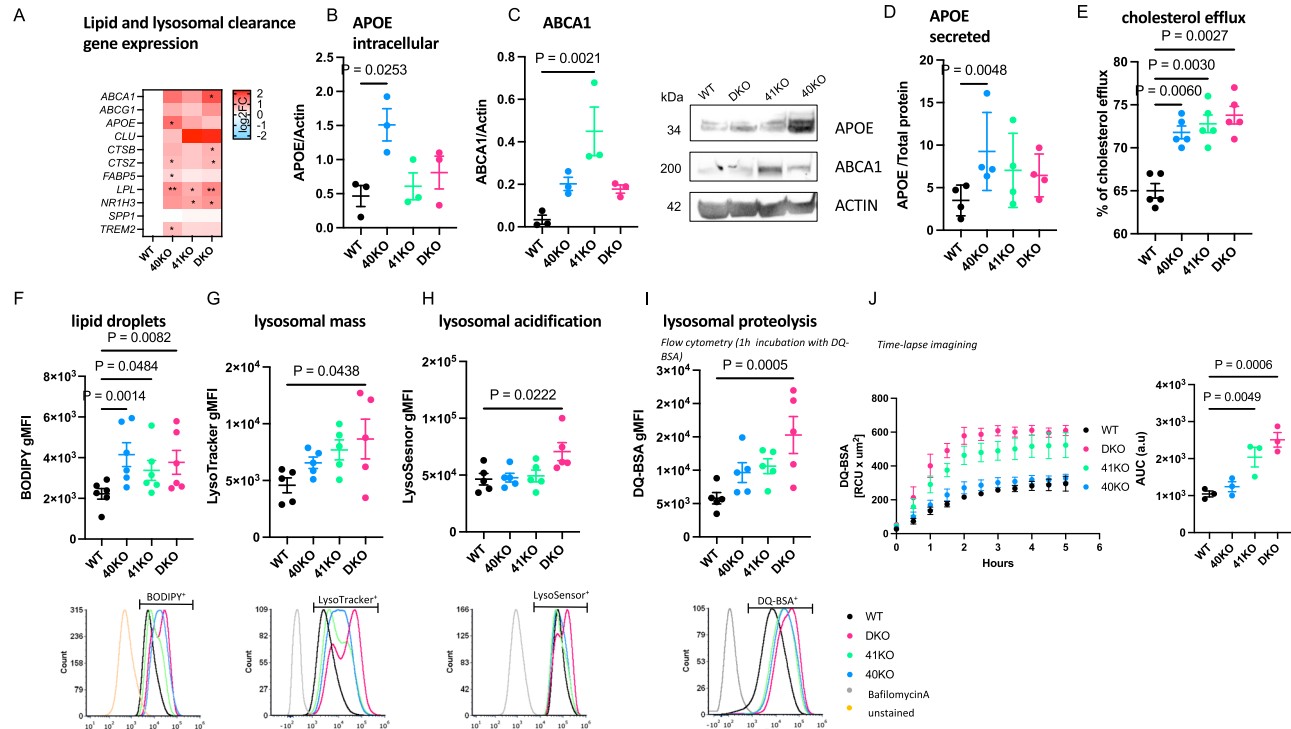

**Fig. 5 | Knockout of *BHLHE40/41* increases expression of lipid and lysosomal clearance genes, cholesterol efflux and lipid droplet content, lysosomal mass and degradative capacity in human iPSC-derived microglia (iMGLs).**
**A** Expression of lipid and lysosomal clearance genes measured by RT-qPCR, *N* = 5/group. log2(fold change) (log2FC) is calculated with WT iMGLs as reference. Separate plots for each gene in Supplementary Fig. 4. **B** Intracellular APOE normalized to Actin measured by western blot, *N* = 3/group. **C** ABCA1 normalized to Actin measured by western blot, *N* = 3/group. **D** Secreted APOE normalized to total protein measured by ELISA, *N* = 5/group. **E** Percentage of cholesterol efflux, *N* = 5/group. **F** LD content (BODIPY gMFI) measured by flow cytometry in BODIPY-positive cells (top), representative flow-cytometry histograms (bottom), *N* = 6/group. **G** Lysosomal mass (LysoTracker gMFI) measured by flow cytometry in LysoTracker-positive cells (top), representative flow-cytometry histograms (bottom), *N* = 5/group. **H** Lysosomal acidification (LysoSensor gMFI) measured by flow cytometry in LysoSensor-positive cells (top), representative flow-cytometry

histograms (bottom), *N* = 5/group. **I** Lysosomal proteolysis (DQ-BSA gMFI) measured by flow cytometry in DQ-BSA-positive cells (top) representative flow-cytometry histograms (bottom), *N* = 5/group. **J** DQ-BSA red fluorescent signal (total integrated density) measured over time using the Incucyte S3 live imaging system (left); quantification of area under curve (AUC) (right), *N* = 3/group. *N* represents the number of independent iMGLs differentiations. Flow cytometry gates were drawn based on FMO (fluorescence minus one) controls (F) or cells treated with Bafilomycin A (**G**–**I**) to prevent fusion of autophagosomes with lysosomes and thus obtain LysoSensor, LysoTracker, and DQ-BSA negative signals. Differences of means between groups were tested using one-way ANOVA with repeated measures followed by Dunnett's post-hoc test. **P* value < 0.05, ***P* value < 0.01 or indicated directly on the graph. Data plotted as mean ± SEM. Detailed statistics are shown in Supplementary File 1. 40KO = BHLHE40 KO iMGLs, 41KO = BHLHE41 KO iMGLs, DKO = BHLHE40 and BHLHE41 double KO iMGLs, WT = iMGLs derived from the parental iPSC line.

*APOE* and *ABCA1* are direct LXR:RXR target genes and are involved in cholesterol efflux and lipoprotein metabolism. Therefore, we measured the capacity of KO iMGLs to efflux cholesterol to an HDL acceptor. Cholesterol efflux capacity (measured as percentage of labeled cholesterol effluxed over 4 h, Methods) was increased in all KO iMGLs compared to WT iMGLs ($ES_{40KO-WT}$ = 6.80% [3.76%, 9.84%], *P* value$_{40KO-WT}$ = 0.0002; $ES_{41KO-WT}$ = 7.80% [4.76%, 10.84%], *P* value$_{41KO-WT}$ < 0.0001; $ES_{DKO-WT}$ = 8.80% [5.76%, 11.84%], $P_{DKO-WT}$ < 0.0001; *N* = 6/group), supporting our hypothesis that cholesterol clearance is enhanced by genetic inactivation of *BHLHE40* and/or *BHLHE41* in human microglia (Fig. 5E).

Another consequence of LXR:RXR activation that is also observed in DLAMs is increased accumulation of lipid droplets (LDs)[6,56]. Excess cholesterol delivered to macrophages by efferocytosis of lipid-rich cellular debris must be effluxed to extracellular acceptors like APOE or esterified for storage in intracellular LDs to prevent a cytotoxic buildup of free cholesterol[58]. To address the role of BHLHE40/41 in LDs accumulation, we took advantage of the fluorescent marker BODIPY which stains neutral lipids to measure LD content by flow cytometry. We found that all KO iMGLs showed increased LD content as evidenced by higher BODIPY geometric mean fluorescence intensity (gMFI) compared to WT iMGLs ($ES_{40KO-WT}$ = 1928 [783.7, 3072], *P* value$_{40KO-WT}$ = 0.0014; $ES_{41KO-WT}$ = 1151

[7.441, 2295], *P* value$_{41KO-WT}$ = 0.0484; $ES_{DKO-WT}$ = 1546 [401.8, 2690], *P* value$_{DKO-WT}$ = 0.0082; *N* = 6/group) (Fig. 5F). Of note, LDs accumulation in DKO iMGLs is similar to that observed in alveolar macrophages of *Bhlhe40/41* double knockout mice[59]. To determine whether increased LD content can be induced by LXR:RXR activation in iMGLs, we treated WT iMGLs with an LXR agonist (TO901317, 10 µM for 48 h) and observed higher BODIPY gMFI in these cells compared to WT iMGLs treated with vehicle as a control (Supplementary Fig. 5A). Conversely, we treated KO iMGLs with an LXR antagonist (GSK2033, 2 µM, 24 h) and saw that BODIPY gMFI in these cells was reduced to the levels observed in vehicle-treated WT iMGLs, indicating that LDs accumulation in *BHLHE40/41* KO iMGLs is caused by increased LXR:RXR activity (Supplementary Fig. 5B).

A subpopulation of hippocampal microglia from aged mice accumulate LDs and display a pro-inflammatory phenotype distinct from DLAMs[60]. To determine if increased LD content in KO iMGLs is accompanied by increased secretion of pro-inflammatory cytokines, we measured levels of secreted cytokines in conditioned media. We found significantly lower levels of pro-inflammatory cytokines including MIF, IL13, CCL2, and IL1RA in all KO iMGLs compared to WT iMGLs (Supplementary Fig. 6A, B), suggesting that the elevated LD content in KO iMGLs is not associated with a pro-inflammatory phenotype. This observation is consistent with the suppression of pro-inflammatory

cytokines by increased LXR:RXR activity that has been observed in peripheral macrophages that accumulate LDs (foam cells)[61].

## Knockout of BHLHE40/41 increases lysosomal mass and degradative capacity in iMGLs

RRHO overlapping genes between LAM macrophages from human obese adipose tissue published by ref. 6 (Dataset S6, FDR < 0.05) and all KO iMGLs transcriptional signatures are enriched for pathways related to lysosomal function (Fig. 4B, C), suggesting BHLHE40/41 may regulate lysosomal processing in human microglia. In addition, BHLHE40 and BHLHE41 have been shown to repress the master regulators of lysosomal biogenesis, TFEB and MITF, through binding to the same set of gene regulatory elements[39,62]. These findings suggest that decreasing BHLHE40/41 expression may lead to enhanced activity of MiT/TFE family TFs and therefore increased lysosomal mass and degradative capacity. To test this hypothesis, we measured LysoTracker-Red and LysoSensor-Green gMFI by flow cytometry to quantify lysosomal mass and lysosomal pH, respectively. Consistent with our hypothesis, we observed an increase in lysosomal mass in all KO iMGLs, that reached statistical significance in DKO compared to WT iMGLs ($ES_{DKO-WT}$ = 4080 [109.9, 8049], $P$ value$_{DKO-WT}$ = 0.0438, $N$ = 5/group) (Fig. 5G), as well as a statistically significant decrease in lysosomal pH (and therefore increased acidification of lysosomes) in DKO compared to WT iMGLs ($ES_{DKO-WT}$ = 24256 [3502, 45009], $P$ value$_{DKO-WT}$ = 0.0222; $N$ = 5/group) (Fig. 5H). Lower lysosomal pH and increased expression of lysosomal cathepsins (Fig. 5A) suggest enhanced degradative capacity of lysosomes in DKO iMGLs. To test this, we used a fluorochrome-conjugated Bovine Serum Albumin (DQ-BSA) assay. During endocytosis, DQ-BSA is delivered to the late endosome/lysosome and is subject to proteolysis by lysosomal enzymes leading to an increase in fluorescence. Using flow cytometry, we showed that BSA proteolysis (measured by gMFI of the conjugated fluorochrome) is significantly enhanced in DKO compared to WT iMGLs ($ES_{DKO-WT}$ = 9485 [4678, 14293], $P$ value$_{DKO-WT}$ = 0.0005, $N$ = 5/group) after 1 h incubation with DQ-BSA (Fig. 5I). Bigger differences between genotypes were observed during time-lapse recording (up to 5 h, using Incucyte S3 live-cell imaging system, Fig. 5J). We observed that DQ-BSA proteolysis is unchanged in 40KO iMGLs but it is increased in 41KO and DKO iMGLs compared to WT iMGLs ($ES_{41KO-WT}$ = 192 [47.91, 336.0], $P$ value$_{41KO-WT}$ = 0.0126; $ES_{DKO-WT}$ = 281.6 [137.5, 425.6], $P$ value$_{DKO-WT}$ = 0.0013, $N$ = 3/group) (Fig. 5J). Altogether these data suggest that lysosomal function is increased in 41KO and DKO iMGLs compared to WT iMGLs.

## Knockdown of BHLHE40/41 partially recapitulates LAM transcriptional and cellular responses in human THP-1 macrophages

To further validate the role of BHLHE40/41 as modulators of the DLAM response, we used the THP-1 monocytic leukemia immortalized cell line differentiated to macrophage-like cells by treatment with phorbol 12-myristate 13-acetate (PMA) as a model of human monocyte-derived macrophages (MACs). We transiently transfected MACs with siRNAs targeting *BHLHE40* (40KD), *BHLHE41* (41KD) or both (DKD). Using this approach, *BHLHE40* expression was reduced by ~60% in 40KD and DKD MACs, and *BHLHE41* expression was reduced by ~50% in *41*KD and DKD MACs at both transcript and protein levels (Supplementary Fig. 3C, D) when compared to control cells transfected with scrambled siRNA (SCR). Using THP-1 macrophages with transient reduction of *BHLHE40* and/or *BHLHE41* expression, we performed a similar set of experiments as we did in iMGLs with genetic inactivation of BHLHE40 and/or BHLHE41 (see previous sections) in order to validate our findings in an additional human macrophage cell culture model. First, we performed RNA-seq ($N$ = 4/group) to assess global transcriptomic changes associated with reduction of BHLHE40 and/or BHLHE41 expression in MACs. Results of differential gene expression and gene set enrichment analyses are included in Supplementary Dataset 4.

Next, we used the RRHO method to identify statistically significant overlaps between the signature of LAM macrophages from human obese adipose tissue published by ref. 6 (Dataset S6, FDR Adj.$P$ value < 0.05) and transcriptional changes associated with reduction of BHLHE40 and/or BHLHE41 expression (KD vs SCR) in MACs. A significant overlap was identified between genes down-regulated in each pair of KD MAC and LAM transcriptional signatures (Supplementary Fig. 7A) (hypergeometric overlap P value: P value$_{40KD}$ = 1.44e-5, P value$_{41KD}$ = 7.96e-6, P value$_{DKD}$ = 3.13e-7) (Supplementary Fig. 7B). A significant overlap was also identified between smaller subsets of genes up-regulated in 40KD or DKD MACs compared to SCR MACs and the LAM transcriptional signature (hypergeometric overlap $P$ value: P value$_{40KD}$ = 4.10e-3, P value$_{DKD}$ = 1.36e-6) (Supplementary Fig. 7B). Similarly to what we observed using in vitro cultures of *BHLHE40/41* double-knockout iMGLs, the most statistically significant and concordant overlaps were observed between the LAM signature and transcriptional changes that occur in DKD MACs, suggesting that expression of both BHLHE40 and BHLHE41 have to be eliminated or reduced to more robustly mimic the LAM transcriptional response observed in vivo. Pathway enrichment and activity analysis of RRHO overlapping up- and down-regulated genes using IPA (Supplementary Fig. 7C) revealed that "Efflux of cholesterol" and "Engulfment of cells" are predicted to be more active (i.e., have positive Z-scores) in KD compared to SCR MACs, and that "Lysosomal storage disease", "Accumulation of lipid", "Inflammatory response" are predicted to be less active (i.e., have negative Z-scores) in KD compared to SCR MACs (Supplementary Fig. 7C). To further support these findings, we performed targeted differential gene expression analyses using RT-qPCR. We found that the expression of genes involved in cholesterol efflux/lipoprotein metabolism and lysosomal function was elevated especially in DKD MACs compared to SCR MACs but reached statistical significance only in a few comparisons, likely due to low statistical power and the fact that the extent of upregulation is smaller compared to that observed in iMGLs with complete loss of BHLHE40/41 expression (Fig. 6A). To further investigate whether reduced expression of *BHLHE40* and/or *BHLHE41* in THP-1 macrophages could recapitulate the changes in cellular functions that we observed in iMGLs lacking *BHLHE40* and/or *BHLHE41*, we measured APOE secretion, cholesterol efflux capacity, and LD content in KD and SCR MACs. We found that APOE secretion is significantly increased in 40KD compared to SCR MACs (Fig. 6B) ($ES_{40KD-SCR}$ = 2.33 [0.04, 4.62], $P$ value$_{40KD-SCR}$ = 0.0457, $N$ = 6/group). We also found that cholesterol efflux to APOA-1 acceptor is significantly increased in all KD MACs compared to SCR MACs ($ES_{40KD-SCR}$ = 12.45 [8.47, 16.44], $P_{40KD-SCR}$ < 0.0001; $ES_{41KD-SCR}$ = 4.83 [0.84, 8.81], $P_{41KD-SCR}$ < 0.0170; $ES_{DKD-SCR}$ = 6.91 [2.92, 10.89], $P$ value$_{DKD-SCR}$ = 0.0011; $N$ = 6/group) (Fig. 6C). Furthermore, we found that all KD MACs showed increased LD content as evidenced by higher BODIPY gMFI compared to SCR MACs ($ES_{40KD-SCR}$ = 2348 [567, 4128], $P$ value$_{40KD-SCR}$ = 0.0097; $ES_{41KD-SCR}$ = 2890 [1110, 4671], $P$ value$_{41KD-SCR}$ = 0.0020; $ES_{DKD-SCR}$ = 3563 [1782, 5343], $P$ value$_{DKD-SCR}$ = 0.0003; $N$ = 6/group). We also determined that higher LDs content is not associated with a pro-inflammatory phenotype, as evidenced by reduced levels of pro-inflammatory cytokines secreted by all KD compared to SCR MACs (Supplementary Fig. 6C, D).

Together these data further demonstrate the previously unknown role of BHLHE40/41 in the modulation of DLAM response genes and cellular processes involved in lipid metabolism in another in vitro model of human macrophages and reveal a high degree of similarity between iMGLs with complete loss of BHLHE40/41 and THP-1 macrophages with reduced levels of *BHLHE40/41*.

## Knockout of Bhlhe40/41 partially recapitulates DLAM transcriptional responses in mouse microglia

To investigate whether loss of BHLHE40/41 function could lead to gene expression changes that partially recapitulate the DLAM

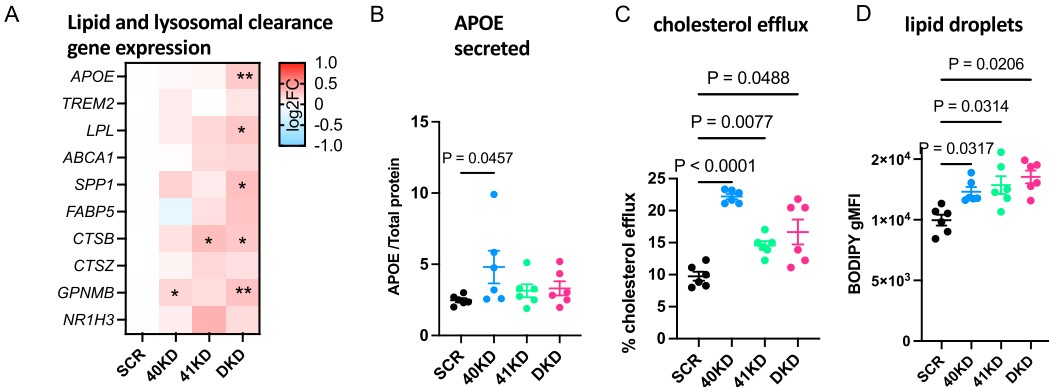

**Fig. 6 | Knockdown of *BHLHE40/41* increases expression of lipid clearance genes, cholesterol efflux and lipid droplet content in human THP1 macrophages (MACs). A** Expression of lipid and lysosomal clearance genes was measured by RT-qPCR, *N* = 7/group. log2(fold change) (log2FC) is calculated with SCR MACs as reference. Separate plots for each gene in Supplementary Fig. 10. **B** Secreted APOE normalized to total protein measured by ELISA, *N* = 6/group. **C** Percentage of cholesterol efflux, *N* = 6/group. **D** LD content (BODIPY gMFI) measured by flow cytometry in BODIPY-positive cells, *N* = 6/group. *N* represents the number of independent siRNA transfections. Flow cytometry gates were drawn based on FMO (fluorescence minus one) controls. Differences of means between groups were tested using one-way ANOVA with repeated measures followed by Dunnett's post-hoc test. *P value < 0.05, **P value < 0.01, or indicated directly on graphs. Data plotted as mean ± SEM. Detailed statistics are shown in Supplementary File 1. 40KD = MACs treated with BHLHE40 siRNA, 41KD = MACs treated with BHLHE41 siRNA, DKD = MACs treated with BHLHE40 and BHLHE41 siRNA, SCR = MACs treated with scrambled siRNA.

transcriptional response across species as well as in vivo, we acutely isolated microglia from *Bhlhe40/Bhlhe41* double knockout (DKO) mice[59] and their WT littermates and performed bulk RNA-seq transcriptome profiling. While previous data suggests that only *Bhlhe40* has been implicated in regulation of APOE-related DLAM responses[13], our GRN predictions along with data from human iMGLs and MACs, suggest that Bhlhe40 and Bhlhe41 may have an additive effect on DLAM response. Consequently, we decided to use DKO instead of single KO mice. Results of differential gene expression and gene set enrichment analyses are included in Supplementary Dataset 5. Similarly to what we observed using in vitro cultures of *BHLHE40/41* double-knockout human iPSC-derived microglia, transcriptional changes in *Bhlhe40/41* double-knockout mouse microglia are significantly and positively correlated ($\rho_{DKO}$ = 0.31, P value$_{DKO}$ < 0.0001) with the DAM mouse microglia signature published by ref. 8 (Table S3 FDR Adj.*P* value < 0.05). Next, we used the RRHO method to identify statistically significant overlaps between the DAM mouse microglia signature ([8] Table S3, FDR Adj.*P* value < 0.05) and transcriptional changes associated with inactivation of both *Bhlhe40* and *Bhlhe41* (DKO vs WT) in mouse microglia. This analytical approach revealed statistically significant and concordant overlaps in the bottom-left quadrant (i.e., between genes up-regulated in both transcriptional signatures) (hypergeometric overlap P value: P value$_{DKO}$ = 9.30e-18) and in the top-right quadrant (i.e., between genes downregulated in both transcriptional signatures) (hypergeometric overlap P value: P value$_{DKO}$ = 8.50e-24) (Fig. 7A). Pathway enrichment and activity analysis of RRHO overlapping up- and down-regulated genes using IPA (Fig. 7B) revealed that "Endocytosis", "Phagocytosis", and "Fatty acid metabolism" are predicted to be more active (i.e., have positive Z-scores) in DKO compared to WT mouse microglia, and that "Lysosomal storage disease", "Gliosis", and "Amyloidosis" are predicted to be less active (i.e., have negative Z-scores) in DKO compared to WT mouse microglia (Fig. 7C).

In addition to RRHO analysis, we performed GSEA analysis to investigate whether transcriptional changes that occur in DKO mouse microglia are positively or negatively enriched for genes up- and down-regulated in mouse DLAMs from several studies: DAM microglia from the brain of 5XFAD mice by ref. 8, CD11c-positive microglia from the brain of APPswe/PS1dE9 mice by ref. 63, LAM macrophages from epididymal visceral white adipose tissue by ref. 6, TREM2$^{hi}$ macrophages from atherosclerotic

plaques of Ldlr-/- mice on high-fat diet by ref. 9, activated-response mouse microglia (ARM) by ref. 28, and genes in mouse microglia neurodegeneration- and demyelination-related co-expression modules by ref. 19. Collectively, we found that transcriptional changes that occur in DKO mouse microglia are positively and negatively enriched for, respectively, genes up- and down-regulated in the mouse DLAMs mentioned above (Fig. 7D). Interestingly, using RRHO method we found no significant correlation between LDAM microglia (BODIPY positive microglia from aged mouse hippocampus) and mouse DKO transcriptome, suggesting LDAM proinflammatory microglia are a distinct population from DLAM caused by loss of Bhlhe40/41 (Supplementary Fig. 9B). Taken together, these results indicate that transcriptional changes associated with loss of *Bhlhe40* and *Bhlhe41* in mouse microglia can partially recapitulate the DAM transcriptional response in vivo.

## Discussion

Subpopulations of macrophages that reside in cholesterol/lipid-rich tissues display similar TREM2-dependent transcriptional responses to host tissue damage (e.g., disease-associated microglia in the aging or degenerating brain, the most cholesterol/lipid-rich organ of the human body, TREM2$^{hi}$ macrophages in atherosclerotic plaques, and lipid-associated macrophages in fatty liver and obese adipose tissue)[9,10,14,20–29]. We collectively refer to these subpopulations of microglia and peripheral macrophages as DLAMs. In this study we report the identification of transcriptional regulators of the DLAM response (DLAM TFs) through gene regulatory network analyses of microglia and peripheral macrophage transcriptomes from human and mouse single-nucleus and single-cell RNA-seq datasets. In particular, we leveraged data not only from microglia in healthy and diseased brains, but also from macrophages in atherosclerotic plaques and adipose tissue from healthy and obese animals as well Kupffer cells in healthy and diseased livers[9,10,14,20–29]. Taking advantage of the breadth of challenge conditions in mouse models and human diseases as well as the high resolution of single-cell and single-nucleus RNA-seq data, we were able to nominate 11 strong candidate DLAM TFs (*BHLHE41, HIF1A, ID2, JUNB, MAF, MAFB, MEF2A, MEF2C, NACA, POU2F2*, and *SPI1*) that were highly replicated within and across species and conditions.

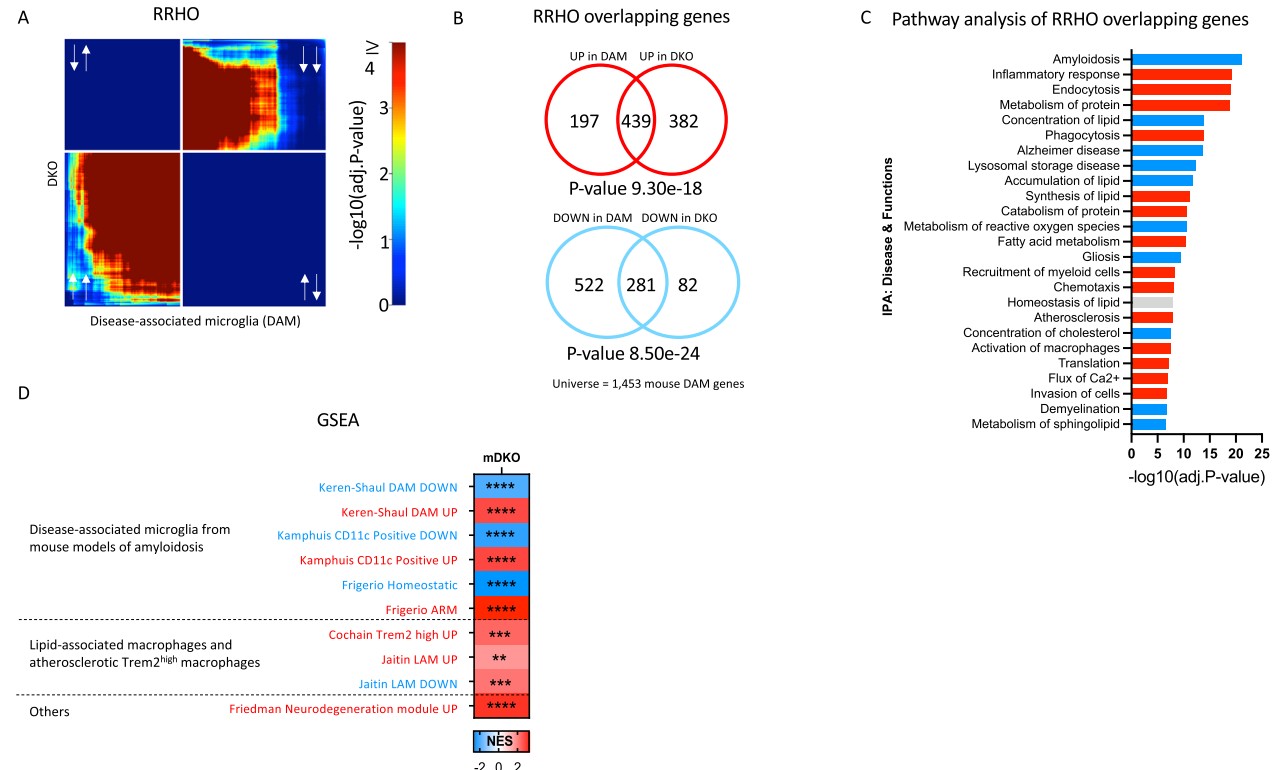

**Fig. 7 | Knockout of *Bhlhe40/41* partially recapitulates DLAM transcriptional responses in mouse microglia. A** Rank-rank hypergeometric overlap (RRHO) heatmaps visualizing significant overlaps in gene expression changes between mouse *Bhlhe40* and *Bhlhe41* double knockout (DKO) mouse microglia and DAM mouse microglia (ref. 8, Table S3, FDR Adj.*P* value < 0.05) transcriptional signatures. Adj.*P* value in color temperature scale represents FDR corrected *P* value of hypergeometric overlap test (equivalent to one-sided Fisher's exact test)[53]. White arrows indicate whether genes are up- or downregulated. **B** Venn diagram of the most significant overlap between genes up-regulated in both *Bhlhe40/41* DKO mouse microglia and the DAM mouse microglia transcriptional signature, corresponding to the warmest pixel in the bottom-left quadrant of the heatmap (top) and Venn diagram of the most significant overlap between genes down-regulated in both *Bhlhe40/41* DKO mouse microglia and the DAM mouse microglia transcriptional signature, corresponding to the warmest pixel in the upper-right quadrant of

the heatmap (bottom). *P* values are calculated using the hypergeometric overlap test (one-sided Fisher's Exact test) restricted to the universe of 1453 mouse DAM genes (ref. 8, Table S3, FDR Adj.*P* value < 0.05). **C** Pathways ("diseases and biological functions" category) found by IPA to be significantly enriched for RRHO overlapping genes. Red bars represent pathways predicted by IPA to be more active (i.e., with positive Z-scores), blue bars represent pathways predicted by IPA to be less active (i.e., with negative Z-scores), gray bars represent pathways with non-attributed Z-scores. Adj.*P* value on the x-axis represents the Benjamini-Hochberg (B-H) multiple testing-corrected *P* value calculated using right-tailed Fisher's Exact Test. DKO = *Bhlhe40/41* DKO mouse microglia, compared to microglia derived from wild-type control mice. **D** Gene sets enrichment analysis (GSEA) of DLAM genesets in DKO transcriptome. DLAM genesets are listed in Supplementary Dataset 1 NES - normalized enrichment score, ** FDR *q* value < 0.01, *** FDR *q* value < 0.001, **** FDR *q* value < 0.0001.

## The DLAM response is likely regulated by the cooperation of multiple TFs

Interestingly, several of these DLAM TFs have been shown to interact with each other in various ways, suggesting that they might regulate the DLAM response cooperatively. For example, PU.1 binds an enhancer element located downstream of the *Mef2c* hematopoietic-specific promoter and is required for regulating *Mef2c* expression during myeloid cell development[64]. Similarly, Bhlhe40 acts together with PU.1 to drive the selection of enhancers in large peritoneal macrophages in the presence of retinoic acid[65]. These data suggest that the candidate DLAM TFs nominated in this study likely interact with each other in complex ways to cooperatively regulate the transcriptional and cellular responses of macrophages/microglia to cholesterol/lipid-rich tissues damaged by aging and disease.

Of note, several DLAM response genes in both human and mouse GRNs are predicted to be strongly influenced by only a small number of candidate DLAM TFs. There are multiple reasons why this could be the case. Genuine biological specificity might be one of the reasons for this observation. Additionally, if the genes are highly specific to the DLAM response (e.g., *LPL*), the presence of only a small fraction of cells that significantly upregulate this marker might make it difficult to infer a robust association with a TF. Furthermore, genes that are lowly

expressed but significantly induced in DLAMs may not pass the expression threshold in our analyses for a proportion of the GRNs and thus do not display more associations with a TF. Finally, robust changes in mRNA levels are required to infer associations between a gene and a TF with our approach; hence, for those genes whose mRNA values are not robustly varied, our approach would be unable to detect TF-gene associations.

## The role of DLAM TFs and their upstream regulators in AD risk modification

We next sought to identify which of the nominated DLAM TFs are more likely to directly regulate the DLAM response, e.g., through binding to promoters of DLAM genes. To achieve this, we used open chromatin profiles of macrophages/microglia and sequence motifs to which each TF is predicted to bind in order to identify TF proxy-binding sites, i.e., open chromatin regions (ATAC-seq peaks) that contain binding motifs for a specific TF. The highest proportion of DLAM gene promoters contained a *BHLHE40/41* proxy-binding site in human and mouse macrophages/microglia, highlighting these two highly homologous and functionally redundant TFs as the most likely direct regulators of DLAM genes.

The interplay between AD risk genes and regulation of the DLAM response has been previously highlighted[66], suggesting that AD risk genes may modulate disease susceptibility by regulating the gene expression profiles and in turn the cellular activities of macrophages in response to dyshomeostasis of cholesterol/lipid-rich tissues[11,12]. Consistent with this hypothesis, previous studies have reported that several well-established AD risk genes (APOE, TREM2, and PLCG2) are critically important for the development and function of DAM microglia in response to cholesterol/lipid overload due to host tissue damage during aging and disease[8,13,67]. Furthermore, a recent study showed that APOE risk-increasing (APOE ε4/ε4, similar to APOE, TREM2, and PLCG2 loss-of-function mutations) and risk-decreasing (APOE ε2/ε2) genotypes are associated with decreased and increased DAM transcriptional response (as well as decreased and increased efferocytosis) by microglia, respectively, compared to the risk-neutral APOE ε3/ε3 genotype[68]. Consistently, recent studies demonstrated that APOE4 expression in mouse microglia leads to decreased DAM response, lipid droplet accumulation, decreased lysosomal function, and exacerbated amyloid pathology[69]. Mechanistically, APOE4 expression in mouse microglia impairs the DAM response by inducing TGFβ-mediated checkpoints[70]. One of the proposed checkpoint is another candidate DLAM TF nominated in this study, SPI1/PU.1, which is also a candidate AD risk gene and master regulator of myeloid cell development and function[33]. Genetic or pharmacological inhibition of SPI1/PU.1 restored expression of DAM markers and mitigated plaque pathology in APP/PS1 and P301S mouse models of AD[70]. Additionally, other TFs that we nominated in this study as candidate DLAM TFs (MAF, MEF2C, and SPI1) were also nominated as candidate AD risk genes in genome-wide association studies (GWAS)[33,71,72]. We and others have also previously shown that AD heritability is enriched within PU.1 (proxy- as well as ChIP-seq) binding sites in macrophages/microglia and that AD risk alleles in the SPI1 AD GWAS locus affect PU.1 expression, further implicating this DLAM TF in the modulation of AD risk[4,33,35]. Here we show that AD heritability is similarly enriched within proxy-binding sites of BHLHE41 (and its close homolog BHLHE40) in macrophages/microglia, implicating the BHLHE40/41 target gene network not only in DLAM response regulation but also in AD risk modification. Interestingly, BHLHE40 is located near an AD locus recently identified in an African-American GWAS[37] and therefore should be considered as a candidate AD risk gene for future studies in macrophages/microglia derived from this human population.

## Putative mechanism of BHLHE40/41-mediated regulation of the DLAM response

Bhlhe40 is upregulated in mouse DLAMs and has been proposed to regulate the expression of genes in neurodegeneration- and demyelination-related co-expression modules in mouse microglia[13,19,48,73]. However, the role of BHLHE40 (and its close homolog BHLHE41) in the regulation of DLAM response genes and pathways has not been experimentally validated or further investigated. In the current study, we used human iPSC-derived microglia and THP-1 macrophages to show that complete loss or reduced levels of BHLHE40 and/or BHLHE41 in these cells is associated with enhanced expression of DLAM response genes and increased activity of DLAM response pathways (i.e., cholesterol clearance and lysosomal processing) suggesting that these TFs, when induced during the DLAM response, may act as transcriptional repressors of these functional components of the DLAM response. Inhibition of the DLAM transcriptional response by BHLHE40/41 may be part of a negative feedback loop initiated by other TFs such as LXR:RXR (and other RXR-containing) nuclear receptors and MiT/TFE family members[38,39,74]. Indeed, studies have shown that LXR:RXR activation induces expression of Bhlhe40[75] which in turn (together with Bhlhe41) represses expression of genes induced by LXR:RXR activation[38]. Similar repression by Bhlhe40 have been shown for other TFs such as PPARγ:RXRα whose target genes are also involved

in lipid metabolism (i.e., lipid storage, lipolysis) and are markedly increased in Bhlhe40 deficient mice in white adipose tissue and liver[40,76]. Similarly, BHLHE40 and BHLHE41 gene expression is induced by nuclear TFEB and act in opposition to TFEB, competing for the same binding sites[39]. Another example is MITF, which shares similar functions with TFEB and promotes the expression of genes involved in lysosomal biogenesis. In melanoma, BHLHE40 and BHLHE41 transcription is induced by direct binding of MITF to their promoters[51]. In turn, MITF transcription is repressed by direct binding of BHLHE40 to the MITF promoter[62]. Consistent with these observations, the results of our experiments using human iPSC-derived microglia and THP-1 macrophages indicate that interrupting/weakening this negative feedback loop by eliminating/reducing BHLHE40 and/or BHLHE41 gene expression in these cells is associated with increased expression of select LXR:RXR and MiT/TFE target genes and with increased activity of pathways regulated by these TFs, such as cholesterol clearance and lysosomal processing (Fig. 8). This model implies that LXR:RXR nuclear receptors and MiT/TFE family members may themselves be DLAM TFs. Indeed, MITF has been nominated and functionally validated as a candidate driver of a DAM-like transcriptional state observed in cultured iMGLs exposed to cholesterol/lipid-rich cellular debris in vitro[32]. In addition, LXRα/β has been nominated as candidate drivers of an anti-inflammatory DAM transcriptional state observed in microglia acutely isolated from the brain of 5XFAD mice[17].

## Lipid droplet phenotype associated with BHLHE40/41 loss-of-function in macrophages/microglia

In this study, we show that complete or partial loss of BHLHE40/41 function in human iPSC-derived microglia (iMGLs) and THP-1 macrophages (MACs) cultured in vitro is associated with increased levels of neutral lipids/lipid droplets (LDs), a phenotype that we have also shown to be dependent on LXR:RXR activation. Increased LDs content has also been observed in vivo in alveolar macrophages of Bhlhe40/41 double knockout mice[59]. These findings are in accordance with recent studies reporting a similar LDs accumulation phenotype in DLAM populations of human and mouse macrophages/microglia[6,56]. For example, a subset of macrophages found in adipose tissue of obese individuals and characterized by high levels of DLAM markers such as CD9 and LPL also show increased LDs accumulation[6]. In the context of AD, a recent study showed that plaque-associated xenotransplanted microglia (xMGLs, human iPSC-derived microglia injected into the brain of immunocompetent mice[57]) accumulate PLIN2 (perilipin 2)-positive LDs[56]. In contrast, other studies identified a subset of microglia that accumulate LDs in the hippocampus of aged mice (referred to by the authors as LDAM, for LD-associated microglia) and are characterized by increased expression of pro-inflammatory genes and secretion of pro-inflammatory mediators[60]. In BHLHE40/41 KO/KD iMGLs and MACs, increased LDs content is, on the contrary, associated with lower levels of pro-inflammatory mediators secreted in the culture media. These data suggest that increased accumulation of LDs due to lack of Bhlhe40/41 is not associated with a pro-inflammatory phenotype and may instead be part of a protective adaptive response in which sequestration of free cholesterol and other potentially toxic lipids prevents cellular stress and inflammation[77-80]. Interestingly, pharmacological activation of LXR:RXR nuclear receptors leads to an increase in LDs in human retinal pigmented epithelial cells and promotes both cholesterol efflux and LDs formation[81] similar to what we have observed in BHLHE40/41 KO/KD iMGLs and MACs. This suggests that LXR:RXR activation downstream of BHLHE40/41 loss-of-function in macrophages/microglia promotes cholesterol clearance by stimulating both its efflux to extracellular acceptors like APOE and its esterification for non-toxic accumulation and intracellular storage in LDs[81]. All together, these findings support our hypothesis that the higher LD content observed in macrophages/microglia in the absence of BHLHE40/41 may be attributed to increased LXR:RXR activity in

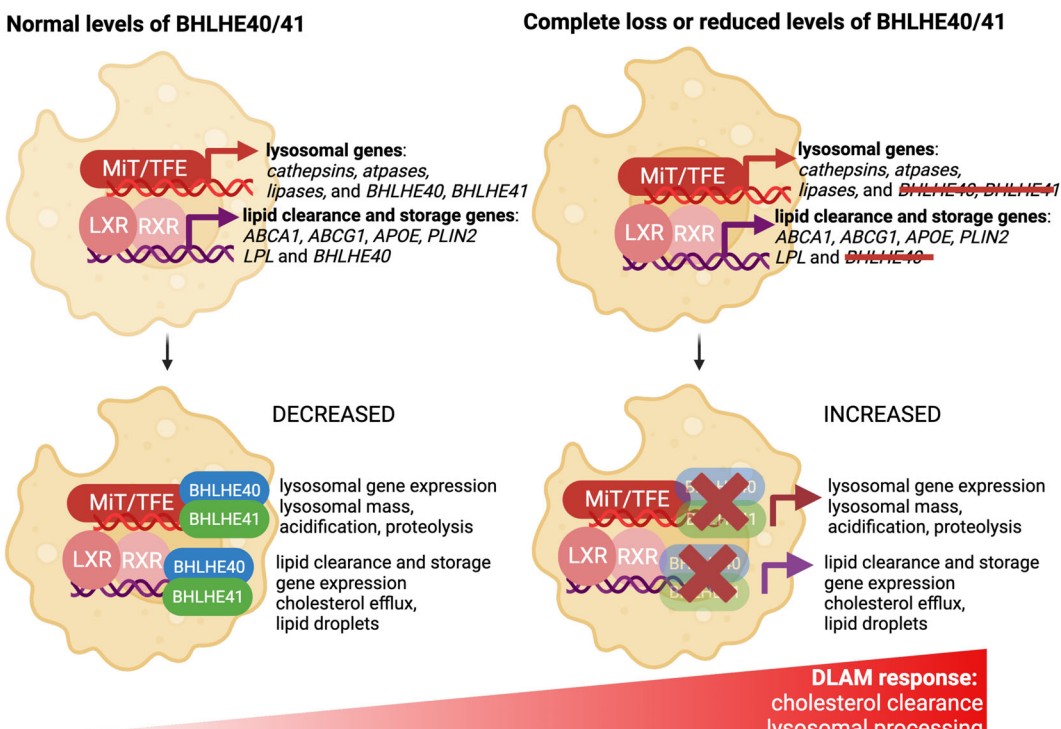

**Fig. 8 | Proposed mechanism for DLAM response regulation by BHLHE40/41.**
Left) In wild-type mouse microglia or human iPSC-derived microglia (iMGLs) and THP-1 macrophages (MACs) treated with scrambled siRNA (and thus with normal levels of BHLHE40/41), when LXR:RXR and MiT/TFE family TFs stimulate the expression of cholesterol clearance and lysosomal processing genes that contribute to the LAM transcriptional program, they also stimulate the expression of *BHLHE40* and *BHLHE41*. In turn, BHLHE40/41 oppose the activity of LXR:RXR and MiT/TFE family TFs by binding to the promoters of their target genes and leading to decreased expression of cholesterol clearance and lysosomal processing genes, in a negative feedback loop. Right) In mouse microglia or human iMGLs and MACs with complete loss or reduced levels of BHLHE40/41, this negative feedback loop is interrupted or weakened resulting in increased expression of cholesterol clearance and lysosomal processing genes to partially recapitulate the DLAM transcriptional and cellular response, including increased cholesterol efflux to APOE and storage in LDs, as well as increased lysosomal mass, acidification and proteolysis. The figure was created with BioRender.

these cells and may be a part of a protective mechanism preventing cellular stress and inflammation induced by potentially toxic levels of lipids like free cholesterol.

## Study limitations

Although we were able to pinpoint and functionally validate transcriptional regulators of the DLAM response, there are several limitations to our study. The DAM gene expression signature has been robustly identified in microglia from several mouse models of aging and disease, however its identification in human microglia has been more challenging[82]. One of the challenges is the prevalent use of snRNA-seq to profile human microglial transcriptional states[83]. On the other hand, a recent publication using scRNA-seq identified a transcriptional state in human iPSC-derived microglia cultured in vitro and treated with myelin fragments, apoptotic cells and synaptosomes that resembles the DAM signature observed in mouse microglia in vivo, suggesting an evolutionarily conserved pattern of microglial gene expression changes in response to cholesterol/lipid-rich cellular waste[32]. In our current study, we assumed conservation of the DAM signature from mouse to human microglia; however, as more single-nucleus and single-cell transcriptomic and multi-omic datasets from the aging and diseased human brain are generated, our understanding of the DAM signature in human microglia may evolve. Additionally, our analysis of microglial gene expression from double knockout of *Bhlhe40/41* shows that the DAM transcriptional program is partially recapitulated, including pathways involved in "Lipid transport", "Oxidative phosphorylation" and "Phagosome maturation". These analyses were conducted on relatively young (2 and 6-month-old) and healthy mice; however, in future studies we will examine the DLAM response in older mice or in the context of a disease-relevant challenge (e.g., mouse models of demyelination or amyloid deposition). Another limitation of the study is the mouse model with a global knock-out of Bhlhe40 and Bhlhe41 (mouse DKO) which may affect other cell types such as astrocytes where Bhlhe40 and Bhlhe41 are also expressed. We cannot exclude that there is a non-cell-autonomous effect mediated possibly by astrocytes lacking Bhlhe40/41 that would affect DKO microglia. Finally, we have utilized open chromatin data to generate sets of proxy-binding sites by stratifying ATAC-seq peaks based on the presence of known TF binding motifs. Since these motifs can be quite similar for related TFs, our proxy-binding sites likely reflect putative binding of multiple TFs within the same family. As more TF-specific ChIP-seq datasets from human and mouse microglia are generated, we will be able to further validate and refine our findings based on proxy-binding sites.

In summary, we have utilized single-cell and single-nucleus RNA-seq data from various human and mouse macrophage/microglia subpopulations (DLAMs) that were shown to exhibit similar transcriptional responses to cholesterol/lipid overload. We implicated multiple candidate transcriptional regulators of the DLAM response, with *BHLHE40/41* being the most likely direct regulators. We further implicated *BHLHE40/41* in AD risk modification, highlighting an intriguing interplay between transcriptional regulation of the DLAM response and modulation of disease susceptibility. Finally, we validated these bioinformatic findings experimentally in human and mouse macrophages/microglia, nominating BHLHE40/41 as candidate transcriptional regulators of the DLAM response and putative drug targets for therapeutic modulation of macrophage/microglial function in AD and other disorders of lipid-rich tissues.

## Methods

### Reconstruction of gene regulatory networks

Single-cell and single-nucleus RNA-seq datasets from microglia, Kupffer cells, atherosclerotic plaque macrophages, and adipose tissue macrophages were obtained from Gene Expression Omnibus (GEO)[9,10,14,20–29] (Supplementary Dataset 1). Cell type annotations reported by the authors were used to extract microglia, Kupffer cells, and other macrophages. Count matrices were obtained from each study and CPM normalized. Pearson correlation matrices were used as distance matrices for metacell reconstruction. The pipeline for reconstruction of metacells was adopted from the PISCES tool[84,85] and example code can be found at our github page https://github.com/marcoralab/bhlhe_manuscript. Briefly, this approach constructs a kNN graph using the data, partitions the data into an appropriate number of metacells, taking into account the desired number of neighbors. It then aggregates the counts from closest neighbors into MetaCells. The MetaCells function was used in the following manner: *MetaCells(data, dist.mat, numNeighbors = numNeighbors, subSize = subSize). numNeighbors* and *subSize* for each dataset are provided on our github page. Resulting metacells were filtered to include genes that showed non-zero counts in at least 75% of metacells. ARACNe was then used to reconstruct gene regulatory networks using default parameters[30]. The threshold for MI was calculated as follows: *java -Xmx5G -jar Aracne.jar -e Matrix_file.txt -o. --tfs list_of_TFs.txt --pvalue 1E-8 --seed 1 --calculateThreshold*. The network was built using 200 bootstraps using the following command for each: java -Xmx5G -jar Aracne.jar -e *Matrix_file.txt* -o. --tfs *list_of_TFs.txt* --pvalue 1E-8 --seed *seed_number*. Bootstraps were further consolidated into a single network in the following manner: *java -Xmx5G -jar Aracne.jar -o. --consolidate*. Only significant interactions (FDR Adj.*P* value < 0.05) were used for further downstream analyses. Meta-analyzed networks (displayed in Fig. 1A, B and used for final TF regulon extraction) were generated by aggregating all the bootstraps from individual networks generated by ARACNe using the *consolidate* function mentioned above[30,86].

### Enrichment analyses in TF regulons

Each TF regulon was tested for significant overlap with three DLAM genesets (human or mouse DAM, LAM, and TREM2[hi] genesets, Supplementary Dataset 1) by using the hypergeometric test (*phyper* function in R *stats* package). The TFs whose regulons were significantly enriched (FDR Adj.*P* value < 0.2) were nominated as DLAM TFs. The DLAM genesets for the enrichment analyses were obtained from their respective studies in the following manner: mouse DAM geneset was obtained from Table S2 of ref. 8, mouse LAM geneset was obtained from Dataset S5 of ref. 6, and mouse TREM2[hi] macrophage geneset was obtained from Online Table I (Cluster 11) of ref. 9. The human LAM geneset was obtained from Dataset S6 of ref. 6 and filtered for FDR Adj.*P* value < 0.05. The human DAM and TREM2[hi] genesets were constructed by obtaining the human orthologs of the mouse genes in the corresponding mouse geneset using biomaRt[87]. High-confidence TFs were selected if they were 1) enriched for all three DLAM genesets in at least half of human and mouse networks and 2) conserved between species.

### Processing of ATAC-seq data and peak calling

Macrophage ATAC-seq data were obtained through the Gene Expression Omnibus (GEO, GSE100383)[88]. Human microglial ATAC-seq data were obtained from the database of Genotypes and Phenotypes (dbGaP, phs001373.v2.p2)[31]. Mouse macrophage and microglia ATAC-seq data were obtained through GEO (GSE151015, GSE89960)[31,89]. To generate epigenomic annotations, FASTQ files were obtained from the Sequence Read Archive (SRA)[90]. Technical replicates were merged and Bowtie2 was used for alignment for both single and paired-end files[91]. FASTQC was used for quality control of the files[92]. Resulting SAM files

were filtered by MAPQ score and duplicates were removed using samtools[93]. MACS2 was used to call peaks from ATAC-seq data[94].

### Generation of promoter annotations and overlap with TF proxy-binding sites

We obtained TSS positions of genes using biomaRt[87] and extended them by 2 kb on each side of the TSS to generate gene promoter annotations. To assess if a TF is likely to be bound at a gene promoter, we created *GRanges* objects for both promoter and open chromatin regions (ATAC-seq peaks) and utilized the *subsetByOverlaps* function from the *GenomicRanges* package[95] to identify open chromatin regions that contain binding motifs for the TF of interest (TF proxy-binding sites) and overlap with the gene promoter.

### De novo TF binding motif discovery and TF binding motif frequency analysis

We used HOMER to quantify the frequencies of TF binding motif occurrence in open chromatin regions that overlap gene promoter annotations[36]. The following command was used to quantify motif frequencies depicted in Fig. 3: *annotatePeaks.pl Peaks.bed hg19 -m TF.motif -size 2000 -hist 20*. To identify open chromatin regions that contain binding motifs for the TF of interest, we used the following commands: *findMotifsGenome.pl Peaks.bed hg19. -find motif.motif -size given* and *annotatePeaks.pl Peaks.bed hg19 -m motif.motif -size given*.

### Partitioned SNP heritability analysis

We used LD Score regression[96] to estimate AD SNP heritability partitioned by proxy-binding sites of candidate DLAM TFs. We used GWAS summary statistics (excluding the APOE [chr19:45000000–45800000] and MHC/HLA [chr6:28477797–33448354] regions) from the Lambert *et al.* AD GWAS study[71].

### Generation of human BHLHE40 and/or BHLHE41 knockout iPSC lines

BHLHE40 and BHLHE41 single homozygous knock-out iPSC lines were generated by CRISPR/Cas9-mediated homologous direct repair (HDR). For BHLHE40 the HDR was designed to introduce p.R67>X, a premature stop codon in the bHLH domain. For BHLHE41 the HDR was designed to introduce p.T41>X, a premature stop codon in the bHLH domain. Briefly, one million human iPSCs from the WTC-11 donor line (UCSFi001-A (RRID:CVCL_Y803), Coriell Institute for Medical Research, GM25256) at passage 37 were electroporated using the 4D-Nucleofector System (Lonza, AAF-1003X, V4XP-3024) with a pre-assembled complex of 10 μg of S.p. Cas9 Nuclease (IDT, 1081058), and 8 μg of synthetic sgRNA (Synthego, CRISPRevolution sgRNA EZ Kit,) and 100 pmol of single-stranded oligodeoxynucleotides (ssODNs) (Life Technologies). Cells were replated on Geltrex (Thermo Fisher, A1413301)-coated plates and incubated in mTeSR Plus (Stemcell Technologies, 100-0276). After five days, single cells were dissociated with Accutase (Innovative Cell Technologies, AT104) and sorted into 96-well plates using a WOLF benchtop microfluidic cell sorter (Nanocellect) in mTeSR Plus and CloneR (Stemcell Technologies, 05888). After thirteen days, the genomic DNA was extracted from single clones using QuickExtract solution (Lucigen, QE090500) and HDR was verified by locus-specific PCR using DreamTaq DNA Polymerase (Thermo Fisher, EP0702) followed by Sanger sequencing. To generate the BHLHE40 and BHLHE41 double homozygous knockout iPSC line, the cells were electroporated with CRISPR/Cas9 reagents for BHLHE41, and after five days were electroporated again with CRISPR/Cas9 reagents for BHLHE40. One clone per genotype was randomly selected for transcriptional and functional studies. Sequences of guide RNAs, single-stranded oligodeoxynucleotides (ssODNs), and PCR primers used for CRISPR/Cas9-mediated HDR can be found in Supplementary Dataset 6.

## Genomic integrity of human iPSC lines

The WTC-11 donor human iPSC line (Coriell Institute for Medical Research, GM25256) was tested for genomic integrity at passage 36 using SNP-array technology (Global Diversity Array v1.0 BeadChip, Illumina). No detection of CNV larger than 1.5 Mb or AOH larger than 3 Mb were detected on somatic chromosomes. The typical WTC-11 deletion of 2.9 Mb on Yp11.2 was detected. This deletion is known to be present in the donor, from whom the cell line was derived[97].

## Knockdown of BHLHE40 and/or BHLHE41 by siRNA treatment in human THP-1 macrophages

Human BHLHE40 siRNA, pool of 4 (L-010318-00), human BHLHE41 siRNA, pool of 4 (L-010043-00), and non-targeting pool (D-001810-10) were purchased from Horizon Discovery Biosciences.

THP-1 monocytes were seeded in six-well plates at $10^6$ cells per well in RPMI medium supplemented with 10% FBS, 1× Penicillin Streptomycin, and 10 mM HEPES and 25 ng/ml of phorbol-myristate-acetate (PMA) to differentiate monocytes to macrophages (MACs). After 3 days, PMA was removed and replaced with serum-free media (10% FBS was replaced with 1% BSA). Differentiated macrophages were transfected using Lipofectamine RNAiMAX (Thermo Fisher Scientific, LMRNA015) with 20 μM BHLHE40 siRNA (L-010318), 20 μM BHLHE41 siRNA (L-010043), 10 μM BHLHE40 plus 10 μM BHLHE41, and 20 μM of non-targeting control pool (D-001810-10) to generate single (40KD and 41KD) and double (DKD) knock down and scrambled (SCR) groups, respectively. Changes in BHLHE40 and BHLHE41 expression were confirmed at the mRNA level using RT-qPCR and at the protein level using western blotting. After PMA removal, THP-1 macrophages rested up to 72 h (including transfection) in serum-free media (1% BSA supplementation) prior to collection. There was no additional supplementation of media with lipid that would affect the final results.

## Generation of human iPSC-derived microglia (iMGLs)

Human induced pluripotent stem cells (iPSCs) were maintained on Matrigel (BD Biosciences) in complete mTeSR Plus (STEMCELL Technologies, 100-0276). iPSCs were passaged every 5–6 days using ReLeSR dissociation reagent (STEMCELL Technologies, 05872) and used for hematopoietic stem cell differentiation with STEMdiff Hematopoietic kit (STEMCELL Technologies, 05310) followed by differentiation to induced microglial-like cells (iMGLs) using a previously published protocol[52]. An example of mature iMGLs culture and the expression level of known transcription factors and microglial markers is shown in Supplementary Fig. 8. iPSC lines were confirmed to have a normal karyotype (KaryoStat assay, Thermo Fisher Scientific). iMGLs were maintained and fed with a microglial medium supplemented with three factors (100 ng/ml IL34, 50 ng/ml TGFβ, 25 ng/ml MCSF) for 25 days. On day 25, iMGLs were additionally supplemented with two factors (CX3CL1 and CD200, 100 ng/ml each) for an additional three days. Mature iMGLs (day 28) were used for bulk RNA-seq analyses and functional assays.

## Lipid droplet assay

Lipid droplet (LD) quantification was performed using FACS. Cells were collected and stained with 3.7 μM BODIPY for 30 min at room temperature (RT), protected from light. Single-cell data were acquired using Attune flow cytometer (Thermo Fisher Scientific) and analyzed using FCS Express 7 (De Novo Software). Gates were set up based on fluorescence minus one (FMO) controls. Gating strategy is provided in Supplementary Fig. 11. LXR agonist TO901317 was purchased from (Sigma-Aldrich, T2320), dissolved in DMSO to 0.01 M and used at 10 μM final concentration. LXR antagonist GSK2033 was purchased from (Sigma-Aldrich, SML1617), dissolved in DMSO to 5 mM and used at 2 μM final concentration.

## Cholesterol efflux assay

Cholesterol efflux was performed using Cholesterol Efflux Fluorometric Assay kit (Biovision, K582-100) following manufacturer's instructions. For this assay, cells were seeded in a 96-well plate at 33,000 cells/well (MACs) or 40,000 cells/well (iMGLs). Briefly, 24 h after transfection (MACs) or at day 28 (iMGLs) cells were labeled with Labeling Reagent for 1 h at 37 °C followed by loading cells with Equilibration Buffer. After overnight incubation, media containing Equilibration Buffer was aspirated and replaced with media containing cholesterol acceptor APOA-1 (10 μg/ml) for 4 h at 37 °C (MACs) or human HDL (25 μg/ml) for 4 h at 37 °C (iMGLs). At the end of incubation, supernatants were transferred to flat bottom clear 96-well white polystyrene microplates (Greiner Bio-one, 655095). Adherent cell monolayers were lysed with Cell Lysis Buffer and incubated for 30 min at RT with gentle agitation followed by pipetting up and down to disintegrate cells. Cell lysates were transferred into flat bottom clear 96-well white polystyrene microplates. Fluorescent intensity (Ex/Em = 485/523 nm) of supernatants and cell lysates was measured using Varioskan LUX multimode microplate reader (Thermo Fisher Scientific, VL0000D0). Percentage of cholesterol efflux was quantified as follows: % cholesterol efflux = Fluorescence intensity of supernatant/fluorescence intensity of supernatant plus fluorescence intensity of cell lysate × 100. We performed a total of 5–6 independent experiments with 2-3 technical replicates per experiment that were averaged within each experiment.

## Lysosomal assays

At day 28 iMGLs cultures were stained with 100 nM LysoTracker-Red (Thermo Fisher Scientific, L7525) for 30 min at 37 °C followed by 1 μM LysoSensor-Green (Thermo Fisher Scientific, L7535) staining for 1 min at 37 °C. To characterize the hydrolytic capacity of lysosomes, cells were stained with 1 μg/ml DQ Red BSA (Thermo Fisher Scientific, D12051) for 1 h at 37 °C. Cells were also stained with LIVE/DEAD Fixable Violet Dead Cell Stain (Life Technologies, L34973) to exclude dead cells. After collecting the cells, single-cell data were acquired using Attune flow cytometer (Thermo Fisher Scientific) and analyzed using FCS Express 7 (De Novo Software). Gates were set up based on fluorescence minus one (FMO) controls. Gating strategy is presented in Supplementary Fig. 11. In addition, we quantified DQ-BSA red fluorescent signal over time using the Incucyte S3 live imaging system. Cells were plated in 96-well plates (40,000 cells/well) and treated with 1 μg/ml of DQ Red BSA. Images were acquired every 30 min over 5 h at 37 °C. Total integrated density was calculated as mean red fluorescent intensity multiplied by surface area of masked object (i.e., cell), [RCU × μm²].

## Quantification of secreted APOE by ELISA

Secreted APOE was measured using human APOE ELISA kit (Thermo Fisher Scientific, EHAPOE) following manufacturer's instructions. Briefly, non-diluted culture supernatants (100 μl) were added to wells coated with human APOE antibody and incubated overnight at 4 °C with gentle agitation followed by incubation with solution containing biotin-conjugate (1 h at RT) and then solution containing streptavidin-HRP (1 h at RT). Reaction was developed using TMB substrate incubated for 30 min at RT followed by adding stop solution. Absorbance was read at 450 nm using Varioskan LUX multimode microplate reader (Thermo Fisher Scientific, VL0000D0). Sample concentrations were quantified based on the standard curve and normalized to total protein obtained through BCA protein assay kit (Thermo Fisher Scientific, 23225) We performed 5–6 independent experiments with 2–3 technical replicates per experiment that were averaged within each experiment.

## Western blotting

Cells were lysed in NE-PER kit to extract nuclear and cytoplasmic fractions separately (Thermo Fisher Scientific, 78833) supplemented with Protease/Phosphatase Inhibitor Cocktail (Cell Signaling, 5872) following manufacturer's instructions. Protein concentration was measured using BCA kit (Thermo Fisher Scientific, 23225) and equal quantities were used to prepare samples for western blotting. Samples were resolved by electrophoresis with Bolt 4–12% Bis-Tris Plus Gels (Invitrogen) in Bolt MES SDS running buffer (Invitrogen, B0002) and transferred using iBlot 2 nitrocellulose transfer stacks (Invitrogen). Membranes were blocked for 1 h and probed with antibodies: BHLHE40 1:500 (Thermo Fisher Scientific, PA5-83044), BHLHE41 1:500 (Biorbyt, orb224120), APOE 1:1000 (Millipore, AB947), ABCA1 1:1000 (Abcam, 018180), ACTIN 1:10,000 (Sigma-Aldrich, A5441) in 5% non-fat dry milk in PBS/0.1% Tween-20 buffer overnight at 4 °C. Secondary antibody staining 1:10000 was applied for 1 h at RT, visualized using WesternBright ECL HRP Substrate Kit (Advansta, K-12045), and measured using iBrigh imagining system (Applied Bioscience). Images were analyzed using ImageJ (NIH). Uncropped western blot images are pasted in Source Data.

## Quantification of gene expression by RT-qPCR

iMGLs (40, 41KO, KO, DKO, and WT as control) were collected on day 28. MACs (40KD, 41KD, DKD and SCR as control) were collected 48 h after siRNA transfection. Cell pellets were used for mRNA extraction using the RNeasy Plus Mini kit (Qiagen, 74136) following manufacturer's instructions. mRNA quantity was measured using Nanodrop 8000 (Thermo Fisher Scientific) and reverse transcription reaction was performed with 1000 ng of RNA using High-Capacity RNA-to-cDNA kit (Thermo Fisher #4387406). 10 ng cDNA was used in the qPCR reaction with Power SYBR Green Master Mix (Applied Biosystems, 4368706) run using QuantStudio 7 Flex Real-Time PCR System (Thermo Fisher Scientific). Primers were designed using the IDT software and are listed in Supplementary Dataset 6. Ct values were averaged from two technical replicates for each gene, *GAPDH* Ct values were used for normalization. Gene expression levels were quantified using the $2^{-ddCt}$ method relative to control. We performed a total of 6-7 independent experiments with three technical replicates (i.e., separate wells) per experiment. Technical replicates were averaged within each experiment.

## Quantification of secreted cytokines by antibody array

Conditioned media (1 ml) was collected, cleared of cell debris with 10 min centrifugation at 4000 rpm and used for the Proteome Profiler Human Cytokine Array Kit, Panel A (R&D Systems, ARY005B) following manufacturers' instructions.

## Statistical analysis

Data were analyzed and visualized in GraphPad Prism 9 (GraphPad Software). In each analysis, three to six independent experiments were performed. The researcher was not blinded to siRNA treatment or genotype. Differences of means between groups were tested with one-way repeated measures ANOVA followed by Dunnett's post-hoc tests. All data are represented as group mean ± standard error of the mean (SEM). Detailed statistics are shown in Supplementary File 1.

## Mice

All mice used in this study were maintained on the C57BL/6 genetic background. Mice analyzed in this study were at least 2 and 6 months old. Mice were bred and maintained at Comparative Medicine Biomedicum facility of Karolinska Institutet (Stockholm, Sweden). All animal experiments were carried out according to valid project licenses, which were approved and regularly controlled by Swedish Veterinary Authorities. All mice were maintained in the animal facility under a 12-h light/dark cycle and access to food/water ad libitum.

## Acute isolation of microglia from Bhlhe40/41 double knockout mice

Female Bhlhe40$^{-/-}$Bhlhe41$^{-/-}$ (double knock-out, DKO) or wildtype (C57BL/6J or Rosa26$^{Stop-YFP}$) mice were anesthetized with ketamine (144 mg/kg, i.p.) and xylazine (14 mg/kg, i.p.). and transcardially perfused with 10 ml cold PBS prior to brain collection. Olfactory bulb and cerebellum were discarded, followed by enzymatic digestion of the remaining brain in IMDM medium (Cytiva, SH30259.02) supplemented with 1 mg/ml Collagenase type IV (Worthington, LS004186) and 33.3 U/ml DNase I (Roche, 11284932001) at 37 °C for 45 min with occasional mixing and dissociation by pipetting. Subsequently, cells were kept on ice or at 4 °C throughout further processing. Enzymatic digestion was followed by filtering through a 70 µm cell strainer, washing with ice-cold 2% FCS/PBS and spinning at 300 × g for 10 min. To remove myelin, the pellet was resuspended in 38% percoll/PBS (Cytiva, GE17-0891-02) and spun at 800 × g for 15 min (no break). Cells were washed in ice-cold 2% FCS/PBS, followed by staining with the following antibodies in ice-cold 2% FCS/PBS: CD38-PE (Miltenyi, 130-103-008), Ly6G-APC (Miltenyi, 130-120-803), Ly6C-BV510 (Biolegend, 127633), CD11b-FITC (Biolegend, 101206), MHCII-BV421 (Biolegend, 107632), CD45.2-PE-Cy7 (Biolegend, 109830), and FcR block (in-house, clone 2.4G2). Cells were washed in PBS, followed by staining with fixable viability dye eFluor 780 (eBioscience, 65-0865-14). Microglia were double sorted on a BD FACSAria Fusion using the following gating strategy: viability dye$^-$Ly6C$^-$Ly6G$^-$MHCII$^-$CD38$^-$CD45.2$^+$CD11b$^+$. The gating strategy is presented in Supplementary Fig. 11. Immediately after completion of the sort, cells were pelleted, washed with ice-cold PBS, and cells were disrupted by adding RLT plus buffer with β-mercaptoethanol following manufacturer's instructions (Qiagen, 1053393), followed by storage at −80 °C until further processing.

## RNA-seq analysis

RNA from human iPSC-derived microglia (iMGLs), human THP-1 macrophages (MACs), and acutely isolated mouse microglia were extracted using the RNeasy Plus Mini kit (Qiagen, 74136) following manufacturer's instructions. RNA was submitted to Azenta (New Jersey, NJ, USA) for QC, library preparation, and next-generation sequencing. Samples passed quality control with Qubit and BioAnalyzer showing RIN > 7.8. RIN for mouse samples were not assessed due to low RNA abundance. Libraries were prepared using TruSeq RNA Sample Prep Kit v2 and paired-end sequenced using HiSeq2500 at a read length of 150 bp to obtain 20−30 M mapped fragments per sample. Sequenced reads were assessed for quality (FastQC v0.11.8), trimmed for adapter contamination (Cutadapt v2.6), and aligned to the mouse genome mm10 or human genome hg38 (STAR v2.5.3a). Differential gene expression analysis (DGEA) was performed using a linear mixed model implemented in dream (differential expression for repeated measures, variancePartition R package v1.23.1 and R v3.5.3[98]). Genes with FDR Adj.*P* value < 0.05 were considered differentially expressed (DEGs). To identify pathways enriched in human iMGLs lacking BHLHE40 and/or BHLHE41, THP-1 macrophages with reduced levels of *BHLHE40* and/or *BHLHE41*, and mouse microglia lacking *Bhlhe40/41*, we used Gene Set Enrichment Analysis (GSEA)[99]. Briefly, ranked lists were generated from differential gene expression analyses by ordering genes according to the signed test statistic. This metric takes into account both the fold change across conditions as well as the standard error of the fold change. Ranked lists were analyzed using the "GSEA Preranked" module using default GSEA settings including 1000 permutations. Our preranked lists were tested for enrichment against genesets from the Molecular Signatures Database (MSigDB v7.5.1, Broad Institute, C5.all.v2022.1.Hs.symbols.gmt). Enrichment scores were normalized by geneset size to generate normalized enrichment scores (NES) according to the standard protocol[99] and these normalized enrichment scores were used to determine significance of enrichment[8]. DGEA and GSEA results are shown in

Supplementary Datasets (see Supplementary Dataset 3 for human iPSC-derived microglia, Supplementary Dataset 4 for human THP-1 macrophages, and Supplementary Dataset 5 for acutely isolated mouse microglia).

## Rank-rank hypergeometric overlap (RRHO) and Ingenuity pathway (IPA) analyses

Transcriptional signatures from human lipid-associated macrophages (ref. 6, Dataset S6, FDR Adj.*P* value < 0.05) and from BHLHE40-KO, BHLHE41-KO, and BHLHE40/41-DKO human iPSC-derived microglia (iMGLs), BHLHE40-KD, BHLHE41-KD, BHLHE40/41-DKD human THP-1 macrophages as well as from mouse disease-associated microglia (ref. 8, Table S3, FDR Adj.*P* value < 0.05) and mouse Bhlhe40/41-DKO were compared pairwise using the RRHO2 R package[53,54,100]. The recommended $-log10(P\ value) * sign(log2FC)$ metric was used to generate ranked lists of genes for each transcriptional signature. RRHO2 was then used to visualize both concordant and discordant gene expression changes across each pair of signatures as rank-rank hypergeometric overlap (RRHO) heatmaps. The color temperature of each pixel in an RRHO heatmap represents the negative log10-transformed hypergeometric overlap test *P* value of subsections of the two ranked gene lists, adjusted for multiple testing using the Benjamini-Hochberg correction method. Heatmaps generated using RRHO2 have top-right (both decreasing) and bottom-left (both increasing) quadrants, representing concordant gene expression changes, while the top-left and bottom-right quadrants represent discordant gene expression changes. The default step size and *P* value representation method (*hyper*) were used. Gene lists that provide the most significant overlap (expression changes going down or up in compared signatures) were retrieved using overlap_uu or overlap_dd options. Pathway (biological functions and diseases) enrichment and activity analyses of overlapping genes were performed using Ingenuity Pathway Analysis (Qiagen Inc., https://digitalinsights.qiagen.com/IPA) using Z statistics of differential gene expression retrieved from bulk RNA-seq analyses (Supplementary Dataset 2).

## Gene sets enrichment analysis (GSEA)

Gene sets enrichment analysis were performed in a ranked list of human and mouse DKO transcripts provided in Supplementary Dataset 3 (human) and Supplementary Dataset 5 (mouse). Gene sets and their sources are listed in Supplementary Dataset 1. We used MSigDB app (ver 4.3.2) and 1000 permutations.

## Reporting summary

Further information on research design is available in the Nature Portfolio Reporting Summary linked to this article.

## Data availability

All aligned read counts for WT, 40KO, 41KO, DKO human iPSC-derived microglia (iMGLs), and SCR, 40KD, 41KD, and DKD human THP-1 macrophages (MACs), as well as microglia acutely isolated from the brain of *Bhlhe40/41* DKO mice has been deposited to the Gene Expression Omnibus and available and are available under following accession numbers GSE253943 (iMGLs), GSE253992 (THP-1 macrophages), GSE254233 (mouse microglia). Source data are provided with this paper.

## Code availability

Code used for network reconstruction can be found at our github page https://github.com/marcoralab/bhlhe_manuscript. This repository has been linked to Zenodo (https://doi.org/10.5281/zenodo.10516418)[101].

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

## Acknowledgements

This work was funded by grants from the NIH: RF1AG054011 (A.M.G.), U01AG058635 (A.M.G., E.M.), R56AG081417 (E.M. A.M.G.) from NIA: U01AG066757 (A.M.G., E.M.), The JPB Foundation (A.M.G.), BrightFocus Foundation A2021014F (A.P-D.), Training Program in Stem Cell Biology fellowship from the New York State Department of Health (NYSTEM-C32561GG) (A.P-D.). C.G. also acknowledges support from grants NIH/NHLBI R01HL153712 and AHA 20SFRN35210252. G.N. acknowledges support from Graduate Women in Science Fellowship. This work was supported in part through the computational and data resources and staff expertise provided by Scientific Computing and Data at the Icahn School of Medicine at Mount Sinai and supported by the Clinical and Translational Science Awards (CTSA) grant UL1TR004419 from the National Center for Advancing Translational Sciences. Research reported in this publication was also supported by the Office of Research Infrastructure of the National Institutes of Health under award number S10OD026880 and S10OD030463.

## Author contributions

Conceptualization, study design and methodology: G.N., A.P-D., A.M.G., E.M.; data collection, analysis and visualization: G.N., A.P-D.; sample/dataset generation: R.T. (iMGLs), J.D., T.K. (mouse microglia), C.G. (human atherosclerotic macrophage dataset); strategy for CRISPR/Cas9 genome editing: S.M.; RNA-seq data processing: Y.L.; writing of original draft, G.N., A.P-D.; writing, review, revising, A.M.G., E.M., T.K. All authors read and approved the final manuscript.

## Competing interests

A.M.G.: Scientific Advisory Board (SAB) Genentech; SAB Muna Therapeutics; S.M.: consultant Dorian Therapeutics, Turn Biotechnologies. C.G. is listed as an inventor on Tech 160808G PCT/US2022/017777 filed by the Icahn School of Medicine at Mount Sinai, which has no competing interest in this work. G.N. is an employee of Genentech, a member of the Roche group, and owns company stock. The remaining authors declare that they have no competing interests.

## Additional information

[1]Department of Genetics and Genomic Sciences, Icahn School of Medicine at Mount Sinai, New York, NY 10029, USA. [2]Department of Medicine, Division of Immunology and Allergy, Karolinska Institutet, Karolinska University Hospital, Stockholm, Sweden. [3]Center for Molecular Medicine, Karolinska Institutet, Stockholm, Sweden. [4]Department of Medicine, Division of Cardiology, NYU Cardiovascular Research Center, New York University School of Medicine, New York, NY, USA. [5]Department of Pathology, New York University School of Medicine, New York, NY, USA. [6]Nash Family Department of Neuroscience, Friedman Brain Institute, Icahn School of Medicine at Mount Sinai, New York, NY 10029, USA. [7]Black Family Stem Cell Institute, Icahn School of Medicine at Mount Sinai, New York, NY 10029, USA. [8]Present address: OMNI Bioinformatics Department, Genentech, Inc., South San Francisco, CA, USA. [9]These authors contributed equally: Anna Podleśny-Drabiniok, Gloriia Novikova. [10]These authors jointly supervised this work: Edoardo Marcora, Alison Mary Goate. ✉e-mail: edoardo.marcora@mssm.edu; alison.goate@mssm.edu

