## [Peer Review File · Nature Communications]

BHLHE40/41 regulate microglia and peripheral macrophage responses associated with Alzheimer's disease and other disorders of lipid-rich tissuesREVIEWER COMMENTS

Reviewer #1 (Remarks to the Author):

This manuscript by Podlesny-Drabiniok, Novikova, and colleagues leverages existing snRNAseq datasets from various disease states involving lipid-accumulating macrophages or microglia in order to identify transcriptional regulators of lipid metabolism. In addition to several other TFs, the authors identify BHLHE40/41 as a primary regulator of the 'lipid-associated macrophage' ("LAM") response. They then use gene editing in human iPSC derived microglia to knockout BHLHE40 and/or 41, as well as siRNA knockdown in THP-1 human macrophages, and finally acutely isolated microglia from knockout mice, to functionally assess the effects of 40/41 manipulation in macrophages and microglia. Their results show that across these model systems, loss (or significant reduction) of 40/41 expression leads to increased expression of "LAM" genes, and increase in lipid storage in the form of lipid droplets, an increase in lysosomal mass and decreased pH, a reduction in cytokine production, and an increase in cholesterol efflux. A primary weakness of the study is a lack of clarity and justification in how the "LAM" profile was defined based on previous studies. Nonetheless, the study has several strengths, including the use of multiple model systems for validation and phenotyping experiments, and its focus on a novel and little studied AD GWAS gene. Conceptually, this manuscript is exciting and important, as it nicely ties together several important disease-relevant themes in efferocytosis, GWAS risk genes, and macrophage/microglia function in disorders of lipid rich tissue, with a special emphasis and importance for Alzheimer's disease.

Some specific comments, concerns, and questions are listed below:

The abstract and introduction conflate macrophages and microglia in several instances, several of which are worded in a way that is not entirely accurate. The authors should be clear and consistent in their phrasing (i.e. using either macrophage or microglia, or / when talking about both simultaneously).

The text in the graphical abstract is too small in almost all areas. It also was not entirely clear how the "LAMs" and "sc/nRNAseq" sections were different or not. Several figures also have text that is too small to read – Figure 1 is a prime example, but there are many others.

Previously used acronyms for microglial states, of which there are now too many to keep track of, are based on defined gene signatures (a list of genes). The authors refer to LAM genes or LAM signature genes multiple times, but unless the reviewer missed it, a list of genes that constitute LAM genes is never defined or provided. Selected lipid/lysosomal genes such as those in Figure 6 for example, are logical candidates, but with the focus of the manuscript being the LAM 'state', this concept must be defined much more clearly.

Two of the three human datasets used for TF validation are mouse datasets that are simply translated to their respective human homologs. Thus, it doesn't seem appropriate to refer to these as distinct lists or as a 'human' dataset per se.

In figures employing the RRHO, the LAM dataset is simply the Keren-Shaul gene list. Thus, it's the DAM gene list renamed LAM. Similar to the points above, the justification is unclear here as is the definition of 'LAM' in this sense.

The datasets from Marschallinger (LDAM) and Claes (plaque associated LD) do not appear to be included in the LAM lists the authors include. These seem like highly relevant studies given the focus; was there a reason they were not utilized? These references (57 and 60) should also probably be mentioned much earlier in the manuscript given their relevance. The manuscript states that these microglia are "distinct from LAMs" – what does this mean exactly, and how is this concluded?

What are the “technical differences” mentioned in the first section of Results? sc vs sn RNAseq?

For reference, what does a “control” look like in the analysis in Supp Fig 1a-b? (i.e. non-prominent peaks)

The last sentence of the Results section “ BHLHE41 and SPI1/PU.1 likely regulate” seems to be a possible overstatement.

THP-1 macrophages should be defined at first mention (i.e. what is THP-1)

What were the media conditions that the iMGLs were cultured in? specifically the lipid content of the media – FBS concentration? Lipid supplementation? How might this be affecting the results?

The authors mentions “likely due to low statistical power (figure 5a, ...” What is the actual n =5 here? It is 5 wells of the same iMGL line, not 5 distinct iMGL lines, correct? If 5 wells/samples, that would seem to reduce variation dramatically.

What is known regarding the functional consequences of BHLHE40/41 mutations in AD? This should be addressed in the Discussion and perhaps introduction as well.

How do the authors think APOE4 fits into their findings here? Particularly, the findings from this same group (and others) that show E4 expression is tied to increased lipid droplet formation, but also increased cytokine release and decreased efflux.

References 60 and 76 appear to be the same paper

Reviewer #2 (Remarks to the Author):

In the present manuscript Podlesny-Drabiniok, Novikova et al. postulate that the transcription factors BHLHE40 and BHLHE41 at least partially drive the gene expression signature of LAMs (Lipid-Associated Macrophages). Both TFs are helix-loop-helix proteins with repressive functions. The authors used publicly-available transcriptomic data from different myeloid cell types (e.g., microglia, Kupffer cells, macrophages from adipose tissue) in the context of different diseases (Alzheimer’s disease, Obesity, Atherosclerosis and Steatosis) from human and mouse tissue. The authors defined human and mouse LAM signature genes and based on this gene list and nominated a gene regulatory network consisting of 74 transcription factors as potential drivers of the LAM phenotype. Next, the authors show that BHLHE41 DNA-binding motif is enriched in promoters of LAM genes and AD risk SNPs. Knockout of BHLHE41 and the double knockout of BHLHE40/41 in human iPSC-derived microglia like cells resulted in a partially up-and down-regulation of genes associated with LAM gene expression signature. In addition, functional assays showed increased cholesterol efflux, lipid droplets and lysosomal acidification/proteolysis. Finally, in vivo double knockout of Bhlhe40 and Bhlhe41 resulted in an altered gene expression signature of mouse microglia partially recapitulating the LAM transcriptional response.

Overall, the study is of interest because it nominates potential drivers of the LAM phenotype and the major conclusions as stated in the abstract are supported by experimental data. There are some points for which clarifications would strengthen the manuscript.

Major points:

- The finding that loss of function of BHLHE40/41 results in increased cholesterol efflux and lipid droplet formation is of interest. The increase in cholesterol efflux is consistent with the proposed role of BHLHE40/41 as negative regulators of LXRs. However, the increase in lipid droplets is not necessarily linked to increased functions of LXRs. Although not central to the major conclusions, the

manuscript would be strengthened by lipidomic analyses to establish the lipid classes that are associated with these lipid droplets. This is important because lipid droplets in macrophage foam cells of atherosclerotic lesions are enriched in cholesterol esters, whereas those that accumulate in microglia in the context of aging are enriched for triglycerides, diglycerides, and phospholipids with very little cholesterol ester. Mechanisms underlying accumulation of these different lipid classes are not the same, with accumulation of TGs invoking the SREBP pathway. This could potentially involve LXRs via LXR activation of SREBP1c, the signal for which should be evident in the data. Thus, further information on lipid content and LXR/SREBP target genes would be of interest.

- It is difficult to reconstruct how the authors arrived at the different human and mouse “LAM” marker genes. A list of LAM marker genes is not provided in the manuscript. The number of LAM genes in mouse seems to be very high with 1,453 genes as written in Figure 7B. The authors should clarify this.
- In studies using *Bhlhe40*^{-/-}*Bhlhe41*^{-/-} double knockout mice, the authors should comment on the extent to which *Bhlhe40/41* are expressed by other brain cell types. If they are expressed, the authors should include the possibility that some of the effects of the knockout could be non-cell autonomous in the limitations section. It would also be of interest to know whether lipid droplets were observed in microglia from these mice.

Minor comment:

- RNA-seq data is presented as RRHO heatmaps. It would be informative to present the data also as Volcano plots depicting fold change and adj p-value and to state, what the fold-cut off were to determine differentially expressed genes. This can be shown as supplementary data.

Reviewer #3 (Remarks to the Author):

Communications – Review March 2023

BHLHE40/41 regulate macrophage/microglia responses associated with Alzheimer’s disease and other disorders of lipid-rich tissues

The manuscript by Podlesny-Drabiniok and Novikova et al. outlines the core gene-regulatory-network (GRN) function of the transcription factors; BHLHE40/41, in microglia and macrophages in lipid-rich environments in various diseases. Combining multiple publicly available single-cell and single-nucleus RNA-seq datasets, as well as ATAC-seq, the authors show that *Bhlhe40/41* is a core transcriptional regulators of LAM microglia/macrophages and provide evidence of functional responses of microglia upon genetic perturbation of *Bhlhe40/41*. Overall, the manuscript provides an important insight to the transcriptional impact of *Bhlhe40/41* on cholesterol metabolism and lysosomal activity in microglia. However, the initial GRN analysis leaves several questions to be addressed as indicated below:

Major comments:

1. The methods for the reconstruction of gene-regulatory-network must be described in more detail. Figure 1 shows the consolidated results for the GRN analysis. It is not clear whether metacells were constructed separately for the different expression matrix files (different datasets) or processed together as one merged expression matrix. If individual networks were built for each expression matrix, the intermediate results should be presented.
2. Why were all metacells created using standard parameters. The authors should provide evidence that standard parameters provided the most robust K-means clustering across different datasets.
3. Most of the input datasets for GRN analysis are not associated to Alzheimer’s disease and/or microglia. Yet, the authors are using the consolidated results in Figure 1 to justify why BHLHE40/41 is

studied in microglia. In silico, the authors need to provide more insight of the GRN analysis in microglia.

4. The authors implement the PISCES tool to strengthen the expression profile of low expression genes. The tool is not yet peer-reviewed and it is not clear why the authors only used single-cell and single-nucleus RNA-seq although bulk RNA-seq provides much better sequencing depth.

5. It is not clear to what extent the LAM signature is present in the datasets used. How many cells are positive for the LAM signature, and do they cluster separately from DAM microglia? Where only LAM signature microglia/macrophages implemented for the GRN analysis?

6. The authors mention efferocytosis as common hallmark pathway for LAM microglia/macrophages, however, do not show any data supporting this statement.

7. When generating the short list of candidate transition factors, the researchers require the transcription factor to appear in at least half the human and mouse networks and present in human microglia. The authors need to provide data visualization displaying which networks contain these transcription factors.

8. The reason for the expression-based selection of top transcription factor markers is not clear. Protein levels of transcription factors should be considered. Even low expressed transcription factors could have biological and functional meaning. Selecting only the top 11 candidates before investigating AD risk allele enrichment and TF binding sites could skew the results and hide other potentially important transcription factors. The authors must present a less biased approach.

9. The significance values of enrichment for BHLHE40 binding to ATAC seq peaks for LAM genes is missing. The authors should demonstrate more transparency and show HOMER results for Figure 3.

10. There are several datasets on BHLHE40 ChIP-seq, which should be investigated and used to validate potential BHLHE40 binding site enrichment at LAM genes (Citation: 59, PMID: 31061528).
Minor comments:

1. It is not clear how the Human and mouse LAM TFs in Figure 1 were clustered. Were the LAM TFs ranked, and if so, using what method?

2. The authors should provide all code used for the GRN analysis for more transparency.

3. The authors should comment on the low MI scores between LAM genes and TFs in mice in Figure 1.

4. Peak profiles of ATAC-seq for Bhlhe40 motif enrichment sites at LAM genes should be presented, as well as ChIP of Bhlhe40.

5. Why did the authors choose to select LPS and IL4 treated microglia?

6. Using stratified LD score regression to perform AD heritability analysis it is not clear if the P-values are adjusted for false discovery?

REVIEWER COMMENTS

Reviewer #1 (Remarks to the Author):

This manuscript by Podlesny-Drabiniok, Novikova, and colleagues leverages existing snRNAseq datasets from various disease states involving lipid-accumulating macrophages or microglia in order to identify transcriptional regulators of lipid metabolism. In addition to several other TFs, the authors identify BHLHE40/41 as a primary regulator of the ‘lipid-associated macrophage’ (“LAM”) response. They then use gene editing in human iPSC derived microglia to knockout BHLHE40 and/or 41, as well as siRNA knockdown in THP-1 human macrophages, and finally acutely isolated microglia from knockout mice, to functionally assess the effects of 40/41 manipulation in macrophages and microglia. There results show that across these model systems, loss (or significant reduction) of 40/41 expression leads to increased expression of “LAM” genes, and increase in lipid storage in the form of lipid droplets, an increase in lysosomal mass and decreased pH, a reduction in cytokine production, and an increase in cholesterol efflux. A primary weakness of the study is a lack of clarity and justification in how the “LAM” profile was defined based on previous studies. Nonetheless, the study has several strengths, including the use of multiple model systems for validation and phenotyping experiments, and it’s focus on a novel and little studied AD GWAS gene. Conceptually, this manuscript is exciting and important, as it nicely ties together several important disease-relevant themes in efferocytosis, GWAS risk genes, and macrophage/microglia function in disorders of lipid rich tissue, with a special emphasis and importance for Alzheimer’s disease.

Some specific comments, concerns, and questions are listed below:

1.1. The abstract and introduction conflate macrophages and microglia in several instances, several of which are worded in a way that is not entirely accurate. The authors should be clear and consistent in their phrasing (i.e. using either macrophage or microglia, or / when talking about both simultaneously).

We thank reviewers for this comment. We have revised the text accordingly.

1.2. The text in the graphical abstract is too small in almost all areas. It also was not entirely clear how the “LAMs” and “sc/nRNAseq” sections were different or not. Several figures also have text that is too small to read – Figure 1 is a prime example, but there are many others. We apologize for the inconvenience. We would like to clarify that the figure miniatures at the bottom of the submitted manuscript were placed there for convenience; high resolution images were also uploaded separately. We made sure to provide a high resolution image to reveal the details included in each figure for the resubmission. We have also modified the graphical abstract and other figures to make the text more legible when printed out at the final size and resolution.

1.3. Previously used acronyms for microglial states, of which there are now too many to keep track of, are based on defined gene signatures (a list of genes). The authors refer to LAM genes or LAM signature genes multiple times, but unless the reviewer missed it, a list of genes that

constitute LAM genes is never defined or provided. Selected lipid/lysosomal genes such as those in Figure 6 for example, are logical candidates, but with the focus of the manuscript being the LAM 'state', this concept must be defined much more clearly.

We would like to clarify that the term "LAM" (lipid-associated macrophages) was used to collectively refer to specific microglial/peripheral macrophages subpopulations identified in 5xFAD mice, AD brains, adipose tissue, atherosclerotic plaques etc. It does not indicate a new name for a new subpopulation. The reviewer is right that there are several different acronyms for microglia/peripheral macrophage states/subpopulations such as DAM, LAM, TREM2^{high} among others, which share similar gene signatures and cellular responses to damage of lipid-rich tissues. Therefore, in the previous version of this manuscript, we decided to collectively refer to these microglial/peripheral macrophages subpopulations as LAM (for lipid associated macrophages). However, we understand that this collective name "LAM" may cause confusions because LAM is also a specific gene signature of adipose tissue macrophages identified by Jaitin et al., 2019. In the revised version of the manuscript we decided to use the term DLAMs to collectively refer to these microglial/peripheral macrophages subpopulations. This is also explained in the Introduction: "For brevity in this manuscript, we collectively refer to these subpopulations of microglia and peripheral macrophages as DLAMs" (page 3).

To compare gene signatures from BHLHE40 and/or BHLHE41 knockout iMGLs and THP-1 macrophages and Bhlhe40/41-DKO mice using RRHO, we used specific gene signatures. For human data, we used LAM signature published by Jaitin et al., 2019 (Dataset S6, Adj.P-value < 0.05)(Jaitin et al. 2019), for mouse data, we used DAM signature from Keren-Shaul et al., 2017 (Table S3, Adj.P-value 0.05) (Keren-Shaul et al. 2017). We referenced each specific gene signature in the legend, main text, and list the genes in each gene signature in the Supplementary Table 1. Additionally, in this version of the manuscript we also included other disease-relevant genesets to test their enrichment in human and mouse DKO transcriptome, please see Response 1.5 below.

1.4. Two of the three human datasets used for TF validation are mouse datasets that are simply translated to their respective human homologs. Thus, it doesn't seem appropriate to refer to these as distinct lists or as a 'human' dataset per se. In figures employing the RRHO, the LAM dataset is simply the Keren-Shaul gene list. Thus, it's the DAM gene list renamed LAM. Similar to the points above, the justification is unclear here as is the definition of 'LAM' in this sense. We apologize that it was unclear but in the figures showing RRHO results obtained from human cells (iMGL in Figure 4 and THP-1 macrophages in Supplementary Figure 7) we used the human LAM signature from (Jaitin et al. 2019) Dataset S6, Adj.P-value < 0.05. On the other hand, in the figures showing RRHO results obtained from mouse microglia (Figure 7) we used the mouse DAM signature from (Keren-Shaul et al. 2017) Table S3 Adj.P-value < 0.05). We have now changed the name to "lipid-associated macrophages" and "disease-associated microglia", we also reference the Supplementary Table 1 in which these gene lists are included in addition to the main text and the figure legends.

1.5. The datasets from Marschallinger (LDAM) and Claes (plaque associated LD) do not appear to be included in the LAM lists the authors include. These seem like highly relevant studies

given the focus; was there a reason they were not utilized? These references (57 and 60) should also probably be mentioned much earlier in the manuscript given their relevance. The manuscript states that these microglia are “distinct from LAMs” – what does this mean exactly, and how is this concluded?

We thank the reviewer for the comment.

Marschallinger et al provided an important and in-depth characterization of lipid droplet-accumulating microglia (LDAMs) in the aging brain. However, these cells appear to be distinct from those observed in DLAMs. More specifically, DLAMs are enriched in phagocytosis and lipid metabolism genes, suggesting enhancement of these processes, while (Marschallinger et al. 2020) suggest that LDAMs exhibit phagocytosis deficits. At the transcriptional level, DLAMs express canonical markers, such as *SPP1*, *LPL*, *APOE*, *CD9*, *AXL*, and *CLEC7A*. Marschallinger et al conducted a comparative analysis to DAM/MGnD microglia and concluded that LDAMs are a distinct population, with some important marker genes that were downregulated in LDAMs being upregulated in DAM/MGnD microglia (e.g., *AXL*, *CLEC7A*). The authors conclude that LDAMs show a “unique transcriptome signature that is distinct from previously described microglia states observed in aging and neurodegeneration.” Hence, we did not include this gene expression signature in our analyses because a) the signature is largely distinct from the DLAM genesets we set out to study and b) the data provided in the study are bulk RNA-seq generated on isolated CD11b⁺CD45^{low} cells, while we solely focus on single cell and single nucleus RNA-seq data in this study. In addition, we have performed RRHO analysis comparing the entire LDAM and mouse DKO transcriptomes and we found no significant correlation (Supplementary Figure 9B).

(Claes et al. 2021) have reported exciting findings using xenografted iPSC-derived microglia (xMGLs), suggesting that human microglia response to beta-amyloid plaques could be similar to that of foam cells in atherosclerotic plaques. However, a previous study by the same group (Hasselmann et al. 2019) showed that xMGLs demonstrate “limited overlap existing between human xMG and mouse DAM genes”. Altogether this suggests that xMGLs acquire a foam

cell-like signature and accumulate lipid droplets but they do not closely resemble the DAM phenotype observed in mouse 5xFAD brains.

Nevertheless, we understand the value of including additional gene signatures that may be relevant to our study. To this end, we performed gene set enrichment analysis using the following human genesets: DAM clusters from iMGLs exposed to CNS-relevant phagocytic substrate (Dolan et al. 2023); amyloid plaque-associated microglia from human brains (Gerrits et al. 2021); lipid associated macrophages from adipose tissue (Jaitin et al. 2019); human foam cell signature (Fernandez et al. 2019); xenografted iPSC-derived human microglia from 5xFAD mouse brains (Claes et al. 2021). We split each gene set into up- and downregulated genes and examined their enrichment in the transcriptome of iMGLs lacking BHLHE40 and BHLHE41 (DKO) because DKO iMGLs recapitulated most of the functional aspects of DLAMs. The results are presented in Figure 4D. Briefly, we found significant positive enrichment of genes upregulated in DAM, LAM, and foam cell signatures and negative enrichment of genes downregulated in DAM and LAM clusters (Page 12-13).

We observed a highly positive enrichment of DAM markers reported by (Dolan et al. 2023) in the DKO transcriptome which prompted us to use RRHO to support GSEA analysis (similar to what we did for lipid-associated macrophages, see Figure 4A). As expected, we found that Cluster 2 DAM markers and Cluster 8 DAM markers are highly correlated with the DKO transcriptome suggesting that reduction of BHLHE40/41 levels facilitates transition of iMGLs toward a DLAM-like phenotype (Supplementary Figure 9A). Of note, (Dolan et al. 2023) used highly lipid-rich substrates such as myelin, synaptosomes and apoptotic neurons supporting our observation that exposure to lipid overload activates the DLAM clearance program similar to the one observed in peripheral macrophages in proximity of adipose tissue or atherosclerotic plaques.

Using GSEA we also found a negative enrichment of upregulated genes upregulated in disease-associated xMGLs and a positive enrichment of gene downregulated in disease-associated xMGLs suggesting the opposite regulation of xMGL DAM genes in our BHLH40/41 DKO iMGLs. As stated above it might be due to the fact that the DAM population in xMGL is different from the DAM reported by (Keren-Shaul et al. 2017) from 5xFAD brains. Here we also show that xMGL DAM is different from DAM reported by (Dolan et al. 2023) We performed RRHO and found no overlap between xMGL DAM (by (Claes et al. 2021)) and DAM reported by (Dolan et al. 2023) (Cluster 2) supporting the statement that xMGL DAM is a distinct population.

Next, we performed gene set enrichment analysis using the following mouse gene sets: disease associated microglia (DAM) from 5xFAD mouse brains (Keren-Shaul et al. 2017); CD11c-positive microglia sorted from 5xFAD mouse brains (Kamphuis et al. 2016); activated-response microglia (ARM) and Homeostatic microglia (Sala Frigerio et al. 2019); lipid-associated macrophages (LAM) from visceral adipose tissue of obese mice (Jaitin et al. 2019); Trem2^{high} atherosclerotic macrophages (Cochain et al. 2018); neurodegeneration module (Friedman et al. 2018). We found positive enrichment of upregulated genes in almost all gene

sets and negative enrichment of downregulated (or Homeostatic upregulated) genes in the transcriptome of mouse microglia lacking Bhlhe40 and Bhlhe41 (DKO) (Page 20). We included these analyses in Figure 7D. All gene sets used to run GSEA are listed in Supplementary Table 1.

1.6. What are the “technical differences” mentioned in the first section of Results? sc vs sn RNAseq?

Technical differences between human and mouse datasets include differences in sample preparation such as single-cell vs single-nuclei RNAseq, differences in brain regions where human microglia were isolated from, and overall viability of isolated microglia from the human brain that vary across multiple experiments. Please see the revised text in the end of paragraph titled: Geneset enrichment analysis of TF regulons nominates candidate DLAM TFs in human and mouse macrophages/microglia (Page 7).

1.7. For reference, what does a “control” look like in the analysis in Supp Fig 1a-b? (i.e. non-prominent peaks)

Supplementary Figure 1A-B shows that SPI1 and BHLHE41 regulons are also enriched in those respective motifs, suggesting that the genes we identify as regulated by SPI1 and BHLHE41 are also potentially bound by those TFs in the promoter. Below we include two negative examples. Although JUNB and MAFB are both nominated as DLAM regulators through co-expression, their motifs do not seem to be enriched in the promoters of DLAM genes (as DLAM genes we tested human LAM reported by (Jaitin et al. 2019) Dataset S6, FDR < 0.05, and mouse DAM reported by (Keren-Shaul et al. 2017) Table S2)(Figure 3A-D). Additionally, selecting TFs that are

expressed highly and specifically in microglia and examining their motif instances in SPI1 and BHLHE41 regulons also suggest that the regulons for these two TFs are enriched for their putative direct targets.

As an example, when we look at motif instances of ATF3 in BHLHE41 and SPI1 regulons, we do not see a peak, indicating lack of enrichment:

When we examine at CEBPB motif instances in BHLHE41 and SPI1 regulons, we see that those motif instances are depleted:

This suggests that SPI1 and BHLHE41 regulons include putative direct targets of SPI1 and BHLHE41 as indicated by the enrichment of their motif instances in regulon gene promoters.

1.8. The last sentence of the Results section “ BHLHE41 and SPI1/PU.1 likely regulate” seems to be a possible overstatement.

Analysis of available ChIPseq in microglial cells (mouse) showed PU.1/SPI1 binds ~50% of DAM genes (Keren-Shaul et al., 2017 (Keren-Shaul et al. 2017), Table S2). In addition, our previous work on PU.1 in BV2 mouse microglia showed that reduction of PU.1 (associated with AD protection) is positively correlated with DAM gene expression and related pathways (Pimenova et al. 2021). Finally, we added an example of epigenomic tracks highlighting open-chromatin regions that contain a BHLHE41 motif in DLAM gene promoters for CD63 and CTSB loci in mouse and human macrophages and microglia (Page 9). Please see revised Figure 3E–F.

1.9. THP-1 macrophages should be defined at first mention (i.e. what is THP-1).

We added a brief explanation that THP-1 is a monocytic leukemia line. Please see paragraph *Knockdown of BHLHE40/41 partially recapitulates the LAM...*, first sentence. (Page 17)

1.10. What were the media conditions that the iMGLs were cultured in? specifically the lipid content of the media – FBS concentration? Lipid supplementation? How might this be affecting the results?

iPSC-derived microglia (iMGLs) were cultured without FBS or FCS. Media formulation was used exactly as previously described by McQuade et al. (McQuade et al. 2018) THP-1 macrophages also had a limited amount of serum. Briefly, THP-1 monocytes were cultured and differentiated in the presence of serum (10% FBS) and then rested without serum in the presence of 1% BSA fraction V for 24h, followed by transfection 48h. There was no additional lipid supplementation in any cell culture at any time point therefore we think lipid abundance in both culture conditions for iMGL and THP-1 was similar and did not affect the final results. We added this statement in the method section describing culture conditions. (Page 31-32)

1.11. The authors mention “likely due to low statistical power (figure 5a, ...” What is the actual n = 5 here? It is 5 wells of the same iMGL line, not 5 distinct iMGL lines, correct? If 5 wells/samples, that would seem to reduce variation dramatically.

The reviewer is correct, N in this figure is independent iMGL differentiations (different days of iMGL differentiation and collection), and not independent lines (donors or clones). Although some genes whose expression level was assessed by qPCR did not reach statistical significance, the effect (increased expression) was present. We also calculated the effect size for each gene/comparison and listed it in Supplementary File 1. For some genes the effect size is small and therefore would require increased N ($N > 5$) to achieve sufficient statistical power.

1.12. What is known regarding the functional consequences of BHLHE40/41 mutations in AD? This should be addressed in the Discussion and perhaps introduction as well.

As mentioned in the main text, BHLHE40 is a candidate AD risk gene by virtue of the fact it resides in the vicinity of an AD risk locus recently identified in an African-American AD GWAS study (Kunkle et al. 2021). Disease-associated GWAS loci are mostly non-coding common genetic variants that modulate disease risk typically by regulating the expression of one or more nearby genes. It took several years and research groups (including ours) to nominate, based not only on proximity but also on the integration of functional genomic evidence, the most likely causal genes at about half of the AD risk loci identified by much larger and older GWAS studies

in European individuals (Novikova et al. 2021), a population for which functional genomics datasets are already available. Unfortunately, functional follow-up of the genetic associations identified in African-American AD GWAS studies is still lacking also because functional genomics datasets in minority populations are very scarce. We are in the process of generating such datasets from monocyte-derived macrophages obtained from African-American individuals, but it is an effort that will take several years to complete. Therefore, at the present time, proximity is the only criterion (albeit usually ~70% accurate) used to nominate BHLHE40 as a candidate AD risk gene.

1.13. How do the authors think APOE4 fits into their findings here? Particularly, the findings from this same group (and others) that show E4 expression is tied to increased lipid droplet formation, but also increased cytokine release and decreased efflux.

We thank the reviewer for this comment, this is a very interesting point. Our large RNAseq-studies of APOE44 population and isogenic iPSC-derived microglia suggested that APOE44 iMGL showed 1) decreased LXR- and Mit/TFE-mediated responses, 2) lower expression of core DLAM genes, 3) fewer cells in the DAM cluster in scRNAseq. Therefore, our current efforts are to increase LXR activity using either novel pharmacological approaches or a genetic inactivation of BHLHE40/41 which are LXR and Mit/TFE repressors. We hypothesize that inactivation of BHLHE40/41 in APOE44 iMGL will rescue lipid accumulation and lysosomal storage deficits as well as normalize the proportion of microglia in the DAM cluster observed in our preliminary scRNAseq. One caveat is related to increased accumulation of lipid droplets in the risk (APOE44 iMGL) and protective model (BHLHE40/41-DKO). We think that accumulation of lipid droplets in APOE44 may be an adaptation to increased cholesterol biosynthesis and decreased cholesterol efflux. In that setting, LD is one of the buffering mechanisms sequestering the excess of toxic free cholesterol. We believe that when we facilitate lipid and lysosomal clearance processes in APOE44 iMGL, we will be able to restore lipid droplets phenotype or at least not increase the LD content. Future studies should also focus on profiling LD content in BHLHE40/41 and APOE44 iMGL to understand their composition.

1.14. References 60 and 76 appear to be the same paper

We thank the reviewer for that comment, we have now corrected the reference list.

Reviewer #2 (Remarks to the Author):

In the present manuscript Podlesny-Drabiniok, Novikova et al. postulate that the transcription factors BHLHE40 and BHLHE41 at least partially drive the gene expression signature of LAMs (Lipid-Associated Macrophages). Both TFs are helix-loop-helix proteins with repressive functions. The authors used publicly-available transcriptomic data from different myeloid cell types (e.g., microglia, Kupffer cells, macrophages from adipose tissue) in the context of different diseases (Alzheimer's disease, Obesity, Atherosclerosis and Steatosis) from human and mouse tissue. The authors defined human and mouse LAM signature genes and based on this gene list and nominated a gene regulatory network consisting of 74 transcription factors as potential drivers of the LAM phenotype. Next, the authors show that the BHLHE41 DNA-binding motif is

enriched in promoters of LAM genes and AD risk SNPs. Knockout of BHLHE41 and the double knockout of BHLHE40/41 in human iPSC-derived microglia like cells resulted in a partially up-and down-regulation of genes associated with LAM gene expression signature. In addition, functional assays showed increased cholesterol efflux, lipid droplets and lysosomal acidification/proteolysis. Finally, in vivo double knockout of Bhlhe40 and Bhlhe41 resulted in an altered gene expression signature of mouse microglia partially recapitulating the LAM transcriptional response.

Overall, the study is of interest because it nominates potential drivers of the LAM phenotype and the major conclusions as stated in the abstract are supported by experimental data. There are some points for which clarifications would strengthen the manuscript.

Major points:

2.1. The finding that loss of function of BHLHE40/41 results in increased cholesterol efflux and lipid droplet formation is of interest. The increase in cholesterol efflux is consistent with the proposed role of BHLHE40/41 as negative regulators of LXRs. However, the increase in lipid droplets is not necessarily linked to increased functions of LXRs. Although not central to the major conclusions, the manuscript would be strengthened by lipidomic analyses to establish the lipid classes that are associated with these lipid droplets. This is important because lipid droplets in macrophage foam cells of atherosclerotic lesions are enriched in cholesterol esters, whereas those that accumulate in microglia in the context of aging are enriched for triglycerides, diglycerides, and phospholipids with very little cholesterol ester. Mechanisms underlying accumulation of these different lipid classes are not the same, with accumulation of TGs invoking the SREBP pathway. This could potentially involve LXRs via LXR activation of SREBP1c, the signal for which should be evident in the data. Thus, further information on lipid content and LXR/SREBP target genes would be of interest.

We thank the reviewer for this interesting question and we agree that lipidomic analysis would be of interest but, at the present time, very challenging because bulk lipidomics is not very sensitive to compositional changes in small microglial subpopulations such as the DAM-like clusters that typically account for 5-10% of all iMGLs *in vitro*.

We also want to point out that we performed the experiment where we tested whether increased lipid droplets content in BHLHE40/41-DKO and single KO is LXR-dependent. To this end, we treated cells with an LXR antagonist (GSK2033) and we measured the level of lipid droplets by flow cytometry using BODIPY. We found that the level of lipid droplets in KO and DKO is decreased after inhibiting LXR with an antagonist as compared to KO and DKO treated with a vehicle. Interestingly, the level of lipid droplets in KO and DKO treated with LXR antagonists reached the level of lipid droplets in WT suggesting increased lipid droplets content in KO and DKO is mediated by increased LXR (see Supplementary Figure 5 and Page 15).

Although, our BHLHE40/41-KO and DKO lines showed increased LXR activity such as elevated levels of APOE, ABCA1, cholesterol efflux and lipid droplets content, in our RNAseq experiment we did not observe an increased expression of genes involved in "Regulation of cholesterol biosynthesis by SREBP SREBF" (Normalized enrichment score, NES = 0.69 for DKO vs WT

contrast). In addition, based on our gene sets enrichment analysis, we found negative enrichment of pathways involved in “Triglyceride metabolic process” (NES = -1.33) which may suggest that the SREBP axis is not activated in DKO iMGLs. We also compared the induction of lipid catabolism, sterol transport, fatty acid metabolism gene sets/pathways in DKO iMGLs with the transcriptomic effect of synthetic LXR agonist (TO901317, 10uM, 48h, Goate lab unpublished data). We found the LXR agonist has a stronger effect on all aforementioned processes as compared to DKO.

2.2. It is difficult to reconstruct how the authors arrived at the different human and mouse “LAM” marker genes. A list of LAM marker genes is not provided in the manuscript. The number of LAM genes in mouse seems to be very high with 1,453 genes as written in Figure 7B. The authors should clarify this.

We thank the reviewer for this clarifying question. Since the term “LAM” used in various contexts in the previous version of the manuscript seems to be causing confusion, we have decided to use the term DLAMs to collectively refer to subpopulations of microglia and peripheral macrophages such as DAM, LAM, TREM2high among others, which share similar gene expression signatures and cellular responses to damage of lipid-rich tissues. We use the term “LAM” when using lipid-associated macrophages gene signature from Jaitin et al., (Jaitin et al. 2019) Dataset S6, FDR < 0.05. We used this signature specifically in Figure 4A, Supplementary Figure 7a and Figure 3A,C. Which is specified in Supplementary Table 1 and listed in the main text and each figure legend when this gene signature has been used. We use the term “DAM” referring to the mouse DAM signature (not just the top marker genes) profiled by Keren-Shaul et al (Keren-Shaul et al. 2017) Supplementary Table S3 (FDR <0.05). We used it in Figure 7A.

Due to the fact that RRHO analysis requires genes to be ranked by a signed differential gene expression statistic (for example $\log_2FC * \log_{10}(\text{adj.P-value})$), we have used Table S3 from (Keren-Shaul et al. 2017) listing all the DAM genes along with the two necessary parameters (\log_2FC and adj.P-value) for the RRHO analysis. Table S2 from (Keren-Shaul et al. 2017) contains a list of only 500 genes (frequently referred to as “DAM genes”) but does not include the \log_2FC parameter that is necessary to perform RRHO analysis. To investigate whether the DAM genes from Table S2 from (Keren-Shaul et al. 2017) are also positively enriched in mouse DKO transcriptome, we performed GSEA using DAM upregulated genes and DAM downregulated genes from Table S2. We found significant positive enrichment of DAM upregulated genes and significant negative enrichment of DAM downregulated genes in mouse DKO transcriptome. These results are added to Figure 7D and Page 20. Please also see section 1.5 of this document.

2.3. In studies using *Bhlhe40*^{-/-}*Bhlhe41*^{-/-} double knockout mice, the authors should comment on the extent to which *Bhlhe40/41* are expressed by other brain cell types. If they are expressed, the authors should include the possibility that some of the effects of the knockout could be non-cell autonomous in the limitations section. It would also be of interest to know whether lipid droplets were observed in microglia from these mice.

We thank the reviewer for this comment and we have now added the following information in the Study Limitations section: “Another limitation of the study is the mouse model with a global knock-out of *Bhlhe40* and *Bhlhe41* (mouse DKO) which may affect other cell types such as astrocytes where *Bhlhe40* and *Bhlhe41* are also expressed. We cannot exclude that there is a non-cell-autonomous effect mediated possibly by astrocytes lacking *Bhlhe40/41* that would affect DKO microglia” (Page 26). *Bhlhe40* expression is low in microglia/macrophages but it is significantly induced in disease-associated microglia (DAM) (Friedman et al. 2018; Keren-Shaul et al. 2017). *Bhlhe40* is highly expressed in astrocytes and endothelial cells. *Bhlhe41* is highly expressed by myeloid cells (microglia, macrophages, neutrophils) and to a lesser extent by astrocytes (data source: <https://brainrnaseq.org/>).

Although we have not done any experiments checking the level of lipid droplets in mouse microglia lacking Bhlhe40/41, we expect increased lipid droplets content in DKO microglia. There are couple of reasons for that: 1) It has already been shown that alveolar macrophages from DKO mice accumulate lipid droplets Rauschmeier et al., (Rauschmeier et al. 2019) (please see Figure 4, also pasted below) 2) GSEA analysis of transcriptome from microglia lacking Bhlhe40/41 (this study) showed that GOCC_LIPID_DROPLET and GOBP_REGULATION_OF_LIPID_STORAGE are significantly and positively enriched with Normalized Enrichment Score (NES = 1.59 NOM P value = 0.007 and NES = 1.47 NOM P value = 0.047, respectively).

Figure 4C from (Rauschmeier et al. 2019)

GSEA (mouse microglia lacking Bhlhe40/41as compared to WT)

Minor comment:

2.4. RNA-seq data is presented as RRHO heatmaps. It would be informative to present the data also a Volcano plots depicting fold change and adj p-value and to state, what the fold-cut off were to determine differentially expressed genes. This can be shown as supplementary data.

We thank the reviewer for this comment. DLAM genes did not reach statistical significance for differential gene expression when filtered with standard RNA-seq criteria (FDR < 0.05). Therefore, as we did previously with APOE4 microglia TCW et al., (Tcw et al. 2022), we moved away from threshold-based methods toward ranked-based methods (RRHO and - in the revised version - GSEA) that do not rely on arbitrary cutoffs and take full advantage of the information contained in whole-transcriptome profiles. The reason why the effect sizes of DLAM marker genes are small in DKO cells may be due to the fact that typically DLAMs make for only a small proportion of microglia/macrophages both *in vitro* and *in vivo*. Therefore, scRNA-seq may be more sensitive for the detection of gene expression changes associated with compositional shifts in microglia/macrophage subpopulations upon genetic inactivation of BHLHE40/41. We are currently performing such experiments in the context of follow-up studies and future manuscripts.

Reviewer #3 (Remarks to the Author):

Communications – Review March 2023

BHLHE40/41 regulate macrophage/microglia responses associated with Alzheimer's disease and other disorders of lipid-rich tissues

The manuscript by Podlesny-Drabiniok and Novikova et al. outlines the core gene-regulatory-network (GRN) function of the transcription factors; BHLHE40/41, in microglia and macrophages in lipid-rich environments in various diseases. Combining multiple publicly available single-cell and single-nucleus RNA-seq datasets, as well as ATAC-seq, the authors show that Bhlhe40/41 is a core transcriptional regulators of LAM microglia/macrophages and provide evidence of functional responses of microglia upon genetic perturbation of Bhlhe40/41. Overall, the manuscript provides an important insight to the transcriptional impact of Bhlhe40/41 on cholesterol metabolism and lysosomal activity in microglia. However, the initial GRN analysis leaves several questions to be addressed as indicated below:

Major comments:

3.1. The methods for the reconstruction of gene-regulatory-network must be described in more detail. Figure 1 shows the consolidated results for the GRN analysis. It is not clear whether metacells were constructed separately for the different expression matrix files (different datasets) or processed together as one merged expression matrix. If individual networks were built for each expression matrix, the intermediate results should be presented.

We thank the reviewer for this clarifying question. Metacells were generated for every dataset separately and a network was generated for every individual dataset as well. Meta-analysis was

done using all of the networks combined and is shown in Figure 1. We have now edited the methods section to describe the network reconstruction procedure in more detail. We also added intermediate results, TFs nominated by each dataset prior to meta-analysis, to Supplementary Table 1.

3.2. Why were all metacells created using standard parameters. The authors should provide evidence that standard parameters provided the most robust K-means clustering across different datasets.

Parameters for k-means clustering were different for each dataset, they were chosen such as the number of neighbors * the number of metacells was as close to the total number of cells as possible to avoid the same cell being included in multiple metacells. In our conversations with the authors of PISCES, they suggested selecting the number of neighbors > 10 and number of metacells > 100 and that this selection did not dramatically affect the outcome of ARACNE. In our own original analyses with changing the number of neighbors and number of metacells, the list of potential regulators of the LAM signature was not dramatically affected. We now include the parameters that were used for metacells generation for each dataset on our github page. We edited the Methods section to address this question and it now reads:

“The pipeline for reconstruction of metacells was adopted from the PISCES tool (Obradovic, Vlahos, et al. 2021) and can be found at our github page https://github.com/marcoralab/bhlhe_manuscript. Briefly, this approach constructs a kNN graph using the data, partitions the data into an appropriate number of metacells, taking into account the desired number of neighbors. It then aggregates the counts from closest neighbors into MetaCells. The MetaCells function was used in the following manner: *MetaCells(data, dist.mat, numNeighbors = numNeighbors, subSize = subSize)*. *numNeighbors* and *subSize* for each dataset are provided on our github page.”

3.3. Most of the input datasets for GRN analysis are not associated to Alzheimer’s disease and/or microglia. Yet, the authors are using the consolidated results in Figure 1 to justify why BHLHE40/41 is studied in microglia. In silico, the authors need to provide more insight of the GRN analysis in microglia.

We thank the reviewer for this suggestion. In the manuscript, we attempted to focus on both peripheral tissue-resident macrophages and microglia, given that the lipid-associated macrophage response occurs in various disease contexts, such as Alzheimer’s disease, fatty liver disease, obesity, lung fibrosis and others. Hence, some of our datasets are associated with Alzheimer's disease (e.g., human AD brains from Zhou et al and APP knockin mouse data from Sala Frigerio et al) and others are associated with other disorders of lipid-rich tissues (e.g., human liver cirrhosis from Ramachandran et al and mouse induced nonalcoholic steatohepatitis from Xiong et al). Indeed, our downstream validation focuses not only on human and mouse microglia, but also on human macrophages shown in Supplementary Figures 6 and 7, attempting to show that this response is present in different types of macrophages. However, we now include all the TFs nominated by each individual dataset (Supplementary Table 1),

including microglial datasets, so that the interested reader can further examine microglia-specific findings if needed.

3.4. The authors implement the PISCES tool to strengthen the expression profile of low expression genes. The tool is not yet peer-reviewed and it is not clear why the authors only used single-cell and single-nucleus RNA-seq although bulk RNA-seq provides much better sequencing depth.

We thank the reviewer for this thoughtful question. Unfortunately, given the small number of microglia in the brain, ~5-10 %, bulk brain RNA-seq datasets fail to capture microglial activation states. Indeed, there have been many studies that looked at differential expression between AD and control brains, for example, and the findings mostly relate to the increased number of microglia in a disease brain as opposed to capturing specific activation states. Hence, we focused on single-cell and single-nucleus datasets to leverage a higher diversity of transcriptional responses that are captured in those datasets, despite lower sequencing depth. Given our collaboration with the authors, we used a small portion of the PISCES pipeline that deals specifically with metacells reconstruction and obtained the code from the team a long time prior to publication. We now share the code used for metacells reconstruction on our github page https://github.com/marcoralab/bhlhe_manuscript. In addition, this pipeline has been used and published (and some of its findings experimentally validated) in several peer-reviewed articles (Hawley et al. 2023; Pan et al. 2020; Obradovic, Chowdhury, et al. 2021).

3.5. It is not clear to what extent the LAM signature is present in the datasets used. How many cells are positive for the LAM signature, and do they cluster separately from DAM microglia? Where only LAM signature microglia/macrophages implemented for the GRN analysis?

We thank the reviewer for this clarifying question. We would first like to clarify the DAM/LAM nomenclature. In this manuscript, we attempted to communicate that several populations carrying different names, such as disease-associated microglia (DAM) (Keren-Shaul et al. 2017), TREM2^{hi} macrophages identified in atherosclerotic plaques (Cochain et al. 2018) and lipid-associated macrophages (LAM) in adipose tissue (Jaitin et al. 2019) are similar to each other in that they mount similar transcriptional and cellular response to damage of lipid-rich tissues. Using datasets from relevant disease and control tissues, we aimed to identify transcriptional regulators that might be shared between these populations, focusing specifically on disease-associated microglia population (DAM) in the brain, TREM2^{hi} population in atherosclerotic plaques, and lipid-associated macrophages (LAM) in adipose tissue. In the revised manuscript we collectively referred to this microglia/macrophage subpopulations as DLAMs (please see response to comment 1.3). In our computational analyses, we make sure that the TFs we nominate exhibited enrichment of their regulons for all the three DLAM genesets mentioned above. Our question of interest is to identify transcriptional regulators of the DLAM response, so we focused on those signatures in our GRN analysis. By the nature of the datasets we selected, most of them showed positive expression of DLAM markers. Below we show the findings from original authors, when possible, or include UMAPs that we generated ourselves. Starting with mouse datasets, adipose macrophages reported in

Jaitin et al (Jaitin et al. 2019) express LAM markers, such as *Ctsb*, *Gpnmb*, *Lgals3*, *ApoE* and others.

Xiong et al (Xiong et al. 2008) reported macrophages in the liver that responded to the induction of nonalcoholic steatohepatitis that express Trem2 and Gpnmb among other DLAM markers.

Ramachandran et al (Ramachandran et al. 2019) reported scar-associated macrophages in both mouse and humans that highly express DLAM makers, such as TREM2 and CD9, as shown below in their cross-species integrative analysis.

Cochain et al (Cochain et al. 2018) reported macrophages in atherosclerotic aortas that again highly express DLAM markers, such as Trem2, Cd9 and Ctsd among others.

Lin et al (Lin et al. 2019) also identified a similar Trem2^{high} macrophage population in their atherosclerotic model, where these macrophages highly express *Spp1*, *Cd9* and other DLAM markers.

Sala Frigerio et al (Sala Frigerio et al. 2019) uses Alzheimer's disease mouse models that have been previously reported to show DAM activation, expressing *ApoE*, *Spp1*, *Gpnmb* and other LAM markers. The authors term this population Activated response microglia or ARMs and characterize this activation state more deeply than the original DAM study (Keren-Shaul et al. 2017).

Given previous reports that aged mice show an activation state similar to that observed in mouse models of neurodegeneration albeit to a much smaller extent (Keren-Shaul et al. 2017), we included a microglial dataset from aging mouse brains. Please see microglial UMAPs below showing LAM gene expression in Ximerakis et al dataset (Ximerakis et al. 2019); a cluster of cells expressing markers, such as *Gpnmb*, *Lpl* and *Spp1* as well as an increased expression of *Apoe* can be seen.

Given that the DLAM response is, at least in part, TREM2 dependent, we included data from a demyelination mouse model with wild-type and knockout Trem2. Please see microglial UMAPs below showing DLAM gene expression in Nugent et al (Nugent et al. 2020) dataset similarly showing a cluster of cells expressing *Gpnmb*, *Lpl* and *Spp1* as well as an increased expression of *Apoe*.

Human DAM and TREM2^{hi} signatures have not been convincingly described yet. Hence, in our analyses, we lifted 2 mouse signatures to the human genome, the DAM signature (Keren-Shaul et al. 2017) and the TREM2^{hi} signature ((Cochain et al. 2018), along with using a human LAM signature that was derived from human macrophages in obese individuals (Jaitin et al. 2019). We assume that although the mouse and human activation signatures will surely be, at least in part, different, we decided to leverage human datasets from the same disease conditions as our mouse data, e.g. Alzheimer's disease, atherosclerosis and non-alcoholic steatohepatitis. We also used a dataset with a very large number of primary human microglia from fresh resected tissues. Assuming that at least some of the markers are conserved, we subsequently aim to use co-expression patterns in microglial cells to identify regulators of the DLAM signature. Please see UMAP plots showing microglial expression of select DLAM genes from human datasets used in this study.

Mathys et al

Olah et al

Zhou et al

Jaitin et al

Fernandez et al

MacParland et al

Ramachandran et al

3.6. The authors mention efferocytosis as a common hallmark pathway for LAM microglia/macrophages, however, do not show any data supporting this statement.

We thank the reviewer for this comment and we apologize that it was not well explained. We think that efferocytosis, understood as clearance of lipid-rich cellular debris (i.e. myelin fragments, apoptotic cells and synapses, dystrophic neurites, amyloid plaques, etc.), is a main microglial/macrophage process affected by genetic variants associated with AD risk. Efferocytosis is a four step mechanism that includes proper Chemotaxis and Recognition of extracellular waste (step1), Engulfment that requires actin polymerization and cytoskeleton rearrangements to take up extracellular debris (step 2), Digestion that comprises degradation of engulfed material in the endolysosomal system (step 3) and Adaptation that includes activation of transcription factors that increase phagocytosis, cholesterol efflux and storage, lysosomal biogenesis, bioenergetics and other metabolic processes (step 4). AD risk genes are enriched in each of these steps including genes with rare coding variants such as *TREM2*, *ABCA7*, *ABI3*, *PLCG2*, and genes implicated by common non-coding variants (e.g. *ABCA1*, *ZYX*, *BIN1*, *RIN3*, *MEF2C*, *SPI1*). Genes identified through a variety of different approaches including coloc, TWAS, and SMR show that candidate causal genes for AD fall into one of these four steps supporting our hypothesis that abnormal microglial efferocytosis plays an important role in the etiology of AD. We have presented this hypothesis with supporting genetic and experimental evidence in three reviews (Romero-Molina et al. 2022; Andrews et al. 2023; Podleśny-Drabiniok, Marcora, and Goate 2020). Consistent with this hypothesis is the

observation that several DLAM genes are AD risk genes and that DLAM genes are enriched in several pathways (for example phagocytosis and lipid metabolism) that are core components of efferocytosis (Deczkowska et al., (Deczkowska et al. 2018)). Interestingly, from the perspective of AD evolution, primate microglia (compared to rodent microglia) are also enriched for AD risk and efferocytosis genes (Geirsdottir et al., (Geirsdottir et al. 2020)).

3.7. When generating the short list of candidate transcription factors, the researchers require the transcription factor to appear in at least half the human and mouse networks and present in human microglia. The authors need to provide data visualization displaying which networks contain these transcription factors.

We have now included results from individual network enrichment analyses in Supplementary Table 1, where the reader can see which TF was nominated by which study/network.

3.8. The reason for the expression-based selection of top transcription factor markers is not clear. Protein levels of transcription factors should be considered. Even low expressed transcription factors could have biological and functional meaning. Selecting only the top 11 candidates before investigating AD risk allele enrichment and TF binding sites could skew the results and hide other potentially important transcription factors. The authors must present a less biased approach.

We absolutely agree with the reviewer that protein levels of transcription factors are important; however, in this study we focus on single-cell and single-nucleus RNA-seq data because the population we are most interested in (lipid-associated macrophages) was identified at the mRNA level. The expression filter that we implemented is very relaxed; we only require a TF to be expressed at a level ≥ 1 TPM in human microglia; in fact, when examining the list of TFs pre and post-filter, all 11 TFs are expressed in human microglia ≥ 1 TPM. Given this observation, we removed the description of the filter from the manuscript. The text now reads: "High confidence TFs were selected if they were 1) enriched for all three LAM genesets in at least half of human and mouse networks and 2) conserved between species."

Since microglia represent a small fraction of brain-resident cells and proteins from more abundant cells are much more likely to be captured, we avoided the use of whole brain proteomics datasets. Hence, we used datasets, where microglia were purified prior to the proteomics experiment. Unfortunately, although there are multiple mouse microglial proteomics datasets published, the number of detected proteins in microglia is still quite low. For example, we examined a dataset by Rangaraju et al (Rangaraju, Dammer, Raza, Gao, et al. 2018), where the authors performed a TMT proteomics on isolated Cd11b+ microglial cells from wild-type, LPS-stimulated and Alzheimer's disease mouse models (5xFAD). The authors detected 4,133 proteins across the three experimental groups, but none of our 11 top transcriptional regulator candidates were detected. Out of 74 mouse TFs reported in Figure 1, only 11 were detected in the dataset. Human microglia proteomics datasets are therefore, unfortunately, limited at this time. A recent preprint by Lloyd et al (Lloyd et al. 2022) reported more than 9,000 microglial proteins, but the paper has not been peer reviewed and the data are not yet available. Taken

together, given a relatively small number of proteins that are detected in current proteomics experiments, a filter on proteomic expression is not optimal and expression of many transcriptional regulators cannot yet be assessed.

3.9. The significance values of enrichment for BHLHE40 binding to ATAC seq peaks for LAM genes is missing. The authors should demonstrate more transparency and show HOMER results for Figure 3.”

We would like to clarify the data shown in Figure 3. We are not performing an enrichment analysis in Figure 3. Instead, we are taking 11 TFs that we are nominating through co-expression analysis and quantifying the number of LAM genes. We used LAM genes from (Jaitin et al. 2019) Dataset S6, FDR Adj.P-value < 0.05) that contain a motif of that TF in their promoter (Figure 3A, C) and we used DAM genes from (Keren-Shaul et al. 2017) Table S2 . Hence, we used HOMER only to pinpoint ATAC-seq regions that are positive for the motif of interest; we have not performed a global motif enrichment analysis because it does not address the question we are interested in. With the analysis presented in Figure 3, we are trying to assess which of the TFs that are nominated through co-expression could also potentially bind LAM genes. Additionally, we are not showing a percent of LAM gene promoters that contain a BHLHE40 motif; BHLHE41 is the TF that is nominated through the network approach.

3.10 There are several datasets on BHLHE40 ChIP-seq, which should be investigated and used to validate potential BHLHE40 binding site enrichment at LAM genes (Citation: 59, PMID: 31061528).

We thank the review for this suggestion. We would like to highlight that our network analyses nominate BHLHE41 in particular, not BHLHE40. Our analyses of open chromatin regions in DLAM gene promoters presented in Figure 3 also highlights that a large proportion of LAM genes contain a BHLHE41 motif (not BHLHE40). Although these TFs are closely related and demonstrated a level of compensatory activity in previous studies, we show individual and double KD/KO in our validation studies. However, our computational analyses suggest that BHLHE41 could have different/broader binding patterns than BHLHE40. In our analyses of human microglial open chromatin regions, we identified around 30K BHLHE41 proxy-binding sites compared to only ~8.7K BHLHE40 proxy-binding sites. Indeed, more than 70% of LAM genes contain a BHLHE41 motif (Figure 1), while only 28% contain a BHLHE40 motif. This suggests that there may be fewer BHLHE40 binding sites around the genome. Unfortunately, there are no studies to date that have profiled open chromatin regions and Bhlhe40/Bhlhe41 binding pattern, making it difficult for us to validate our proxy-binding sites for these TFs. However, we analyzed two additional datasets in support of our observation that Bhlhe40 has more limited binding throughout the genome than Bhlhe41. We analyzed ChIP-seq data from large peritoneal mouse macrophages (Rauschmeier et al. 2019) and identified only 1,046 *Bhlhe40* peaks with 6% of DAM genes having a Bhlhe40 binding site in their promoter (as a DAM we used (Keren-Shaul et al. 2017), Table S2). We also analyzed ATAC-seq data from mouse bone-marrow derived macrophages (Daniel et al. 2020), which showed a similar pattern with only 3,854 Bhlhe40 proxy-binding sites out of ~77K open chromatin regions with 17% of DAM genes having a Bhlhe40 binding site in their promoter (in comparison, Bhlhe41

proxy-binding sites were present in around 40% of DAM genes in mouse BMDMs as shown in Figure 3) (as a DAM we used (Keren-Shaul et al. 2017), Table S2). This observation could be driven by the lower expression of BHLHE40; for example, in human microglia, BHLHE41 is expressed almost 14 times higher than BHLHE40 (Gosselin et al. 2017). Interestingly, Bhlhe40 is upregulated in mouse DAM signature, suggesting that detecting binding patterns of Bhlhe40 might be harder in baseline tissues. Taken together, we would like to highlight that BHLHE41 has been nominated by our network analyses and its motif is contained in a large proportion of DLAM gene promoters. Although BHLHE40 is a closely related TF, its potential binding patterns do seem to differ from BHLHE41 and they do not suggest potential direct binding to DLAM gene promoters. We would, however, like to highlight the limitations of using motifs as opposed to CHIP-seq data to ascertain TF binding sites.

Minor comments:

3.11. It is not clear how the Human and mouse LAM TFs in Figure 1 were clustered. Were the LAM TFs ranked, and if so, using what method?

Heatmap was created using the R pheatmap package with clustering_method = "complete".

3.12. The authors should provide all code used for the GRN analysis for more transparency.

We have now created a github page https://github.com/marcoralab/bhlhe_manuscript where an example of GRN generation is included to enhance transparency.

3.13. The authors should comment on the low MI scores between LAM genes and TFs in mice in Figure 1.

There could be many potential reasons why some MI scores are low between a subset of DLAM genes and TFs in mouse datasets. One could be because some of the human datasets are quite large (Olah et al., n.d.; Mancuso et al. 2022), providing more microglial cells and allowing for detection of weaker co-expression patterns between DLAM genes and TFs. Another reason could be that the mouse and human DLAM signatures depicted in Figure 1, although significantly overlapping, are distinct, which can also drive differences observed in Figure 1.

3.14. Peak profiles of ATAC-seq for Bhlhe40 motif enrichment sites at LAM genes should be presented, as well as CHIP of Bhlhe40.

We thank the reviewer for this suggestion. We would like to highlight that our network analyses nominate BHLHE41, not BHLHE40. Although these TFs are closely related and demonstrate a level of compensatory mechanisms in certain contexts, our analyses nominated BHLHE41 in particular through co-expression followed by quantification of proxy-binding at LAM gene promoters. As described in our response to comment 3.10, BHLHE40 likely has different binding patterns than BHLHE41 and proxy-binds a much smaller proportion of LAM gene promoters than BHLHE41. Since BHLHE41 ChIP-seq data in human macrophages have not been generated (likely due to the fact that a quality antibody for BHLHE41 is not commercially available), we now include epigenomic tracks, highlighting open chromatin regions in LAM gene promoters in human and mouse microglia and macrophages that contain a BHLHE41 motif in Figure 3E-F.

3.15. Why did the authors choose to select LPS and IL4 treated microglia?

We have not chosen IL4 or LPS-treated microglia but we cited one article that used 43 existing GEO microarray transcriptomes of Cd11b+ microglia including in vivo microglia from AD mouse models and in vitro microglia stimulated with LPS and IL4 (Rangaraju, Dammer, Raza, Rathakrishnan, et al. 2018). Since this created an impression that we included only LPS and IL4 microglia signatures, we re-worded that sentence, please see Introduction, paragraph started with “The DLAM response like other ...” (Page 4)

3.16. Using stratified LD score regression to perform AD heritability analysis it is not clear if the P-values are adjusted for false discovery?

The P-values shown in Figure 2 are not FDR adjusted; however, dark blue bars indicate significant enrichments (FDR < 0.05), bars in light red indicate nominally significant enrichments (P-value < 0.05), while gray bars indicate non-significant enrichment.

- Andrews, Shea J., Alan E. Renton, Brian Fulton-Howard, Anna Podlesny-Drabiniok, Edoardo Marcora, and Alison M. Goate. 2023. “The Complex Genetic Architecture of Alzheimer’s Disease: Novel Insights and Future Directions.” *EBioMedicine* 90 (April): 104511.
- Claes, Christel, Emma Pascal Danhash, Jonathan Hasselmann, Jean Paul Chadarevian, Sepideh Kiani Shabestari, Whitney E. England, Tau En Lim, et al. 2021. “Plaque-Associated Human Microglia Accumulate Lipid Droplets in a Chimeric Model of Alzheimer’s Disease.” *Molecular Neurodegeneration*. <https://doi.org/10.1186/s13024-021-00473-0>.
- Cochain, Clément, Ehsan Vafadarnejad, Panagiota Arampatzi, Jaroslav Pelisek, Holger Winkels, Klaus Ley, Dennis Wolf, Antoine-Emmanuel Saliba, and Alma Zerneck. 2018. “Single-Cell RNA-Seq Reveals the Transcriptional Landscape and Heterogeneity of Aortic Macrophages in Murine Atherosclerosis.” *Circulation Research* 122 (12): 1661–74.
- Daniel, Bence, Zsolt Czimmerer, Laszlo Halasz, Pal Boto, Zsuzsanna Kolostyak, Szilard Poliska, Wilhelm K. Berger, et al. 2020. “The Transcription Factor EGR2 Is the Molecular Linchpin Connecting STAT6 Activation to the Late, Stable Epigenomic Program of Alternative Macrophage Polarization.” *Genes & Development* 34 (21-22): 1474–92.
- Deczkowska, Aleksandra, Hadas Keren-Shaul, Assaf Weiner, Marco Colonna, Michal Schwartz, and Ido Amit. 2018. “Disease-Associated Microglia: A Universal Immune Sensor of Neurodegeneration.” *Cell* 173 (5): 1073–81.
- Dolan, Michael-John, Martine Therrien, Saša Jereb, Tushar Kamath, Vahid Gazestani, Trevor Atkeson, Samuel E. Marsh, et al. 2023. “Exposure of iPSC-Derived Human Microglia to Brain Substrates Enables the Generation and Manipulation of Diverse Transcriptional States in Vitro.” *Nature Immunology* 24 (8): 1382–90.
- Fernandez, Dawn M., Adeeb H. Rahman, Nicolas F. Fernandez, Aleksey Chudnovskiy, El-Ad David Amir, Letizia Amadori, Nayaab S. Khan, et al. 2019. “Single-Cell Immune Landscape of Human Atherosclerotic Plaques.” *Nature Medicine* 25 (10): 1576–88.
- Friedman, Brad A., Karpagam Srinivasan, Gai Ayalon, William J. Meilandt, Han Lin, Melanie A. Huntley, Yi Cao, et al. 2018. “Diverse Brain Myeloid Expression Profiles Reveal Distinct Microglial Activation States and Aspects of Alzheimer’s Disease Not Evident in Mouse Models.” *Cell Reports* 22 (3): 832–47.
- Geirsdottir, Laufey, Eyal David, Hadas Keren-Shaul, Assaf Weiner, Stefan Cornelius Bohlen, Jana Neuber, Adam Balic, et al. 2020. “Cross-Species Single-Cell Analysis Reveals Divergence of the Primate Microglia Program.” *Cell* 181 (3): 746.
- Gerrits, Emma, Nieske Brouwer, Susanne M. Kooistra, Maya E. Woodbury, Yannick Vermeiren,

- Mirjam Lambourne, Jan Mulder, et al. 2021. "Distinct Amyloid- β and Tau-Associated Microglia Profiles in Alzheimer's Disease." *Acta Neuropathologica* 141 (5): 681–96.
- Gosselin, David, Dylan Skola, Nicole G. Coufal, Inge R. Holtman, Johannes C. M. Schlachetzki, Eniko Sajti, Baptiste N. Jaeger, et al. 2017. "An Environment-Dependent Transcriptional Network Specifies Human Microglia Identity." *Science* 356 (6344).
<https://doi.org/10.1126/science.aal3222>.
- Hasselmann, Jonathan, Morgan A. Coburn, Whitney England, Dario X. Figueroa Velez, Sepideh Kiani Shabestari, Christina H. Tu, Amanda McQuade, et al. 2019. "Development of a Chimeric Model to Study and Manipulate Human Microglia In Vivo." *Neuron* 103 (6): 1016–33.e10.
- Hawley, Jessica E., Aleksandar Z. Obradovic, Matthew C. Dallos, Emerson A. Lim, Karie Runcie, Casey R. Ager, James McKiernan, et al. 2023. "Anti-PD-1 Immunotherapy with Androgen Deprivation Therapy Induces Robust Immune Infiltration in Metastatic Castration-Sensitive Prostate Cancer." *Cancer Cell* 41 (11): 1972–88.e5.
- Jaitin, Diego Adhemar, Lorenz Adlung, Christoph A. Thaiss, Assaf Weiner, Baoguo Li, H el ene Descamps, Patrick Lundgren, et al. 2019. "Lipid-Associated Macrophages Control Metabolic Homeostasis in a Trem2-Dependent Manner." *Cell* 178 (3): 686–98.e14.
- Kamphuis, Willem, Lieneke Kooijman, Sjoerd Schettters, Marie Orre, and Elly M. Hol. 2016. "Transcriptional Profiling of CD11c-Positive Microglia Accumulating around Amyloid Plaques in a Mouse Model for Alzheimer's Disease." *Biochimica et Biophysica Acta* 1862 (10): 1847–60.
- Keren-Shaul, Hadas, Amit Spinrad, Assaf Weiner, Orit Matcovitch-Natan, Raz Dvir-Szternfeld, Tyler K. Ulland, Eyal David, et al. 2017. "A Unique Microglia Type Associated with Restricting Development of Alzheimer's Disease." *Cell* 169 (7): 1276–90.e17.
- Kunkle, Brian W., Michael Schmidt, Hans-Ulrich Klein, Adam C. Naj, Kara L. Hamilton-Nelson, Eric B. Larson, Denis A. Evans, et al. 2021. "Novel Alzheimer Disease Risk Loci and Pathways in African American Individuals Using the African Genome Resources Panel: A Meta-Analysis." *JAMA Neurology* 78 (1): 102–13.
- Lin, Jian-Da, Hitoo Nishi, Jordan Poles, Xiang Niu, Caroline Mccauley, Karishma Rahman, Emily J. Brown, et al. 2019. "Single-Cell Analysis of Fate-Mapped Macrophages Reveals Heterogeneity, Including Stem-like Properties, during Atherosclerosis Progression and Regression." *JCI Insight* 4 (4). <https://doi.org/10.1172/jci.insight.124574>.
- Lloyd, Amy F., Anna Martinez-Muriana, Pengfei Hou, Emma Davis, Renzo Mancuso, Alejandro J. Brenes, Ivana Geric, et al. 2022. "Deep Proteomic Analysis of Human Microglia and Model Systems Reveal Fundamental Biological Differences of in Vitro and Ex Vivo Cells." *bioRxiv*. <https://doi.org/10.1101/2022.07.07.498804>.
- Mancuso, Renzo, Nicola Fattorelli, Anna Martinez-Muriana, Emma Davis, Leen Wolfs, Johanna Van Den Daele, Ivana Geric, et al. 2022. "A Multi-Pronged Human Microglia Response to Alzheimer's Disease A β Pathology." *bioRxiv*. <https://doi.org/10.1101/2022.07.07.499139>.
- Marschallinger, Julia, Tal Iram, Macy Zardeneta, Song E. Lee, Benoit Lehallier, Michael S. Haney, John V. Pluvinage, et al. 2020. "Lipid-Droplet-Accumulating Microglia Represent a Dysfunctional and Proinflammatory State in the Aging Brain." *Nature Neuroscience* 23 (2): 194–208.
- McQuade, Amanda, Morgan Coburn, Christina H. Tu, Jonathan Hasselmann, Hayk Davtyan, and Mathew Blurton-Jones. 2018. "Development and Validation of a Simplified Method to Generate Human Microglia from Pluripotent Stem Cells." *Molecular Neurodegeneration* 13 (1): 67.
- Novikova, Gloriia, Manav Kapoor, Julia Tcw, Edsel M. Abud, Anastasia G. Efthymiou, Steven X. Chen, Haoxiang Cheng, et al. 2021. "Integration of Alzheimer's Disease Genetics and Myeloid Genomics Identifies Disease Risk Regulatory Elements and Genes." *Nature Communications* 12 (1): 1610.

- Nugent, Alicia A., Karin Lin, Bettina van Lengerich, Steve Lianoglou, Laralynne Przybyla, Sonnet S. Davis, Ceyda Llapashtica, et al. 2020. "TREM2 Regulates Microglial Cholesterol Metabolism upon Chronic Phagocytic Challenge." *Neuron* 105 (5): 837–54.e9.
- Obradovic, Aleksandar, Nivedita Chowdhury, Scott M. Haake, Casey Ager, Vinson Wang, Lukas Vlahos, Xinzheng V. Guo, et al. 2021. "Single-Cell Protein Activity Analysis Identifies Recurrence-Associated Renal Tumor Macrophages." *Cell* 184 (11): 2988–3005.e16.
- Obradovic, Aleksandar, Lukas Vlahos, Pasquale Laise, Jeremy Worley, Xiangtian Tan, Alec Wang, and Andrea Califano. 2021. "PISCES: A Pipeline for the Systematic, Protein Activity-Based Analysis of Single Cell RNA Sequencing Data." *bioRxiv*. <https://doi.org/10.1101/2021.05.20.445002>.
- Olah, Marta, Vilas Menon, Naomi Habib, Mariko Taga, Christina Yung, Maria Cimpean, Anthony Khairalla, et al. n.d. "A Single Cell-Based Atlas of Human Microglial States Reveals Associations with Neurological Disorders and Histopathological Features of the Aging Brain." <https://doi.org/10.1101/343780>.
- Pan, Huize, Chenyi Xue, Benjamin J. Auerbach, Jiabin Fan, Alexander C. Bashore, Jian Cui, Dina Y. Yang, et al. 2020. "Single-Cell Genomics Reveals a Novel Cell State During Smooth Muscle Cell Phenotypic Switching and Potential Therapeutic Targets for Atherosclerosis in Mouse and Human." *Circulation* 142 (21): 2060–75.
- Pimenova, Anna A., Manon Herbinet, Ishaan Gupta, Saima I. Machlovi, Kathryn R. Bowles, Edoardo Marcora, and Alison M. Goate. 2021. "Alzheimer's-Associated PU.1 Expression Levels Regulate Microglial Inflammatory Response." *Neurobiology of Disease* 148 (January): 105217.
- Podleśny-Drabiniok, Anna, Edoardo Marcora, and Alison M. Goate. 2020. "Microglial Phagocytosis: A Disease-Associated Process Emerging from Alzheimer's Disease Genetics." *Trends in Neurosciences* 43 (12): 965–79.
- Ramachandran, P., R. Dobie, J. R. Wilson-Kanamori, E. F. Dora, B. E. P. Henderson, N. T. Luu, J. R. Portman, et al. 2019. "Resolving the Fibrotic Niche of Human Liver Cirrhosis at Single-Cell Level." *Nature* 575 (7783): 512–18.
- Rangaraju, Srikant, Eric B. Dammer, Syed Ali Raza, Tianwen Gao, Hailian Xiao, Ranjita Betarbet, Duc M. Duong, et al. 2018. "Quantitative Proteomics of Acutely-Isolated Mouse Microglia Identifies Novel Immune Alzheimer's Disease-Related Proteins." *Molecular Neurodegeneration* 13 (1): 34.
- Rangaraju, Srikant, Eric B. Dammer, Syed Ali Raza, Priyadharshini Rathakrishnan, Hailian Xiao, Tianwen Gao, Duc M. Duong, et al. 2018. "Identification and Therapeutic Modulation of a pro-Inflammatory Subset of Disease-Associated-Microglia in Alzheimer's Disease." *Molecular Neurodegeneration* 13 (1): 24.
- Rauschmeier, René, Charlotte Gustafsson, Annika Reinhardt, Noelia A-Gonzalez, Luigi Tortola, Dilay Cansever, Sethuraman Subramanian, et al. 2019. "Bhlhe40 and Bhlhe41 Transcription Factors Regulate Alveolar Macrophage Self-Renewal and Identity." *The EMBO Journal*, August, e101233.
- Romero-Molina, Carmen, Francesca Garretti, Shea J. Andrews, Edoardo Marcora, and Alison M. Goate. 2022. "Microglial Efferocytosis: Diving into the Alzheimer's Disease Gene Pool." *Neuron* 110 (21): 3513–33.
- Sala Frigerio, Carlo, Leen Wolfs, Nicola Fattorelli, Nicola Thrupp, Iryna Voytyuk, Inga Schmidt, Renzo Mancuso, et al. 2019. "The Major Risk Factors for Alzheimer's Disease: Age, Sex, and Genes Modulate the Microglia Response to A β Plaques." *Cell Reports* 27 (4): 1293–1306.e6.
- Tcw, Julia, Lu Qian, Nina H. Pipalia, Michael J. Chao, Shuang A. Liang, Yang Shi, Bharat R. Jain, et al. 2022. "Cholesterol and Matrisome Pathways Dysregulated in Astrocytes and Microglia." *Cell* 185 (13): 2213–33.e25.
- Ximerakis, Methodios, Scott L. Lipnick, Brendan T. Innes, Sean K. Simmons, Xian Adiconis,

- Danielle Dionne, Brittany A. Mayweather, et al. 2019. "Single-Cell Transcriptomic Profiling of the Aging Mouse Brain." *Nature Neuroscience* 22 (10): 1696–1708.
- Xiong, Shigang, Hongyun She, An-Sheng Zhang, Jiaohong Wang, Hasmik Mkrtchyan, Alla Dynnyk, Victor R. Gordeuk, Samuel W. French, Caroline A. Enns, and Hidekazu Tsukamoto. 2008. "Hepatic Macrophage Iron Aggravates Experimental Alcoholic Steatohepatitis." *American Journal of Physiology. Gastrointestinal and Liver Physiology* 295 (3): G512–21.

REVIEWERS' COMMENTS

Reviewer #1 (Remarks to the Author):

The authors have done a tremendous job of objectively, carefully, and exhaustively responding to this Reviewer's comments. They gave extensive responses to my questions/concerns, have revised the manuscript accordingly, and have added important new data and clarifications where needed. I thoroughly enjoyed reading this manuscript and would like to commend the authors on a beautiful, impactful study.

Reviewer #2 (Remarks to the Author):

The authors have satisfactorily addressed the major and minor concerns that I raised in the initial review. The major conclusions of the manuscript are reasonably well supported and the findings represent a significant contribution to defining genes and pathways regulating lipid metabolism and lysosomal function in microglia that are relevant to neurodegenerative diseases.

Reviewer #3 (Remarks to the Author):

The authors fully addressed majority of my main critiques. There are few remaining minor comments to be addressed before publication as follow:

1. Krasemann et al., (PMID: 28930663) identified and described Bhlhe40, which is regulated by APOE signaling, and specifically induced in plaque associated Clec7a+ microglia isolated from AD mice. However, Bhlhe41 was not affected. The authors used double-KO approach. This should be acknowledged and discussed.
2. On P23 the authors describe: "Furthermore, a recent study showed that APOE risk-increasing (APOE $\epsilon 4/\epsilon 4$, similar to APOE, TREM2, and PLCG2 loss-of-function mutations) and risk-decreasing (APOE $\epsilon 2/\epsilon 2$) genotypes are associated with decreased and increased DAM transcriptional ...[68]. The authors should discuss two comprehensive new studies recently published Nature Immunology, which employed complementary gain-of-function and loss-of-function approaches to provide critical new evidence that APOE4 impairs MGnD response to neurodegeneration including identification of the mechanism related to induction of TGFb-mediated checkpoints which block MGnD response, including induction of SPI1 (PMIDs: 37749326, 37857825).

REVIEWER COMMENTS

Reviewer #1 (Remarks to the Author):

This manuscript by Podlesny-Drabiniok, Novikova, and colleagues leverages existing snRNAseq datasets from various disease states involving lipid-accumulating macrophages or microglia in order to identify transcriptional regulators of lipid metabolism. In addition to several other TFs, the authors identify BHLHE40/41 as a primary regulator of the 'lipid-associated macrophage' ("LAM") response. They then use gene editing in human iPSC derived microglia to knockout BHLHE40 and/or 41, as well as siRNA knockdown in THP-1 human macrophages, and finally acutely isolated microglia from knockout mice, to functionally assess the effects of 40/41 manipulation in macrophages and microglia. There results show that across these model systems, loss (or significant reduction) of 40/41 expression leads to increased expression of "LAM" genes, and increase in lipid storage in the form of lipid droplets, an increase in lysosomal mass and decreased pH, a reduction in cytokine production, and an increase in cholesterol efflux. A primary weakness of the study is a lack of clarity and justification in how the "LAM" profile was defined based on previous studies. Nonetheless, the study has several strengths, including the use of multiple model systems for validation and phenotyping experiments, and it's focus on a novel and little studied AD GWAS gene. Conceptually, this manuscript is exciting and important, as it nicely ties together several important disease-relevant themes in efferocytosis, GWAS risk genes, and macrophage/microglia function in disorders of lipid rich tissue, with a special emphasis and importance for Alzheimer's disease.

Some specific comments, concerns, and questions are listed below:

1.1. The abstract and introduction conflate macrophages and microglia in several instances, several of which are worded in a way that is not entirely accurate. The authors should be clear and consistent in their phrasing (i.e. using either macrophage or microglia, or / when talking about both simultaneously).

We thank reviewers for this comment. We have revised the text accordingly.

1.2. The text in the graphical abstract is too small in almost all areas. It also was not entirely clear how the "LAMs" and "sc/nRNAseq" sections were different or not. Several figures also have text that is too small to read – Figure 1 is a prime example, but there are many others.

We apologize for the inconvenience. We would like to clarify that the figure miniatures at the bottom of the submitted manuscript were placed there for convenience; high resolution images were also uploaded separately. We made sure to provide a high resolution image to reveal the details included in each figure for the resubmission. We have also modified the graphical abstract and other figures to make the text more legible when printed out at the final size and resolution.

1.3. Previously used acronyms for microglial states, of which there are now too many to keep track of, are based on defined gene signatures (a list of genes). The authors refer to LAM genes or LAM signature genes multiple times, but unless the reviewer missed it, a list of genes that constitute LAM genes is never defined or provided. Selected lipid/lysosomal genes such as those

in Figure 6 for example, are logical candidates, but with the focus of the manuscript being the LAM 'state', this concept must be defined much more clearly.

We would like to clarify that the term "LAM" (lipid-associated macrophages) was used to collectively refer to specific microglial/peripheral macrophages subpopulations identified in 5xFAD mice, AD brains, adipose tissue, atherosclerotic plaques etc. It does not indicate a new name for a new subpopulation. The reviewer is right that there are several different acronyms for microglia/peripheral macrophage states/subpopulations such as DAM, LAM, TREM2^{high} among others, which share similar gene signatures and cellular responses to damage of lipid-rich tissues. Therefore, in the previous version of this manuscript, we decided to collectively refer to these microglial/peripheral macrophages subpopulations as LAM (for lipid associated macrophages). However, we understand that this collective name "LAM" may cause confusions because LAM is also a specific gene signature of adipose tissue macrophages identified by Jaitin et al., 2019. In the revised version of the manuscript we decided to use the term DLAMs to collectively refer to these microglial/peripheral macrophages subpopulations. This is also explained in the Introduction: "For brevity in this manuscript, we collectively refer to these subpopulations of microglia and peripheral macrophages as DLAMs" (page 3).

To compare gene signatures from BHLHE40 and/or BHLHE41 knockout iMGLs and THP-1 macrophages and Bhlhe40/41-DKO mice using RRHO, we used specific gene signatures. For human data, we used LAM signature published by Jaitin et al., 2019 (Dataset S6, Adj.P-value < 0.05)(Jaitin et al. 2019), for mouse data, we used DAM signature from Keren-Shaul et al., 2017 (Table S3, Adj.P-value 0.05) (Keren-Shaul et al. 2017). We referenced each specific gene signature in the legend, main text, and list the genes in each gene signature in the Supplementary Table 1. Additionally, in this version of the manuscript we also included other disease-relevant genesets to test their enrichment in human and mouse DKO transcriptome, please see Response 1.5 below.

1.4. Two of the three human datasets used for TF validation are mouse datasets that are simply translated to their respective human homologs. Thus, it doesn't seem appropriate to refer to these as distinct lists or as a 'human' dataset per se. In figures employing the RRHO, the LAM dataset is simply the Keren-Shaul gene list. Thus, it's the DAM gene list renamed LAM. Similar to the points above, the justification is unclear here as is the definition of 'LAM' in this sense.

We apologize that it was unclear but in the figures showing RRHO results obtained from human cells (iMGL in Figure 4 and THP-1 macrophages in Supplementary Figure 7) we used the human LAM signature from (Jaitin et al. 2019) Dataset S6, Adj.P-value < 0.05. On the other hand, in the figures showing RRHO results obtained from mouse microglia (Figure 7) we used the mouse DAM signature from (Keren-Shaul et al. 2017) Table S3 Adj.P-value < 0.05). We have now changed the name to "lipid-associated macrophages" and "disease-associated microglia", we also reference the Supplementary Table 1 in which these gene lists are included in addition to the main text and the figure legends.

1.5. The datasets from Marschallinger (LDAM) and Claes (plaque associated LD) do not appear to be included in the LAM lists the authors include. These seem like highly relevant studies given the focus; was there a reason they were not utilized? These references (57 and 60) should also

probably be mentioned much earlier in the manuscript given their relevance. The manuscript states that these microglia are “distinct from LAMs” – what does this mean exactly, and how is this concluded?

We thank the reviewer for the comment.

Marschallinger et al provided an important and in-depth characterization of lipid droplet-accumulating microglia (LDAMs) in the aging brain. However, these cells appear to be distinct from those observed in DLAMs. More specifically, DLAMs are enriched in phagocytosis and lipid metabolism genes, suggesting enhancement of these processes, while (Marschallinger et al. 2020) suggest that LDAMs exhibit phagocytosis deficits. At the transcriptional level, DLAMs express canonical markers, such as *SPP1*, *LPL*, *APOE*, *CD9*, *AXL*, and *CLEC7A*. Marschallinger et al conducted a comparative analysis to DAM/MGnD microglia and concluded that LDAMs are a distinct population, with some important marker genes that were downregulated in LDAMs being upregulated in DAM/MGnD microglia (e.g., *AXL*, *CLEC7A*). The authors conclude that LDAMs show a “unique transcriptome signature that is distinct from previously described microglia states observed in aging and neurodegeneration.” Hence, we did not include this gene expression signature in our analyses because a) the signature is largely distinct from the DLAM genesets we set out to study and b) the data provided in the study are bulk RNA-seq generated on isolated CD11b⁺CD45^{low} cells, while we solely focus on single cell and single nucleus RNA-seq data in this study. In addition, we have performed RRHO analysis comparing the entire LDAM and mouse DKO transcriptomes and we found no significant correlation (Supplementary Figure 9B).

(Claes et al. 2021) have reported exciting findings using xenografted iPSC-derived microglia (xMGLs), suggesting that human microglia response to beta-amyloid plaques could be similar to that of foam cells in atherosclerotic plaques. However, a previous study by the same group (Hasselmann et al. 2019) showed that xMGLs demonstrate “limited overlap existing between human xMG and mouse DAM genes”. Altogether this suggests that xMGLs acquire a foam cell-like signature and accumulate lipid droplets but they do not closely resemble the DAM phenotype observed in mouse 5xFAD brains.

Nevertheless, we understand the value of including additional gene signatures that may be relevant to our study. To this end, we performed gene set enrichment analysis using the following human genesets: DAM clusters from iMGLs exposed to CNS-relevant phagocytic substrate (Dolan et al. 2023); amyloid plaque-associated microglia from human brains (Gerrits et al. 2021); lipid associated macrophages from adipose tissue (Jaitin et al. 2019); human foam cell signature (Fernandez et al. 2019); xenografted iPSC-derived human microglia from 5xFAD mouse brains (Claes et al. 2021). We split each gene set into up- and downregulated genes and examined their enrichment in the transcriptome of iMGLs lacking BHLHE40 and BHLHE41 (DKO) because DKO iMGLs recapitulated most of the functional aspects of DLAMs. The results are presented in Figure 4D. Briefly, we found significant positive enrichment of genes upregulated in DAM, LAM, and foam cell signatures and negative enrichment of genes downregulated in DAM and LAM clusters (Page 12-13).

We observed a highly positive enrichment of DAM markers reported by (Dolan et al. 2023) in the DKO transcriptome which prompted us to use RRHO to support GSEA analysis (similar to what we did for lipid-associated macrophages, see Figure 4A). As expected, we found that Cluster 2 DAM markers and Cluster 8 DAM markers are highly correlated with the DKO transcriptome suggesting that reduction of BHLHE40/41 levels facilitates transition of iMGLs toward a DLAM-like phenotype (Supplementary Figure 9A). Of note, (Dolan et al. 2023) used highly lipid-rich substrates such as myelin, synaptosomes and apoptotic neurons supporting our observation that exposure to lipid overload activates the DLAM clearance program similar to the one observed in peripheral macrophages in proximity of adipose tissue or atherosclerotic plaques.

Using GSEA we also found a negative enrichment of upregulated genes upregulated in disease-associated xMGLs and a positive enrichment of gene downregulated in disease-associated xMGLs suggesting the opposite regulation of xMGL DAM genes in our BHLH40/41 DKO iMGLs. As stated above it might be due to the fact that the DAM population in xMGL is different from the DAM reported by (Keren-Shaul et al. 2017) from 5xFAD brains. Here we also show that xMGL DAM is different from DAM reported by (Dolan et al. 2023) We performed RRHO and found no overlap between xMGL DAM (by (Claes et al. 2021)) and DAM reported by (Dolan et al. 2023) (Cluster 2) supporting the statement that xMGL DAM is a distinct population.

Next, we performed gene set enrichment analysis using the following mouse gene sets: disease associated microglia (DAM) from 5xFAD mouse brains (Keren-Shaul et al. 2017); CD11c-positive microglia sorted from 5xFAD mouse brains (Kamphuis et al. 2016); activated-response microglia (ARM) and Homeostatic microglia (Sala Frigerio et al. 2019); lipid-associated macrophages (LAM) from visceral adipose tissue of obese mice (Jaitin et al. 2019); Trem2^{high} atherosclerotic macrophages (Cochain et al. 2018); neurodegeneration module (Friedman et al. 2018). We found positive enrichment of upregulated genes in almost all gene sets and negative enrichment of downregulated (or Homeostatic upregulated) genes in the transcriptome of mouse microglia

lacking Bhlhe40 and Bhlhe41 (DKO) (Page 20). We included these analyses in Figure 7D. All gene sets used to run GSEA are listed in Supplementary Table 1.

1.6. What are the “technical differences” mentioned in the first section of Results? sc vs sn RNAseq?

Technical differences between human and mouse datasets include differences in sample preparation such as single-cell vs single-nuclei RNAseq, differences in brain regions where human microglia were isolated from, and overall viability of isolated microglia from the human brain that vary across multiple experiments. Please see the revised text in the end of paragraph titled: Geneset enrichment analysis of TF regulons nominates candidate DLAM TFs in human and mouse macrophages/microglia (Page 7).

1.7. For reference, what does a “control” look like in the analysis in Supp Fig 1a-b? (i.e. non-prominent peaks)

Supplementary Figure 1A-B shows that SPI1 and BHLHE41 regulons are also enriched in those respective motifs, suggesting that the genes we identify as regulated by SPI1 and BHLHE41 are also potentially bound by those TFs in the promoter. Below we include two negative examples. Although JUNB and MAFB are both nominated as DLAM regulators through co-expression, their motifs do not seem to be enriched in the promoters of DLAM genes (as DLAM genes we tested human LAM reported by (Jaitin et al. 2019) Dataset S6, FDR < 0.05, and mouse DAM reported by (Keren-Shaul et al. 2017) Table S2)(Figure 3A-D). Additionally, selecting TFs that are expressed highly and specifically in microglia and examining their motif instances in SPI1 and

BHLHE41 regulons also suggest that the regulons for these two TFs are enriched for their putative direct targets.

As an example, when we look at motif instances of ATF3 in BHLHE41 and SPI1 regulons, we do not see a peak, indicating lack of enrichment:

When we examine at CEBPB motif instances in BHLHE41 and SPI1 regulons, we see that those motif instances are depleted:

This suggests that SPI1 and BHLHE41 regulons include putative direct targets of SPI1 and BHLHE41 as indicated by the enrichment of their motif instances in regulon gene promoters.

1.8. The last sentence of the Results section “ BHLHE41 and SPI1/PU.1 likely regulate” seems to be a possible overstatement.

Analysis of available ChIPseq in microglial cells (mouse) showed PU.1/SPI1 binds ~50% of DAM genes (Keren-Shaul et al., 2017 (Keren-Shaul et al. 2017), Table S2). In addition, our previous work on PU.1 in BV2 mouse microglia showed that reduction of PU.1 (associated with AD protection) is positively correlated with DAM gene expression and related pathways (Pimenova et al. 2021). Finally, we added an example of epigenomic tracks highlighting open-chromatin regions that contain a BHLHE41 motif in DLAM gene promoters for CD63 and CTSB loci in mouse and human macrophages and microglia (Page 9). Please see revised Figure 3E–F.

1.9. THP-1 macrophages should be defined at first mention (i.e. what is THP-1).

We added a brief explanation that THP-1 is a monocytic leukemia line. Please see paragraph *Knockdown of BHLHE40/41 partially recapitulates the LAM...*, first sentence. (Page 17)

1.10. What were the media conditions that the iMGLs were cultured in? specifically the lipid content of the media – FBS concentration? Lipid supplementation? How might this be affecting the results?

iPSC-derived microglia (iMGLs) were cultured without FBS or FCS. Media formulation was used exactly as previously described by McQuade et al. (McQuade et al. 2018) THP-1 macrophages also had a limited amount of serum. Briefly, THP-1 monocytes were cultured and differentiated in the presence of serum (10% FBS) and then rested without serum in the presence of 1% BSA fraction V for 24h, followed by transfection 48h. There was no additional lipid supplementation in any cell culture at any time point therefore we think lipid abundance in both culture conditions for iMGL and THP-1 was similar and did not affect the final results. We added this statement in the method section describing culture conditions. (Page 31-32)

1.11. The authors mention “likely due to low statistical power (figure 5a, ...” What is the actual n =5 here? It is 5 wells of the same iMGL line, not 5 distinct iMGL lines, correct? If 5 wells/samples, that would seem to reduce variation dramatically.

The reviewer is correct, N in this figure is independent iMGL differentiations (different days of iMGL differentiation and collection), and not independent lines (donors or clones). Although some genes whose expression level was assessed by qPCR did not reach statistical significance, the effect (increased expression) was present. We also calculated the effect size for each gene/comparison and listed it in Supplementary File 1. For some genes the effect size is small and therefore would require increased N ($N > 5$) to achieve sufficient statistical power.

1.12. What is known regarding the functional consequences of BHLHE40/41 mutations in AD? This should be addressed in the Discussion and perhaps introduction as well.

As mentioned in the main text, BHLHE40 is a candidate AD risk gene by virtue of the fact it resides in the vicinity of an AD risk locus recently identified in an African-American AD GWAS study (Kunkle et al. 2021). Disease-associated GWAS loci are mostly non-coding common genetic variants that modulate disease risk typically by regulating the expression of one or more nearby genes. It took several years and research groups (including ours) to nominate, based not only on proximity but also on the integration of functional genomic evidence, the most likely causal genes at about half of the AD risk loci identified by much larger and older GWAS studies in European individuals (Novikova et al. 2021), a population for which functional genomics datasets are already

available. Unfortunately, functional follow-up of the genetic associations identified in African-American AD GWAS studies is still lacking also because functional genomics datasets in minority populations are very scarce. We are in the process of generating such datasets from monocyte-derived macrophages obtained from African-American individuals, but it is an effort that will take several years to complete. Therefore, at the present time, proximity is the only criterion (albeit usually ~70% accurate) used to nominate BHLHE40 as a candidate AD risk gene.

1.13. How do the authors think APOE4 fits into their findings here? Particularly, the findings from this same group (and others) that show E4 expression is tied to increased lipid droplet formation, but also increased cytokine release and decreased efflux.

We thank the reviewer for this comment, this is a very interesting point. Our large RNAseq-studies of APOE44 population and isogenic iPSC-derived microglia suggested that APOE44 iMGL showed 1) decreased LXR- and Mit/TFE-mediated responses, 2) lower expression of core DLAM genes, 3) fewer cells in the DAM cluster in scRNAseq. Therefore, our current efforts are to increase LXR activity using either novel pharmacological approaches or a genetic inactivation of BHLHE40/41 which are LXR and Mit/TFE repressors. We hypothesize that inactivation of BHLHE40/41 in APOE44 iMGL will rescue lipid accumulation and lysosomal storage deficits as well as normalize the proportion of microglia in the DAM cluster observed in our preliminary scRNAseq. One caveat is related to increased accumulation of lipid droplets in the risk (APOE44 iMGL) and protective model (BHLHE40/41-DKO). We think that accumulation of lipid droplets in APOE44 may be an adaptation to increased cholesterol biosynthesis and decreased cholesterol efflux. In that setting, LD is one of the buffering mechanisms sequestering the excess of toxic free cholesterol. We believe that when we facilitate lipid and lysosomal clearance processes in APOE44 iMGL, we will be able to restore lipid droplets phenotype or at least not increase the LD content. Future studies should also focus on profiling LD content in BHLHE40/41 and APOE44 iMGL to understand their composition.

1.14. References 60 and 76 appear to be the same paper

We thank the reviewer for that comment, we have now corrected the reference list.

Reviewer #2 (Remarks to the Author):

In the present manuscript Podlesny-Drabiniok, Novikova et al. postulate that the transcription factors BHLHE40 and BHLHE41 at least partially drive the gene expression signature of LAMs (Lipid-Associated Macrophages). Both TFs are helix-loop-helix proteins with repressive functions. The authors used publicly-available transcriptomic data from different myeloid cell types (e.g., microglia, Kupffer cells, macrophages from adipose tissue) in the context of different diseases (Alzheimer's disease, Obesity, Atherosclerosis and Steatosis) from human and mouse tissue. The authors defined human and mouse LAM signature genes and based on this gene list and nominated a gene regulatory network consisting of 74 transcription factors as potential drivers of the LAM phenotype. Next, the authors show that the BHLHE41 DNA-binding motif is enriched in promoters of LAM genes and AD risk SNPs. Knockout of BHLHE41 and the double knockout of BHLHE40/41 in human iPSC-derived microglia like cells resulted in a partially up-and down-

regulation of genes associated with LAM gene expression signature. In addition, functional assays showed increased cholesterol efflux, lipid droplets and lysosomal acidification/proteolysis. Finally, in vivo double knockout of Bhlhe40 and Bhlhe41 resulted in an altered gene expression signature of mouse microglia partially recapitulating the LAM transcriptional response.

Overall, the study is of interest because it nominates potential drivers of the LAM phenotype and the major conclusions as stated in the abstract are supported by experimental data. There are some points for which clarifications would strengthen the manuscript.

Major points:

2.1. The finding that loss of function of BHLHE40/41 results in increased cholesterol efflux and lipid droplet formation is of interest. The increase in cholesterol efflux is consistent with the proposed role of BHLHE40/41 as negative regulators of LXRs. However, the increase in lipid droplets is not necessarily linked to increased functions of LXRs. Although not central to the major conclusions, the manuscript would be strengthened by lipidomic analyses to establish the lipid classes that are associated with these lipid droplets. This is important because lipid droplets in macrophage foam cells of atherosclerotic lesions are enriched in cholesterol esters, whereas those that accumulate in microglia in the context of aging are enriched for triglycerides, diglycerides, and phospholipids with very little cholesterol ester. Mechanisms underlying accumulation of these different lipid classes are not the same, with accumulation of TGs invoking the SREBP pathway. This could potentially involve LXRs via LXR activation of SREBP1c, the signal for which should be evident in the data. Thus, further information on lipid content and LXR/SREBP target genes would be of interest.

We thank the reviewer for this interesting question and we agree that lipidomic analysis would be of interest but, at the present time, very challenging because bulk lipidomics is not very sensitive to compositional changes in small microglial subpopulations such as the DAM-like clusters that typically account for 5-10% of all iMGLs *in vitro*.

We also want to point out that we performed the experiment where we tested whether increased lipid droplets content in BHLHE40/41-DKO and single KO is LXR-dependent. To this end, we treated cells with an LXR antagonist (GSK2033) and we measured the level of lipid droplets by flow cytometry using BODIPY. We found that the level of lipid droplets in KO and DKO is decreased after inhibiting LXR with an antagonist as compared to KO and DKO treated with a vehicle. Interestingly, the level of lipid droplets in KO and DKO treated with LXR antagonists reached the level of lipid droplets in WT suggesting increased lipid droplets content in KO and DKO is mediated by increased LXR (see Supplementary Figure 5 and Page 15).

Although, our BHLHE40/41-KO and DKO lines showed increased LXR activity such as elevated levels of APOE, ABCA1, cholesterol efflux and lipid droplets content, in our RNAseq experiment we did not observe an increased expression of genes involved in "Regulation of cholesterol biosynthesis by SREBP SREBF" (Normalized enrichment score, NES = 0.69 for DKO vs WT contrast). In addition, based on our gene sets enrichment analysis, we found negative enrichment of pathways involved in "Triglyceride metabolic process" (NES = -1.33) which may suggest that the SREBP axis is not activated in DKO iMGLs. We also compared the induction of lipid

catabolism, sterol transport, fatty acid metabolism gene sets/pathways in DKO iMGLs with the transcriptomic effect of synthetic LXR agonist (TO901317, 10uM, 48h, Goate lab unpublished data). We found the LXR agonist has a stronger effect on all aforementioned processes as compared to DKO.

2.2. It is difficult to reconstruct how the authors arrived at the different human and mouse “LAM” marker genes. A list of LAM marker genes is not provided in the manuscript. The number of LAM genes in mouse seems to be very high with 1,453 genes as written in Figure 7B. The authors should clarify this.

We thank the reviewer for this clarifying question. Since the term “LAM” used in various contexts in the previous version of the manuscript seems to be causing confusion, we have decided to use the term DLAMs to collectively refer to subpopulations of microglia and peripheral macrophages such as DAM, LAM, TREM2high among others, which share similar gene expression signatures and cellular responses to damage of lipid-rich tissues. We use the term “LAM” when using lipid-associated macrophages gene signature from Jaitin et al., (Jaitin et al. 2019) Dataset S6, FDR < 0.05. We used this signature specifically in Figure 4A, Supplementary Figure 7a and Figure 3A,C. Which is specified in Supplementary Table 1 and listed in the main text and each figure legend when this gene signature has been used. We use the term “DAM” referring to the mouse DAM signature (not just the top marker genes) profiled by Keren-Shaul et al (Keren-Shaul et al. 2017) Supplementary Table S3 (FDR <0.05). We used it in Figure 7A.

Due to the fact that RRHO analysis requires genes to be ranked by a signed differential gene expression statistic (for example $\log_2FC * \log_{10}(\text{adj.P-value})$), we have used Table S3 from (Keren-Shaul et al. 2017) listing all the DAM genes along with the two necessary parameters (\log_2FC and adj.P-value) for the RRHO analysis. Table S2 from (Keren-Shaul et al. 2017) contains a list of only 500 genes (frequently referred to as “DAM genes”) but does not include the \log_2FC parameter that is necessary to perform RRHO analysis. To investigate whether the DAM genes from Table S2 from (Keren-Shaul et al. 2017) are also positively enriched in mouse DKO transcriptome, we performed GSEA using DAM upregulated genes and DAM downregulated genes from Table S2. We found significant positive enrichment of DAM upregulated genes and significant negative enrichment of DAM downregulated genes in mouse DKO transcriptome. These results are added to Figure 7D and Page 20. Please also see section 1.5 of this document.

2.3. In studies using *Bhlhe40*^{-/-}*Bhlhe41*^{-/-} double knockout mice, the authors should comment on the extent to which *Bhlhe40/41* are expressed by other brain cell types. If they are expressed, the authors should include the possibility that some of the effects of the knockout could be non-cell autonomous in the limitations section. It would also be of interest to know whether lipid droplets were observed in microglia from these mice.

We thank the reviewer for this comment and we have now added the following information in the Study Limitations section: “Another limitation of the study is the mouse model with a global knockout of *Bhlhe40* and *Bhlhe41* (mouse DKO) which may affect other cell types such as astrocytes where *Bhlhe40* and *Bhlhe41* are also expressed. We cannot exclude that there is a non-cell-autonomous effect mediated possibly by astrocytes lacking *Bhlhe40/41* that would affect DKO microglia” (Page 26). *Bhlhe40* expression is low in microglia/macrophages but it is significantly induced in disease-associated microglia (DAM) (Friedman et al. 2018; Keren-Shaul et al. 2017). *Bhlhe40* is highly expressed in astrocytes and endothelial cells. *Bhlhe41* is highly expressed by myeloid cells (microglia, macrophages, neutrophils) and to a lesser extent by astrocytes (data source: <https://brainnaseq.org/>).

Although we have not done any experiments checking the level of lipid droplets in mouse microglia lacking Bhlhe40/41, we expect increased lipid droplets content in DKO microglia. There are couple of reasons for that: 1) It has already been shown that alveolar macrophages from DKO mice accumulate lipid droplets Rauschmeier et al., (Rauschmeier et al. 2019) (please see Figure 4, also pasted below) 2) GSEA analysis of transcriptome from microglia lacking Bhlhe40/41 (this study) showed that GOCC_LIPID_DROPLET and GOBP_REGULATION_OF_LIPID_STORAGE are significantly and positively enriched with Normalized Enrichment Score (NES = 1.59 NOM P value = 0.007 and NES = 1.47 NOM P value = 0.047, respectively).

Figure 4C from (Rauschmeier et al. 2019)

GSEA (mouse microglia lacking Bhlhe40/41 as compared to WT)

Minor comment:

2.4. RNA-seq data is presented as RRHO heatmaps. It would be informative to present the data also a Volcano plots depicting fold change and adj p-value and to state, what the fold-cut off were to determine differentially expressed genes. This can be shown as supplementary data.

We thank the reviewer for this comment. DLAM genes did not reach statistical significance for differential gene expression when filtered with standard RNA-seq criteria (FDR < 0.05). Therefore, as we did previously with APOE4 microglia TCW et al., (Tcw et al. 2022), we moved away from threshold-based methods toward ranked-based methods (RRHO and - in the revised version - GSEA) that do not rely on arbitrary cutoffs and take full advantage of the information contained in whole-transcriptome profiles. The reason why the effect sizes of DLAM marker genes are small in DKO cells may be due to the fact that typically DLAMs make for only a small proportion of microglia/macrophages both *in vitro* and *in vivo*. Therefore, scRNA-seq may be more sensitive for the detection of gene expression changes associated with compositional shifts in microglia/macrophage subpopulations upon genetic inactivation of BHLHE40/41. We are currently performing such experiments in the context of follow-up studies and future manuscripts.

Reviewer #3 (Remarks to the Author):

Communications – Review March 2023

BHLHE40/41 regulate macrophage/microglia responses associated with Alzheimer's disease and other disorders of lipid-rich tissues

The manuscript by Podlesny-Drabiniok and Novikova et al. outlines the core gene-regulatory-network (GRN) function of the transcription factors; BHLHE40/41, in microglia and macrophages in lipid-rich environments in various diseases. Combining multiple publicly available single-cell and single-nucleus RNA-seq datasets, as well as ATAC-seq, the authors show that Bhlhe40/41 is a core transcriptional regulators of LAM microglia/macrophages and provide evidence of functional responses of microglia upon genetic perturbation of Bhlhe40/41. Overall, the manuscript provides an important insight to the transcriptional impact of Bhlhe40/41 on cholesterol metabolism and lysosomal activity in microglia. However, the initial GRN analysis leaves several questions to be addressed as indicated below:

Major comments:

3.1. The methods for the reconstruction of gene-regulatory-network must be described in more detail. Figure 1 shows the consolidated results for the GRN analysis. It is not clear whether metacells were constructed separately for the different expression matrix files (different datasets) or processed together as one merged expression matrix. If individual networks were built for each expression matrix, the intermediate results should be presented.

We thank the reviewer for this clarifying question. Metacells were generated for every dataset separately and a network was generated for every individual dataset as well. Meta-analysis was done using all of the networks combined and is shown in Figure 1. We have now edited the methods section to describe the network reconstruction procedure in more detail. We also added

intermediate results, TFs nominated by each dataset prior to meta-analysis, to Supplementary Table 1.

3.2. Why were all metacells created using standard parameters. The authors should provide evidence that standard parameters provided the most robust K-means clustering across different datasets.

Parameters were k-means clustering were different for each dataset, they were chosen such as the number of neighbors * the number of metacells was as close to the total number of cells as possible to avoid the same cell being included in multiple metacells. In our conversations with the authors of PISCES, they suggested selecting the number of neighbors > 10 and number of metacells > 100 and that this selection did not dramatically affect the outcome of ARACNE. In our own original analyses with changing the number of neighbors and number of metacells, the list of potential regulators of the LAM signature was not dramatically affected. We now include the parameters that were used for metacells generation for each dataset on our github page. We edited the Methods section to address this question and it now reads:

“The pipeline for reconstruction of metacells was adopted from the PISCES tool(Obradovic, Vlahos, et al. 2021) and can be found at our github page https://github.com/marcoralab/bhlhe_manuscript. Briefly, this approach constructs a kNN graph using the data, partitions the data into an appropriate number of metacells, taking into account the desired number of neighbors. It then aggregates the counts from closest neighbors into MetaCells. The MetaCells function was used in the following manner: *MetaCells(data, dist.mat, numNeighbors = numNeighbors, subSize = subSize)*. *numNeighbors* and *subSize* for each dataset are provided on our github page.”

3.3. Most of the input datasets for GRN analysis are not associated to Alzheimer’s disease and/or microglia. Yet, the authors are using the consolidated results in Figure 1 to justify why BHLHE40/41 is studied in microglia. In silico, the authors need to provide more insight of the GRN analysis in microglia.

We thank the reviewer for this suggestion. In the manuscript, we attempted to focus on both peripheral tissue-resident macrophages and microglia, given that the lipid-associated macrophage response occurs in various disease contexts, such as Alzheimer’s disease, fatty liver disease, obesity, lung fibrosis and others. Hence, some of our datasets are associated with Alzheimer's disease (e.g., human AD brains from Zhou et al and APP knockin mouse data from Sala Frigerio et al) and others are associated with other disorders of lipid-rich tissues (e.g., human liver cirrhosis from Ramachandran et al and mouse induced nonalcoholic steatohepatitis from Xiong et al). Indeed, our downstream validation focuses not only on human and mouse microglia, but also on human macrophages shown in Supplementary Figures 6 and 7, attempting to show that this response is present in different types of macrophages. However, we now include all the TFs nominated by each individual dataset (Supplementary Table 1), including microglial datasets, so that the interested reader can further examine microglia-specific findings if needed.

3.4. The authors implement the PISCES tool to strengthen the expression profile of low expression genes. The tool is not yet peer-reviewed and it is not clear why the authors only used single-cell and single-nucleus RNA-seq although bulk RNA-seq provides much better sequencing depth.

We thank the reviewer for this thoughtful question. Unfortunately, given the small number of microglia in the brain, ~5-10 %, bulk brain RNA-seq datasets fail to capture microglial activation states. Indeed, there have been many studies that looked at differential expression between AD and control brains, for example, and the findings mostly relate to the increased number of microglia in a disease brain as opposed to capturing specific activation states. Hence, we focused on single-cell and single-nucleus datasets to leverage a higher diversity of transcriptional responses that are captured in those datasets, despite lower sequencing depth.

Given our collaboration with the authors, we used a small portion of the PISCES pipeline that deals specifically with metacells reconstruction and obtained the code from the team a long time prior to publication. We now share the code used for metacells reconstruction on our github page https://github.com/marcoralab/bhlhe_manuscript. In addition, this pipeline has been used and published (and some of its findings experimentally validated) in several peer-reviewed articles (Hawley et al. 2023; Pan et al. 2020; Obradovic, Chowdhury, et al. 2021).

3.5. It is not clear to what extent the LAM signature is present in the datasets used. How many cells are positive for the LAM signature, and do they cluster separately from DAM microglia? Where only LAM signature microglia/macrophages implemented for the GRN analysis?

We thank the reviewer for this clarifying question. We would first like to clarify the DAM/LAM nomenclature. In this manuscript, we attempted to communicate that several populations carrying different names, such as disease-associated microglia (DAM) (Keren-Shaul et al. 2017), TREM2^{HI} macrophages identified in atherosclerotic plaques (Cochain et al. 2018) and lipid-associated macrophages (LAM) in adipose tissue (Jaitin et al. 2019) are similar to each other in that they mount similar transcriptional and cellular response to damage of lipid-rich tissues. Using datasets from relevant disease and control tissues, we aimed to identify transcriptional regulators that might be shared between these populations, focusing specifically on disease-associated microglia population (DAM) in the brain, TREM2^{HI} population in atherosclerotic plaques, and lipid-associated macrophages (LAM) in adipose tissue. In the revised manuscript we collectively referred to this microglia/macrophage subpopulations as DLAMs (please see response to comment 1.3). In our computational analyses, we make sure that the TFs we nominate exhibited enrichment of their regulons for all the three DLAM genesets mentioned above. Our question of interest is to identify transcriptional regulators of the DLAM response, so we focused on those signatures in our GRN analysis.

By the nature of the datasets we selected, most of them showed positive expression of DLAM markers. Below we show the findings from original authors, when possible, or include UMAPs that we generated ourselves. Starting with mouse datasets, adipose macrophages reported in Jaitin et al (Jaitin et al. 2019) express LAM markers, such as *Ctsb*, *Gpnmb*, *Lgals3*, *Apoe* and others.

Xiong et al (Xiong et al. 2008) reported macrophages in the liver that responded to the induction of nonalcoholic steatohepatitis that express Trem2 and Gpnmb among other DLAM markers.

Ramachandran et al (Ramachandran et al. 2019) reported scar-associated macrophages in both mouse and humans that highly express DLAM makers, such as TREM2 and CD9, as shown below in their cross-species integrative analysis.

Cochain et al (Cochain et al. 2018) reported macrophages in atherosclerotic aortas that again highly express DLAM markers, such as Trem2, Cd9 and Ctsd among others.

Lin et al (Lin et al. 2019) also identified a similar Trem2^{high} macrophage population in their atherosclerotic model, where these macrophages highly express Spp1, Cd9 and other DLAM markers.

Sala Frigerio et al (Sala Frigerio et al. 2019) uses Alzheimer's disease mouse models that have been previously reported to show DAM activation, expressing Apoe, Spp1, Gpnmb and other LAM markers. The authors term this population Activated response microglia or ARMs and characterize this activation state more deeply than the original DAM study (Keren-Shaul et al. 2017).

Given previous reports that aged mice show an activation state similar to that observed in mouse models of neurodegeneration albeit to a much smaller extent (Keren-Shaul et al. 2017), we included a microglial dataset from aging mouse brains. Please see microglial UMAPs below showing LAM gene expression in Ximerakis et al dataset (Ximerakis et al. 2019); a cluster of cells expressing markers, such as *Gpnmb*, *Lpl* and *Spp1* as well as an increased expression of *Apoe* can be seen.

Given that the DLAM response is, at least in part, TREM2 dependent, we included data from a demyelination mouse model with wild-type and knockout Trem2. Please see microglial UMAPs below showing DLAM gene expression in Nugent et al (Nugent et al. 2020) dataset similarly showing a cluster of cells expressing *Gpnmb*, *Lpl* and *Spp1* as well as an increased expression of *Apoe*.

Human DAM and TREM2hi signatures have not been convincingly described yet. Hence, in our analyses, we lifted 2 mouse signatures to the human genome, the DAM signature (Keren-Shaul et al. 2017) and the TREM2^{HI} signature ((Cochain et al. 2018), along with using a human LAM signature that was derived from human macrophages in obese individuals (Jaitin et al. 2019). We assume that although the mouse and human activation signatures will surely be, at least in part, different, we decided to leverage human datasets from the same disease conditions as our mouse data, e.g. Alzheimer's disease, atherosclerosis and non-alcoholic steatohepatitis. We also used a dataset with a very large number of primary human microglia from fresh resected tissues. Assuming that at least some of the markers are conserved, we subsequently aim to use co-expression patterns in microglial cells to identify regulators of the DLAM signature. Please see UMAP plots showing microglial expression of select DLAM genes from human datasets used in this study.

Mathys et al

Olah et al

Zhou et al

Jaitin et al

Fernandez et al

MacParland et al

Ramachandran et al

3.6. The authors mention efferocytosis as a common hallmark pathway for LAM microglia/macrophages, however, do not show any data supporting this statement.

We thank the reviewer for this comment and we apologize that it was not well explained. We think that efferocytosis, understood as clearance of lipid-rich cellular debris (i.e. myelin fragments, apoptotic cells and synapses, dystrophic neurites, amyloid plaques, etc.), is a main microglial/macrophage process affected by genetic variants associated with AD risk. Efferocytosis is a four step mechanism that includes proper Chemotaxis and Recognition of extracellular waste (step1), Engulfment that requires actin polymerization and cytoskeleton rearrangements to take up extracellular debris (step 2), Digestion that comprises degradation of engulfed material in the endolysosomal system (step 3) and Adaptation that includes activation of transcription factors that increase phagocytosis, cholesterol efflux and storage, lysosomal biogenesis, bioenergetics and other metabolic processes (step 4). AD risk genes are enriched in each of these steps including genes with rare coding variants such as *TREM2*, *ABCA7*, *ABI3*, *PLCG2*, and genes implicated by common non-coding variants (e.g. *ABCA1*, *ZYX*, *BIN1*, *RIN3*, *MEF2C*, *SPI1*). Genes identified through a variety of different approaches including coloc, TWAS, and SMR show that candidate causal genes for AD fall into one of these four steps supporting our hypothesis that abnormal microglial efferocytosis plays an important role in the etiology of AD. We have presented this hypothesis with supporting genetic and experimental evidence in three reviews (Romero-Molina et al. 2022; Andrews et al. 2023; Podleśny-Drabiniok, Marcora, and Goate 2020). Consistent with this hypothesis is the observation that several DLAM genes are AD risk genes and that DLAM genes are enriched in several pathways (for example phagocytosis and lipid metabolism) that are core components of efferocytosis (Deczkowska et al., (Deczkowska et al. 2018)). Interestingly, from the perspective of AD evolution, primate microglia (compared to rodent microglia) are also enriched for AD risk and efferocytosis genes (Geirsdottir et al., (Geirsdottir et al. 2020)).

3.7. When generating the short list of candidate transcription factors, the researchers require the transcription factor to appear in at least half the human and mouse networks and present in human microglia. The authors need to provide data visualization displaying which networks contain these transcription factors.

We have now included results from individual network enrichment analyses in Supplementary Table 1, where the reader can see which TF was nominated by which study/network.

3.8. The reason for the expression-based selection of top transcription factor markers is not clear. Protein levels of transcription factors should be considered. Even low expressed transcription factors could have biological and functional meaning. Selecting only the top 11 candidates before investigating AD risk allele enrichment and TF binding sites could skew the results and hide other potentially important transcription factors. The authors must present a less biased approach.

We absolutely agree with the reviewer that protein levels of transcription factors are important; however, in this study we focus on single-cell and single-nucleus RNA-seq data because the population we are most interested in (lipid-associated macrophages) was identified at the mRNA level. The expression filter that we implemented is very relaxed; we only require a TF to be expressed at a level ≥ 1 TPM in human microglia; in fact, when examining the list of TFs pre and

post-filter, all 11 TFs are expressed in human microglia ≥ 1 TPM. Given this observation, we removed the description of the filter from the manuscript. The text now reads:
“High confidence TFs were selected if they were 1) enriched for all three LAM genesets in at least half of human and mouse networks and 2) conserved between species.”

Since microglia represent a small fraction of brain-resident cells and proteins from more abundant cells are much more likely to be captured, we avoided the use of whole brain proteomics datasets. Hence, we used datasets, where microglia were purified prior to the proteomics experiment. Unfortunately, although there are multiple mouse microglial proteomics datasets published, the number of detected proteins in microglia is still quite low. For example, we examined a dataset by Rangaraju et al (Rangaraju, Dammer, Raza, Gao, et al. 2018), where the authors performed a TMT proteomics on isolated Cd11b+ microglial cells from wild-type, LPS-stimulated and Alzheimer’s disease mouse models (5xFAD). The authors detected 4,133 proteins across the three experimental groups, but none of our 11 top transcriptional regulator candidates were detected. Out of 74 mouse TFs reported in Figure 1, only 11 were detected in the dataset. Human microglia proteomics datasets are therefore, unfortunately, limited at this time. A recent preprint by Lloyd et al (Lloyd et al. 2022) reported more than 9,000 microglial proteins, but the paper has not been peer reviewed and the data are not yet available. Taken together, given a relatively small number of proteins that are detected in current proteomics experiments, a filter on proteomic expression is not optimal and expression of many transcriptional regulators cannot yet be assessed.

3.9. The significance values of enrichment for BHLHE40 binding to ATAC seq peaks for LAM genes is missing. The authors should demonstrate more transparency and show HOMER results for Figure 3.”

We would like to clarify the data shown in Figure 3. We are not performing an enrichment analysis in Figure 3. Instead, we are taking 11 TFs that we are nominating through co-expression analysis and quantifying the number of LAM genes. We used LAM genes from (Jaitin et al. 2019) Dataset S6, FDR Adj.P-value < 0.05) that contain a motif of that TF in their promoter (Figure 3A, C) and we used DAM genes from (Keren-Shaul et al. 2017) Table S2 . Hence, we used HOMER only to pinpoint ATAC-seq regions that are positive for the motif of interest; we have not performed a global motif enrichment analysis because it does not address the question we are interested in. With the analysis presented in Figure 3, we are trying to assess which of the TFs that are nominated through co-expression could also potentially bind LAM genes. Additionally, we are not showing a percent of LAM gene promoters that contain a BHLHE40 motif; BHLHE41 is the TF that is nominated through the network approach.

3.10 There are several datasets on BHLHE40 ChIP-seq, which should be investigated and used to validate potential BHLHE40 binding site enrichment at LAM genes (Citation: 59, PMID: 31061528).

We thank the review for this suggestion. We would like to highlight that our network analyses nominate BHLHE41 in particular, not BHLHE40. Our analyses of open chromatin regions in DLAM gene promoters presented in Figure 3 also highlights that a large proportion of LAM genes contain

a BHLHE41 motif (not BHLHE40). Although these TFs are closely related and demonstrated a level of compensatory activity in previous studies, we show individual and double KD/KO in our validation studies. However, our computational analyses suggest that BHLHE41 could have different/broader binding patterns than BHLHE40. In our analyses of human microglial open chromatin regions, we identified around 30K BHLHE41 proxy-binding sites compared to only ~8.7K BHLHE40 proxy-binding sites. Indeed, more than 70% of LAM genes contain a BHLHE41 motif (Figure 1), while only 28% contain a BHLHE40 motif. This suggests that there may be fewer BHLHE40 binding sites around the genome. Unfortunately, there are no studies to date that have profiled open chromatin regions and Bhlhe40/Bhlhe41 binding pattern, making it difficult for us to validate our proxy-binding sites for these TFs. However, we analyzed two additional datasets in support of our observation that Bhlhe40 has more limited binding throughout the genome than Bhlhe41. We analyzed ChIP-seq data from large peritoneal mouse macrophages (Rauschmeier et al. 2019) and identified only 1,046 *Bhlhe40* peaks with 6% of DAM genes having a Bhlhe40 binding site in their promoter (as a DAM we used (Keren-Shaul et al. 2017), Table S2). We also analyzed ATAC-seq data from mouse bone-marrow derived macrophages (Daniel et al. 2020), which showed a similar pattern with only 3,854 Bhlhe40 proxy-binding sites out of ~77K open chromatin regions with 17% of DAM genes having a Bhlhe40 binding site in their promoter (in comparison, Bhlhe41 proxy-binding sites were present in around 40% of DAM genes in mouse BMDMs as shown in Figure 3) (as a DAM we used (Keren-Shaul et al. 2017), Table S2). This observation could be driven by the lower expression of BHLHE40; for example, in human microglia, BHLHE41 is expressed almost 14 times higher than BHLHE40 (Gosselin et al. 2017). Interestingly, Bhlhe40 is upregulated in mouse DAM signature, suggesting that detecting binding patterns of Bhlhe40 might be harder in baseline tissues. Taken together, we would like to highlight that BHLHE41 has been nominated by our network analyses and its motif is contained in a large proportion of DLAM gene promoters. Although BHLHE40 is a closely related TF, its potential binding patterns do seem to differ from BHLHE41 and they do not suggest potential direct binding to DLAM gene promoters. We would, however, like to highlight the limitations of using motifs as opposed to ChIP-seq data to ascertain TF binding sites.

Minor comments:

3.11. It is not clear how the Human and mouse LAM TFs in Figure 1 were clustered. Were the LAM TFs ranked, and if so, using what method?

Heatmap was created using the R pheatmap package with clustering_method = "complete".

3.12. The authors should provide all code used for the GRN analysis for more transparency.

We have now created a github page https://github.com/marcoralab/bhlhe_manuscript where an example of GRN generation is included to enhance transparency.

3.13. The authors should comment on the low MI scores between LAM genes and TFs in mice in Figure 1.

There could be many potential reasons why some MI scores are low between a subset of DLAM genes and TFs in mouse datasets. One could be because some of the human datasets are quite large (Olah et al., n.d.; Mancuso et al. 2022), providing more microglial cells and allowing for detection of weaker co-expression patterns between DLAM genes and TFs. Another reason could

be that the mouse and human DLAM signatures depicted in Figure 1, although significantly overlapping, are distinct, which can also drive differences observed in Figure 1.

3.14. Peak profiles of ATAC-seq for Bhlhe40 motif enrichment sites at LAM genes should be presented, as well as ChIP of Bhlhe40.

We thank the reviewer for this suggestion. We would like to highlight that our network analyses nominate BHLHE41, not BHLHE40. Although these TFs are closely related and demonstrate a level of compensatory mechanisms in certain contexts, our analyses nominated BHLHE41 in particular through co-expression followed by quantification of proxy-binding at LAM gene promoters. As described in our response to comment 3.10, BHLHE40 likely has different binding patterns than BHLHE41 and proxy-binds a much smaller proportion of LAM gene promoters than BHLHE41. Since BHLHE41 ChIP-seq data in human macrophages have not been generated (likely due to the fact that a quality antibody for BHLHE41 is not commercially available), we now include epigenomic tracks, highlighting open chromatin regions in LAM gene promoters in human and mouse microglia and macrophages that contain a BHLHE41 motif in Figure 3E-F.

3.15. Why did the authors choose to select LPS and IL4 treated microglia?

We have not chosen IL4 or LPS-treated microglia but we cited one article that used 43 existing GEO microarray transcriptomes of Cd11b+ microglia including in vivo microglia from AD mouse models and in vitro microglia stimulated with LPS and IL4 (Rangaraju, Dammer, Raza, Rathakrishnan, et al. 2018). Since this created an impression that we included only LPS and IL4 microglia signatures, we re-worded that sentence, please see Introduction, paragraph started with “The DLAM response like other ...” (Page 4)

3.16. Using stratified LD score regression to perform AD heritability analysis it is not clear if the P-values are adjusted for false discovery?

The P-values shown in Figure 2 are not FDR adjusted; however, dark blue bars indicate significant enrichments (FDR < 0.05), bars in light red indicate nominally significant enrichments (P-value < 0.05), while gray bars indicate non-significant enrichment.

Second revision

Reviewer #1 (Remarks to the Author):

The authors have done a tremendous job of objectively, carefully, and exhaustively responding to this Reviewer's comments. They gave extensive responses to my questions/concerns, have revised the manuscript accordingly, and have added important new data and clarifications where needed. I thoroughly enjoyed reading this manuscript and would like to commend the authors on a beautiful, impactful study.

We thank the reviewer for the comments that helped to improve the final version of the manuscript

Reviewer #2 (Remarks to the Author):

The authors have satisfactorily addressed the major and minor concerns that I raised in the initial review. The major conclusions of the manuscript are reasonably well supported and the findings represent a significant contribution to defining genes and pathways regulating lipid metabolism and lysosomal function in microglia that are relevant to neurodegenerative diseases.

We thank the reviewer for the comments that helped to improve the final version of the manuscript

Reviewer #3 (Remarks to the Author):

The authors fully addressed majority of my main critiques. There are few remaining minor comments to be addressed before publication as follow:

1. Krasemann et al., (PMID: 28930663) identified and described *Bhlhe40*, which is regulated by APOE signaling, and specifically induced in plaque associated *Clec7a+* microglia isolated from AD mice. However, *Bhlhe41* was not affected. The authors used double-KO approach. This should be acknowledged and discussed.

2. On P23 the authors describe: "Furthermore, a recent study showed that APOE risk-increasing (APOE $\epsilon 4/\epsilon 4$, similar to APOE, TREM2, and PLCG2 loss-of-function mutations) and risk-decreasing (APOE $\epsilon 2/\epsilon 2$) genotypes are associated with decreased and increased DAM transcriptional ...[68]. The authors should discuss two comprehensive new studies recently published Nature Immunology, which employed complementary gain-of-function and loss-of-function approaches to provide critical new evidence that APOE4 impairs MGnD response to neurodegeneration including identification of the mechanism related to induction of TGFb-mediated checkpoints which block MGnD response, including induction of SPI1 (PMIDs: 37749326, 37857825).

We thank the reviewer for the comments that helped to improve the final version of the manuscript.

In mouse studies, we have used the DKO approach because both *Bhlhe40* and *Bhlhe41* were nominated through our GRN as candidate regulators of mouse DLAM responses. Additionally, analysis of global transcriptomic changes in alveolar macrophages isolated from *Bhlhe40/41*-DKO mice showed an induction of several DLAM genes including *ApoE*, *Trem2*, and *Lpl*(Rauschmeier et al. 2019). Finally, our human iMGL and MAC data showed that genetic inactivation or reduction of both *BHLHE40/41* recapitulate an induction of DLAM responses both transcriptionally and functionally as compared to single KO. We think the effect of *Bhlhe40* and *Bhlhe41* may be additive and therefore we decided to use the DKO approach. This is also clarified in the paragraph titled: *Knockout of Bhlhe40/41 partially recapitulates DLAM transcriptional responses in mouse microglia*

We thank the reviewer for highlighting the two recently published articles that shed a light on the role of APOE44 in microglia in the context of AD pathology. We have mentioned these two studies in the discussion.

- Andrews, Shea J., Alan E. Renton, Brian Fulton-Howard, Anna Podlesny-Drabiniok, Edoardo Marcora, and Alison M. Goate. 2023. "The Complex Genetic Architecture of Alzheimer's Disease: Novel Insights and Future Directions." *EBioMedicine* 90 (April): 104511.
- Claes, Christel, Emma Pascal Danhash, Jonathan Hasselmann, Jean Paul Chadarevian, Sepideh Kiani Shabestari, Whitney E. England, Tau En Lim, et al. 2021. "Plaque-Associated Human Microglia Accumulate Lipid Droplets in a Chimeric Model of Alzheimer's Disease." *Molecular Neurodegeneration*. <https://doi.org/10.1186/s13024-021-00473-0>.
- Cochain, Clément, Ehsan Vafadarnejad, Panagiota Arampatzi, Jaroslav Pelisek, Holger Winkels, Klaus Ley, Dennis Wolf, Antoine-Emmanuel Saliba, and Alma Zernecke. 2018. "Single-Cell RNA-Seq Reveals the Transcriptional Landscape and Heterogeneity of Aortic Macrophages in Murine Atherosclerosis." *Circulation Research* 122 (12): 1661–74.
- Daniel, Bence, Zsolt Czimmerer, Laszlo Halasz, Pal Boto, Zsuzsanna Kolostyak, Szilard Poliska, Wilhelm K. Berger, et al. 2020. "The Transcription Factor EGR2 Is the Molecular Linchpin Connecting STAT6 Activation to the Late, Stable Epigenomic Program of Alternative Macrophage Polarization." *Genes & Development* 34 (21-22): 1474–92.
- Deczkowska, Aleksandra, Hadas Keren-Shaul, Assaf Weiner, Marco Colonna, Michal Schwartz, and Ido Amit. 2018. "Disease-Associated Microglia: A Universal Immune Sensor of Neurodegeneration." *Cell* 173 (5): 1073–81.
- Dolan, Michael-John, Martine Therrien, Saša Jereb, Tushar Kamath, Vahid Gazestani, Trevor Atkeson, Samuel E. Marsh, et al. 2023. "Exposure of iPSC-Derived Human Microglia to Brain Substrates Enables the Generation and Manipulation of Diverse Transcriptional States in Vitro." *Nature Immunology* 24 (8): 1382–90.
- Fernandez, Dawn M., Adeeb H. Rahman, Nicolas F. Fernandez, Aleksey Chudnovskiy, El-Ad David Amir, Letizia Amadori, Nayaab S. Khan, et al. 2019. "Single-Cell Immune Landscape of Human Atherosclerotic Plaques." *Nature Medicine* 25 (10): 1576–88.
- Friedman, Brad A., Karpagam Srinivasan, Gai Ayalon, William J. Meilandt, Han Lin, Melanie A. Huntley, Yi Cao, et al. 2018. "Diverse Brain Myeloid Expression Profiles Reveal Distinct Microglial Activation States and Aspects of Alzheimer's Disease Not Evident in Mouse Models." *Cell Reports* 22 (3): 832–47.
- Geirsdottir, Laufey, Eyal David, Hadas Keren-Shaul, Assaf Weiner, Stefan Cornelius Bohlen, Jana Neuber, Adam Balic, et al. 2020. "Cross-Species Single-Cell Analysis Reveals Divergence of the Primate Microglia Program." *Cell* 181 (3): 746.
- Gerrits, Emma, Nieske Brouwer, Susanne M. Kooistra, Maya E. Woodbury, Yannick Vermeiren, Mirjam Lambourne, Jan Mulder, et al. 2021. "Distinct Amyloid- β and Tau-Associated Microglia Profiles in Alzheimer's Disease." *Acta Neuropathologica* 141 (5): 681–96.
- Gosselin, David, Dylan Skola, Nicole G. Coufal, Inge R. Holtman, Johannes C. M. Schlachetzki, Eniko Sajti, Baptiste N. Jaeger, et al. 2017. "An Environment-Dependent Transcriptional Network Specifies Human Microglia Identity." *Science* 356 (6344). <https://doi.org/10.1126/science.aal3222>.
- Hasselmann, Jonathan, Morgan A. Coburn, Whitney England, Dario X. Figueroa Velez, Sepideh Kiani Shabestari, Christina H. Tu, Amanda McQuade, et al. 2019. "Development of a Chimeric Model to Study and Manipulate Human Microglia In Vivo." *Neuron* 103 (6): 1016–33.e10.
- Hawley, Jessica E., Aleksandar Z. Obradovic, Matthew C. Dallos, Emerson A. Lim, Karie Runcie, Casey R. Ager, James McKiernan, et al. 2023. "Anti-PD-1 Immunotherapy with Androgen Deprivation Therapy Induces Robust Immune Infiltration in Metastatic Castration-Sensitive Prostate Cancer." *Cancer Cell* 41 (11): 1972–88.e5.
- Jaitin, Diego Adhemar, Lorenz Adlung, Christoph A. Thaiss, Assaf Weiner, Baoguo Li, Hélène Descamps, Patrick Lundgren, et al. 2019. "Lipid-Associated Macrophages Control Metabolic Homeostasis in a Trem2-Dependent Manner." *Cell* 178 (3): 686–98.e14.
- Kamphuis, Willem, Lieneke Kooijman, Sjoerd Schettters, Marie Orre, and Elly M. Hol. 2016.

- “Transcriptional Profiling of CD11c-Positive Microglia Accumulating around Amyloid Plaques in a Mouse Model for Alzheimer’s Disease.” *Biochimica et Biophysica Acta* 1862 (10): 1847–60.
- Keren-Shaul, Hadas, Amit Spinrad, Assaf Weiner, Orit Matcovitch-Natan, Raz Dvir-Szternfeld, Tyler K. Ulland, Eyal David, et al. 2017. “A Unique Microglia Type Associated with Restricting Development of Alzheimer’s Disease.” *Cell* 169 (7): 1276–90.e17.
- Kunkle, Brian W., Michael Schmidt, Hans-Ulrich Klein, Adam C. Naj, Kara L. Hamilton-Nelson, Eric B. Larson, Denis A. Evans, et al. 2021. “Novel Alzheimer Disease Risk Loci and Pathways in African American Individuals Using the African Genome Resources Panel: A Meta-Analysis.” *JAMA Neurology* 78 (1): 102–13.
- Lin, Jian-Da, Hitoo Nishi, Jordan Poles, Xiang Niu, Caroline Mccauley, Karishma Rahman, Emily J. Brown, et al. 2019. “Single-Cell Analysis of Fate-Mapped Macrophages Reveals Heterogeneity, Including Stem-like Properties, during Atherosclerosis Progression and Regression.” *JCI Insight* 4 (4). <https://doi.org/10.1172/jci.insight.124574>.
- Lloyd, Amy F., Anna Martinez-Muriana, Pengfei Hou, Emma Davis, Renzo Mancuso, Alejandro J. Brenes, Ivana Geric, et al. 2022. “Deep Proteomic Analysis of Human Microglia and Model Systems Reveal Fundamental Biological Differences of in Vitro and Ex Vivo Cells.” *bioRxiv*. <https://doi.org/10.1101/2022.07.07.498804>.
- Mancuso, Renzo, Nicola Fattorelli, Anna Martinez-Muriana, Emma Davis, Leen Wolfs, Johanna Van Den Daele, Ivana Geric, et al. 2022. “A Multi-Pronged Human Microglia Response to Alzheimer’s Disease A β Pathology.” *bioRxiv*. <https://doi.org/10.1101/2022.07.07.499139>.
- Marschallinger, Julia, Tal Iram, Macy Zardeneta, Song E. Lee, Benoit Lehallier, Michael S. Haney, John V. Pluvineau, et al. 2020. “Lipid-Droplet-Accumulating Microglia Represent a Dysfunctional and Proinflammatory State in the Aging Brain.” *Nature Neuroscience* 23 (2): 194–208.
- McQuade, Amanda, Morgan Coburn, Christina H. Tu, Jonathan Hasselmann, Hayk Davtyan, and Mathew Blurton-Jones. 2018. “Development and Validation of a Simplified Method to Generate Human Microglia from Pluripotent Stem Cells.” *Molecular Neurodegeneration* 13 (1): 67.
- Novikova, Gloriia, Manav Kapoor, Julia Tcw, Edsel M. Abud, Anastasia G. Efthymiou, Steven X. Chen, Haoxiang Cheng, et al. 2021. “Integration of Alzheimer’s Disease Genetics and Myeloid Genomics Identifies Disease Risk Regulatory Elements and Genes.” *Nature Communications* 12 (1): 1610.
- Nugent, Alicia A., Karin Lin, Bettina van Lengerich, Steve Lianoglou, Laralynne Przybyla, Sonnet S. Davis, Ceyda Llapashtica, et al. 2020. “TREM2 Regulates Microglial Cholesterol Metabolism upon Chronic Phagocytic Challenge.” *Neuron* 105 (5): 837–54.e9.
- Obradovic, Aleksandar, Nivedita Chowdhury, Scott M. Haake, Casey Ager, Vinson Wang, Lukas Vlahos, Xinzhen V. Guo, et al. 2021. “Single-Cell Protein Activity Analysis Identifies Recurrence-Associated Renal Tumor Macrophages.” *Cell* 184 (11): 2988–3005.e16.
- Obradovic, Aleksandar, Lukas Vlahos, Pasquale Laise, Jeremy Worley, Xiangtian Tan, Alec Wang, and Andrea Califano. 2021. “PISCES: A Pipeline for the Systematic, Protein Activity-Based Analysis of Single Cell RNA Sequencing Data.” *bioRxiv*. <https://doi.org/10.1101/2021.05.20.445002>.
- Olah, Marta, Vilas Menon, Naomi Habib, Mariko Taga, Christina Yung, Maria Cimpean, Anthony Khairalla, et al. n.d. “A Single Cell-Based Atlas of Human Microglial States Reveals Associations with Neurological Disorders and Histopathological Features of the Aging Brain.” <https://doi.org/10.1101/343780>.
- Pan, Huize, Chenyi Xue, Benjamin J. Auerbach, Jiaxin Fan, Alexander C. Bashore, Jian Cui, Dina Y. Yang, et al. 2020. “Single-Cell Genomics Reveals a Novel Cell State During Smooth Muscle Cell Phenotypic Switching and Potential Therapeutic Targets for Atherosclerosis in Mouse and Human.” *Circulation* 142 (21): 2060–75.

- Pimenova, Anna A., Manon Herbinet, Ishaan Gupta, Saima I. Machlovi, Kathryn R. Bowles, Edoardo Marcora, and Alison M. Goate. 2021. "Alzheimer's-Associated PU.1 Expression Levels Regulate Microglial Inflammatory Response." *Neurobiology of Disease* 148 (January): 105217.
- Podleśny-Drabiniok, Anna, Edoardo Marcora, and Alison M. Goate. 2020. "Microglial Phagocytosis: A Disease-Associated Process Emerging from Alzheimer's Disease Genetics." *Trends in Neurosciences* 43 (12): 965–79.
- Ramachandran, P., R. Dobie, J. R. Wilson-Kanamori, E. F. Dora, B. E. P. Henderson, N. T. Luu, J. R. Portman, et al. 2019. "Resolving the Fibrotic Niche of Human Liver Cirrhosis at Single-Cell Level." *Nature* 575 (7783): 512–18.
- Rangaraju, Srikant, Eric B. Dammer, Syed Ali Raza, Tianwen Gao, Hailian Xiao, Ranjita Betarbet, Duc M. Duong, et al. 2018. "Quantitative Proteomics of Acutely-Isolated Mouse Microglia Identifies Novel Immune Alzheimer's Disease-Related Proteins." *Molecular Neurodegeneration* 13 (1): 34.
- Rangaraju, Srikant, Eric B. Dammer, Syed Ali Raza, Priyadharshini Rathakrishnan, Hailian Xiao, Tianwen Gao, Duc M. Duong, et al. 2018. "Identification and Therapeutic Modulation of a pro-Inflammatory Subset of Disease-Associated-Microglia in Alzheimer's Disease." *Molecular Neurodegeneration* 13 (1): 24.
- Rauschmeier, René, Charlotte Gustafsson, Annika Reinhardt, Noelia A-Gonzalez, Luigi Tortola, Dilay Cansever, Sethuraman Subramanian, et al. 2019. "Bhlhe40 and Bhlhe41 Transcription Factors Regulate Alveolar Macrophage Self-Renewal and Identity." *The EMBO Journal*, August, e101233.
- Romero-Molina, Carmen, Francesca Garretti, Shea J. Andrews, Edoardo Marcora, and Alison M. Goate. 2022. "Microglial Efferocytosis: Diving into the Alzheimer's Disease Gene Pool." *Neuron* 110 (21): 3513–33.
- Sala Frigerio, Carlo, Leen Wolfs, Nicola Fattorelli, Nicola Thrupp, Iryna Voytyuk, Inga Schmidt, Renzo Mancuso, et al. 2019. "The Major Risk Factors for Alzheimer's Disease: Age, Sex, and Genes Modulate the Microglia Response to A β Plaques." *Cell Reports* 27 (4): 1293–1306.e6.
- Tcw, Julia, Lu Qian, Nina H. Pipalia, Michael J. Chao, Shuang A. Liang, Yang Shi, Bharat R. Jain, et al. 2022. "Cholesterol and Matrisome Pathways Dysregulated in Astrocytes and Microglia." *Cell* 185 (13): 2213–33.e25.
- Ximerakis, Methodios, Scott L. Lipnick, Brendan T. Innes, Sean K. Simmons, Xian Adiconis, Danielle Dionne, Brittany A. Mayweather, et al. 2019. "Single-Cell Transcriptomic Profiling of the Aging Mouse Brain." *Nature Neuroscience* 22 (10): 1696–1708.
- Xiong, Shigang, Hongyun She, An-Sheng Zhang, Jiaohong Wang, Hasmik Mkrtchyan, Alla Dynnyk, Victor R. Gordeuk, Samuel W. French, Caroline A. Enns, and Hidekazu Tsukamoto. 2008. "Hepatic Macrophage Iron Aggravates Experimental Alcoholic Steatohepatitis." *American Journal of Physiology. Gastrointestinal and Liver Physiology* 295 (3): G512–21.